# Zebra: In-Context Generative Pretraining for Solving Parametric PDEs

**Louis Serrano** [1]  **Armand Kassaï Koupaï** [1]  **Thomas X Wang** [1]  **Pierre Erbacher** [2]  **Patrick Gallinari** [1,3]

## Abstract

Solving time-dependent parametric partial differential equations (PDEs) is challenging for data-driven methods, as these models must adapt to variations in parameters such as coefficients, forcing terms, and initial conditions. State-of-the-art neural surrogates perform adaptation through gradient-based optimization and meta-learning to implicitly encode the variety of dynamics from observations. This often comes with increased inference complexity. Inspired by the in-context learning capabilities of large language models (LLMs), we introduce Zebra, a novel generative auto-regressive transformer designed to solve parametric PDEs without requiring gradient adaptation at inference. By leveraging in-context information during both pre-training and inference, Zebra dynamically adapts to new tasks by conditioning on input sequences that incorporate context example trajectories. As a generative model, Zebra can be used to generate new trajectories and allows quantifying the uncertainty of the predictions. We evaluate Zebra across a variety of challenging PDE scenarios, demonstrating its adaptability, robustness, and superior performance compared to existing approaches.

## 1. Introduction

A major challenge for training neural solvers for time dependent partial differential equations (PDEs) or more generally for modeling spatio-temporal dynamics is to capture the variety of behaviors arising from complex physical phenomena. In particular, neural solvers, trained from a limited number of situations often fail to generalize to new physical contexts and situations (Chen et al., 2018; Raissi et al., 2019;

Our code is available on GitHub. [1]Sorbonne Université, CNRS, ISIR, 75005 Paris, France [2]Naver Labs Europe, France [3]Critéo AI Lab, Paris, France. Correspondence to: Louis Serrano <louis.serrano@sorbonne-universite.fr>, Patrick Gallinari <patrick.gallinari@sorbonne-universite.fr>.

Li et al., 2021; Koupaï et al., 2024).

We address the parametric PDE problem (Cohen & Devore, 2015), where the goal is to train models on a limited set of scenarios representing a given physical phenomenon so that they can generalize across a wide range of new contexts, including different PDE parameters. These parameters may encompass initial and boundary conditions, physical coefficients, and forcing terms. In this work, we focus on purely data-driven approaches that do not incorporate prior knowledge of the underlying equations.

A basic approach to this problem is to sample from the distribution of physical parameters, i.e., to train on different instances of a PDE characterized by varying parameter values, with the goal of generalizing to unseen instances. This approach relies on an i.i.d. assumption and requires a training set that adequately represents the distribution of the underlying dynamical system—a condition that is often difficult to satisfy in practice due to the complexity of physical phenomena. Other approaches explicitly condition on specific PDE parameters (Brandstetter et al., 2022b; Takamoto et al., 2023), relying on the availability of such prior knowledge. This assumes that a physical model of the observed system is known, making the incorporation of PDE parameters into neural solvers challenging beyond physical quantities. Moreover, in many cases, this prior knowledge is incomplete or entirely unknown.

An alternative approach involves adaptation to new PDE instances by leveraging observations from novel *environments*. Here we consider that an environment is characterized by a set of parameters. For data-driven models, adaptation is often performed through fine tuning, which usually requires a significant amount of examples for the new environment. This is for example the setting adopted in many recent development of foundation models (Subramanian et al., 2023; Herde et al., 2024; McCabe et al., 2023). This involves training a large model on a variety of physics-based numerical simulations, each requiring a large amount of simulations with the expectation that it will generalize.

More principled frameworks for adaptation leverage meta-learning, where the model is trained on simulations corresponding to different environments—i.e., varying PDE parameter values—so that it can quickly adapt to new and unseen PDE simulation instances using a few trajectory ex-

amples (Park et al., 2023; Kirchmeyer et al., 2022; Yin et al., 2022). These flexible methods rely on gradient updates for adaptation, adding computational overhead.

We explore a new direction for adaptation inspired by the successes of in-context learning (ICL) in natural language processing (NLP) and its demonstrated ability to generalize to downstream tasks without retraining (Brown et al., 2020; Touvron et al., 2023). We propose a framework, denoted `Zebra`, relying on ICL for solving parametric PDEs with new parameter values, without any additional update of the model parameters.

As for ICL in NLP, the model is trained to generate appropriate responses given context examples and a query. The context examples will be trajectories from the same dynamics starting from different initial conditions. The query will consist for example of a new initial state condition, that will serve as inference starting point for the new forecast. The proposed model is inspired from NLP approaches: it is a causal generative model that processes discrete token sequences encoding observations. It is trained to model the trajectory distributions of parameteric PDEs. This approach offers key advantages and greater flexibility compared to existing methods. It can leverage contexts of different types and sizes, requires only a few context examples to adapt to new dynamics, and allows us to cover a wide range of situations. It provides enhanced capabilities compared to more classical deterministic forecasting models. Notably, generative probabilistic models have been developed for physical problems such as weather forecasting (Price et al., 2023; Couairon et al., 2024) and even PDE solving (Lippe et al., 2023), demonstrating superior performance and capabilities over their deterministic counterparts. However, their setting is different, as they rely on diffusion models and are neither designed for adaptation nor intended to address the parametric PDE problem.

Some recent works have also explored adaptation through in-context learning for dynamics modeling. The closest to ours is probably (Yang et al., 2023), which also targets adaptation to multiple environments of an underlying physical dynamics through prompting with examples. Their model employs a specific deterministic encoder-decoder architecture and is limited to 1D ODEs or sparse 2D data due to scalability issues. More details and further references are provided in Appendix A.

On the technical side, `Zebra` introduces a novel generative autoregressive solver for parametric PDEs. It employs an encode-generate-decode framework: first, a vector-quantized variational auto-encoder (VQ-VAE) (Oord et al., 2017) is learned to compress physical states into discrete tokens and to decode it back to the original physical space. Next, a generative autoregressive transformer is pre-trained with arbitrary size context examples of trajectories using

a next token objective. At inference, `Zebra` can handle varying context sizes for conditioning and supports uncertainty quantification, enabling generalization to unseen PDE parameters without gradient updates.

Our main contributions include:

- We introduce a generative autoregressive transformer for modeling physical dynamics. It operates on compact discretized representations of physical state observations. This framework represents the first successful application of causal generative modeling using quantized representations of physical systems.

- To harness the in-context learning strengths of autoregressive transformers, we develop a new pretraining strategy that conditions the model on example trajectories with similar underlying dynamics but different initial conditions.

- Our generative model predicts trajectory distributions. This provides a richer information than deterministic auto-regressive models. This comes with enhanced capabilities including more accurate predictions, uncertainty measures, or the ability to sample and generate new trajectories conditioned on some examples.

- We propose an accelerated inference procedure that is orders of magnitude faster than existing adaptation methods.

- We evaluate `Zebra` in a one-shot adaptation setting, where it must infer the dynamics from a single context trajectory and unroll the dynamics from a new query initial condition (i.e., a single snapshot). The predicted trajectory is compared to a target trajectory governed by the same underlying dynamics as the context but starting from the query initial condition. `Zebra`'s performance is benchmarked against domain adaptation baselines specifically trained for this task, and it consistently outperforms gradient-based adaptation methods on 2D datasets.

## 2. Problem setting

### 2.1. Solving parametric PDEs

We aim to solve parametric time-dependent PDEs beyond the typical variation in initial conditions. Our goal is to train models capable of generalizing across a wide range of PDE parameters. To this end, we consider time-dependent PDEs with different initial conditions, and with additional degrees of freedom, namely: (1) coefficient parameters — such as fluid viscosity or advection speed — denoted by vector $\mu$ ; (2) boundary conditions $\mathcal{B}$, e.g. Neumann or Dirichlet; (3) forcing terms $\delta$, including damping parameter or sinusoidal forcing with different frequencies. We

denote $\xi := \{\mu, \mathcal{B}, \delta\}$ and we define $\mathcal{F}_\xi$ as the set of PDE solutions corresponding to the PDE parameters $\mu$, boundary conditions $\mathcal{B}$ and forcing term $\delta$, and refer to $\mathcal{F}_\xi$ as a PDE environment. Formally, a solution $\boldsymbol{u}(x,t)$ within $\mathcal{F}_\xi$ satisfies:

$$
\begin{aligned}
\frac{\partial \boldsymbol{u}}{\partial t} = F\big(\delta, \mu, t, x, \boldsymbol{u}, \frac{\partial \boldsymbol{u}}{\partial x}, \frac{\partial^2 \boldsymbol{u}}{\partial x^2}, \dots \big), & \\
\forall x \in \Omega, \forall t \in (0, T] & \\
\mathcal{B}(\boldsymbol{u})(x,t) = 0, \quad \forall x \in \partial\Omega, \forall t \in (0, T] & \\
\boldsymbol{u}(0, x) = \boldsymbol{u}^0, \quad \forall x \in \Omega &
\end{aligned}
\tag{1}
$$

where $F$ is a function of the solution $\boldsymbol{u}$ and its spatial derivatives on the domain $\Omega$, and also includes the forcing term $\delta$ ; $\mathcal{B}$ is the boundary condition constraint (e.g., spatial periodicity, Dirichlet, or Neumann) that must be satisfied at the boundary of the domain $\partial\Omega$; and $\boldsymbol{u}^0$ is the initial condition sampled with a probability measure $\boldsymbol{u}^0 \sim \boldsymbol{p}^0(.)$.

### 2.2. Adaptation for parametric PDE

Solving time-dependent parametric PDEs requires developing neural solvers capable of generalizing to a whole distribution of PDE parameters. In practice, changes in the PDE parameters often lead to distribution shifts in the trajectories which makes the problem challenging. Different directions are currently being explored and are briefly reviewed below. We focus on pure data-driven approaches that do not make use of any prior knowledge on the equations. We make the assumption that the models are learned from numerical simulations so that it is possible to generate from multiple parameters. This emulates real situations where for example, a physical phenomenon is observed in different contexts.

**Fine tuning pre-trained models**  The classical strategy for adapting to new settings is to fine tune models that have been pretrained on a distribution of the PDE parameters. This approach often relies on large fine tuning samples and involves updating all or a subset of parameters (Subramanian et al., 2023; Herde et al., 2024). We do not consider this option that has been shown to underperform SOTA adaptation approaches (Koupaï et al., 2024).

**Gradient-based adaptation**  A more flexible approach relies on adaptation at inference time through meta-learning. It posits that a set of environments $e$ are available from which trajectories are sampled, each environment $e$ being defined by specific PDE parameter values (Zintgraf et al., 2019a; Kirchmeyer et al., 2022). The model is trained from a sampling from the environments distribution to adapt fast to a new environment. The usual formulation is to learn shared and specific environment parameters $\mathcal{G}_{\theta+\theta_\xi}$, where $\theta$

and $\theta_\xi$ are respectively the shared and specific parameters. At inference, for a new environment, only a small number of parameters $\theta_\xi$ is adapted from a small sample of observations. This family of method will be our reference baseline in the following.

## 3. Zebra Framework

We introduce `Zebra`, a novel framework designed to solve parametric PDEs through in-context learning and flexible conditioning. `Zebra` utilizes a generative autoregressive transformer to model partial differential equations (PDEs) within a compact, discrete latent space. A spatial CNN encoder is employed to map physical spatial observations into these latent representations, while a CNN decoder accurately reconstructs them. Our pretraining pipeline consists of two key stages: 1) Learning a finite vocabulary of physical phenomena, and 2) Training the transformer using an in-context pretraining strategy, enabling the model to effectively condition on contextual information. At inference, `Zebra` allows to perform in-context learning from context trajectories as illustrated in Figure 1.

### 3.1. Learning a finite vocabulary of physical phenomena

In order to leverage the auto-regressive transformer architecture and adopt a next-token generative pretraining, we need to convert physical observations into discrete representations. We do not quantize the observations directly but rather quantize compressed latent representations by employing a VQVAE (Oord et al., 2017), an encoder-decoder architecture with a quantizer component. Our encoder spatially compresses the input function $\boldsymbol{u}^t$ through a convolutional model $\mathcal{E}_w$, which maps the input to a continuous latent variable $\mathbf{z}^t = \mathcal{E}_w(\boldsymbol{u}^t)$. The latent variables are then quantized to a vector of discrete codes $\mathbf{z}_q^t$ using a codebook $\mathcal{Z}$ of size $K = |\mathcal{Z}|$ through the quantization component $q$. For each spatial code $\mathbf{z}_{[ij]}^t$ in $\mathbf{z}_q^t$, the nearest codebook entry $z_k$ is selected. The decoder $\mathcal{D}_\psi$ reconstructs the signal $\hat{\boldsymbol{u}}^t$ from the quantized latent codes $\hat{\mathbf{z}}_q^t$. Both models are jointly trained to minimize the reconstruction error between the function $\boldsymbol{u}^t$ and its reconstruction $\hat{\boldsymbol{u}}^t = \mathcal{D}_\psi \circ q \circ \mathcal{E}_w(\boldsymbol{u}^t)$.

Once this training step is done, we can tokenize a trajectory $\boldsymbol{u}^{t:t+m\Delta t}$ by applying our encoder in parallel on each timestamp to obtain vectors of discrete codes $\mathbf{z}_q^{t:t+m\Delta t}$ and retrieve the corresponding index entries $s^{t:t+m\Delta t}$ from the codebook $\mathcal{Z}$. Similarly, we detokenize discrete indices with the decoder. We provide a brief description of the VQVAE model and details on its architecture in Appendix C.

### 3.2. In-context modeling

We design sequences that enable `Zebra` to perform in-context learning on trajectories that share underlying dy-

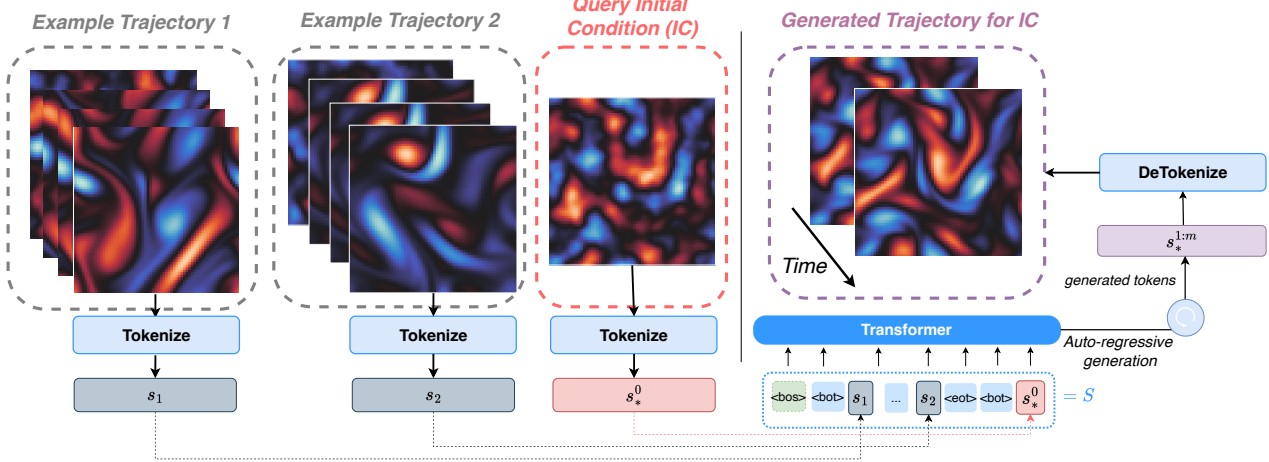

*Figure 1.* Zebra's inference pipeline from context trajectories. The model is provided with a set of example trajectories ($u_1$ and $u_2$) that share the same underlying dynamics (e.g., identical PDE parameters) but differ in initial conditions. Given a new query initial condition $u_*^0$, Zebra aims to generate the corresponding trajectory that follows the same physical behavior. Each trajectory and initial condition is first tokenized into flattened index sequences $s_1$, $s_2$, and $s_*^0$, which are concatenated using a specific formatting scheme. The transformer then autoregressively generates the remaining tokens to obtain a plausible prediction. Finally, the generated indices are detokenized to reconstruct the trajectory in physical space.

namics with different initial states. To incorporate varying amounts of contextual information, we draw a number $n \in \{1, n_{\max}\}$, then sample $n$ trajectories sharing the same dynamics, each with $m$ snapshots starting from time $t$, denoted as $(\boldsymbol{u}_1^{t:t+m\Delta t}, \ldots, \boldsymbol{u}_n^{t:t+m\Delta t})$. These trajectories are tokenized into index representations $(s_1^{t:t+m\Delta t}, \ldots, s_n^{t:t+m\Delta t})$, which are flattened into sequences $s_1, \ldots, s_n$, maintaining the temporal order from left to right. In practice, we fix $n_{\max} = 6$ and $m = 9$.

Since our model operates on tokens from a codebook, we found it advantageous to introduce *special tokens* to structure the sequences. The tokens <bot> (beginning of trajectory) and <eot> (end of trajectory) define the boundaries of each trajectory within the sequence. Furthermore, as we sample sequences with varying context sizes, we maximize the utilization of the transformer's context window by stacking sequences that could also represent different dynamics. To signal that these sequences should not influence each other, we use the special tokens <bos> (beginning of sequence) and <eos> (end of sequence). The final sequence design is:

$$S = \texttt{<bot>}[s_1]\texttt{<eot>}\ldots\texttt{<bot>}[s_n]\texttt{<eot>}$$

And our pretraining dataset is structured as follows:

$$\texttt{<bos>}[S_1]\texttt{<eos>}\ldots\texttt{<bos>}[S_l]\texttt{<eos>}$$

### 3.3. Next-token pretraining

The transformer is trained using self-supervised learning on a next-token prediction task with teacher forcing (Radford

et al., 2018). Given a sequence $S$ of discrete tokens of length $N$, the model is optimized to minimize the negative log-likelihood (cross-entropy loss):

$$\mathcal{L}_{\text{Transformer}} = -\mathbb{E}_S \sum_{i=1}^{N} \log p(S_{[i]}|S_{[i'<i]}),$$

where the model learns to predict each token $S_{[i]}$ conditioned on all previous tokens $S_{[i'<i]}$. Due to the transformer's causal structure, earlier tokens in the sequence are not influenced by later ones, while later tokens benefit from more context, allowing for more accurate predictions. This structure naturally supports both generation in a one-shot and few-shot setting within a unified framework. Our transformer implementation is based on the Llama architecture (Touvron et al., 2023). Additional details can be found in Appendix C. Up to our knowledge, this is the first adaptation of generative auto-regressive transformers to the modeling of physical dynamics.

### 3.4. Flexible inference: prompting and sampling

In this section, we outline the inference pipeline for Zebra across various scenarios. For simplicity, we assume that all observations have already been tokenized and omit the detokenization process. Let $s_*$ represent the target token sequence to be predicted.

• **Prompting** with $n$ examples and an *initial condition*: The prompt is structured as $S = \texttt{<bos><bot>}[s_1^{0:m\Delta t}]\ldots[s_n^{0:m\Delta t}]\texttt{<eot><bot>}[s_*^0],$

allowing the model to adapt based on the provided examples and initial condition.

- **Prompting** with $n$ examples and $\ell$ *frames*: This setup combines context from multiple trajectories with the initial timestamps, structured as

$$S = \texttt{<bos>} \ \texttt{<bot>} \ [s_1^{0:m\Delta t}] \ \ldots \ [s_n^{0:m\Delta t}] \ \texttt{<eot>}$$
$$\texttt{<bot>} \ [s_*^{0:\ell\Delta t}]$$

At inference, we adjust the *temperature parameter* $\tau$ of the classifier layer to calibrate the level of diversity of the next-token distributions. The temperature $\tau$ scales the logits $y_i$ before the softmax function :

$$p(S_{[i]} = k | S_{[i' < i]}) = \text{softmax}\left(\frac{y_k}{\tau}\right) = \frac{\exp\left(\frac{y_k}{\tau}\right)}{\sum_j \exp\left(\frac{y_j}{\tau}\right)}$$

When $\tau > 1$, the distribution becomes more uniform, encouraging exploration, whereas $\tau < 1$ sharpens the distribution, favoring more deterministic predictions. During training, it is kept fixed at $\tau = 1$.

## 4. Experiments

In this section, we experimentally validate that our framework enables one-shot adaptation at inference. We follow the *pretraining* procedure outlined in Section 3 for each dataset described in Section 4.1 and evaluate Zebra across distinct scenarios. We first assess its performance in the one-shot setting for in-distribution parameters, comparing it to adaptation-based baselines (Section 4.2). We then examine its generalization in out-of-distribution settings in Section 4.3. Next we illustrate and analyze the generative abilities of Zebra through two example tasks: *uncertainty quantification* and *new trajectory generation* in Section 4.4, with further analysis in Appendix D.2 and Appendix D.3. Finally, in Section 4.5, we show how we can drastically accelerate the adaptation at inference compared to gradient-based methods. More results are provided in Appendix D, including: deterministic pretraining (Appendix D.1), scaling behavior with the number of training samples (Appendix D.4), the effect of codebook size (Appendix D.6), reconstruction capabilities (Appendix D.7), and the impact of the number of context examples (Appendix D.8).

### 4.1. Datasets details

As in Kirchmeyer et al. (2022), we generate data in batches, where each batch of trajectories corresponds to a single environment and shares the same PDE parameters while having different initial conditions. We consider various factors of variation across multiple datasets. To assess the generalization ability of our model across a wide range of scenarios, we use a significantly larger number of environments—far

exceeding those in previous studies and available simulation datasets (Yin et al. (2022), Kirchmeyer et al. (2022), Blanke & Lelarge (2023), Nzoyem et al. (2024)). We conduct experiments across seven datasets: five 1D—*Advection*, *Heat*, *Burgers*, *Wave-b*, *Combined*—and two 2D—*Wave 2D*, *Vorticity 2D*. These datasets were selected to encompass different physical phenomena and test generalization under changes to various types of PDE parameters, as described below. The spatial resolution is set to 256 for 1D datasets and $64 \times 64$ for 2D datasets. We subsample each trajectory to keep 10 snapshots, except for the *Wave-b* dataset, where we retain 15.

**Varying PDE coefficients**   The changing factor is the set of coefficients $\mu$ in Equation (1). For *Burgers*, *Heat*, and *Vorticity 2D* equations, the viscosity coefficient $\nu$ varies across environments. For *Advection*, the advection speed $\beta$ changes. In *Wave-c* and *Wave-2D*, the wave's celerity $c$ is unique to each environment, and the damping coefficient $k$ varies across environments in *Wave-2D*. In the *Combined* equation, three coefficients $(\alpha, \beta, \gamma)$ vary, each influencing different derivative terms respectively: $-\frac{\partial u^2}{\partial x}, +\frac{\partial^2 u}{\partial x^2}, -\frac{\partial^3 u}{\partial x^3}$ on the right-hand side of Equation (1).

**Varying boundary conditions**   In this case, the varying parameter is the boundary condition $\mathcal{B}$ from Equation (1). For *Wave-b*, we explore two types of boundary conditions—Dirichlet and Neumann—applied independently to each boundary, resulting in four distinct environments.

**Varying forcing term**   The varying parameter is the forcing term $\delta$ in Equation (1). In *Burgers* and *Heat*, the forcing terms vary by the amplitude, frequency, and shift coefficients: $\delta(t, x) = \sum_{j=1}^{5} A_j \sin\left(\omega_j t + 2\pi \frac{l_j x}{L} + \phi_j\right)$.

A detailed description of the datasets is provided in Appendix B, while Table 6 summarizes the number of environments used during training, the number of trajectories sharing the same dynamics, and the varying PDE parameters across environments. For testing, all methods are evaluated on trajectories with new initial conditions in previously unseen environments. These unseen environments include trajectories with both novel initial conditions and varying parameters, which remain within the training distribution for in-distribution evaluation and extend beyond it for out-of-distribution testing. For each testing, we use 120 unseen environments for the 2D datasets and 12 for the 1D datasets, with each environment containing 10 trajectories.

Regarding computational resources, training the VQ-VAE in 1D takes approximately 4 hours on an RTX 24 GB GPU, while the transformer component requires around 15 hours. In the 2D setting, both training times increase to approximately 20 hours each on a single A100 80 GB GPU.

*Table 1.* One-shot adaptation. Conditioning from a similar trajectory. Test results in relative L2 on the trajectory. '–' indicates inference has diverged.

| | *Advection* | *Heat* | *Burgers* | *Wave b* | *Combined* | *Wave 2D* | *Vorticity 2D* |
|---|---|---|---|---|---|---|---|
| CAPE | 0.00941 | 0.223 | 0.213 | 0.978 | **0.00857** | – | – |
| CODA | **0.00687** | 0.546 | 0.767 | 1.020 | 0.0120 | 0.777 | 0.678 |
| `[CLS]` ViT | 0.140 | **0.136** | 0.116 | 0.971 | 0.0446 | 0.271 | 0.972 |
| `ViT-in-context` | 0.0902 | 0.472 | 0.582 | 0.472 | 0.0885 | 0.390 | 0.173 |
| `Zebra` | 0.00794 | 0.154 | **0.115** | **0.245** | 0.00965 | **0.207** | **0.119** |

## 4.2. In-distribution generalization

**Setting** We evaluate `Zebra`'s ability to perform in-context learning by *leveraging example trajectories that follow the same underlying dynamics as the target*. Formally, in the $n$-shot adaptation setting, we assume access to a set of $n$ context trajectories $\{\boldsymbol{u}_1^{0:m\Delta t}, \ldots, \boldsymbol{u}_n^{0:m\Delta t}\}$ at inference time, all of which belong to the same dynamical system $\mathcal{F}_\xi$. The goal of the adaptation task is to accurately predict a future trajectory $\boldsymbol{u}_*^{\Delta t:m\Delta t}$ from a new initial condition $\boldsymbol{u}_*^0$, knowing that the underlying target dynamics is shared with the provided context example trajectories.

**Sampling** For `Zebra`, we use here a random sampling procedure at inference for generating the next tokens for all datasets, setting a low temperature ($\tau = 0.1$) to prioritize accuracy over diversity. Predictions are generated using a single sample under this configuration.

**Baselines** We evaluate Zebra against 4 baselines, CODA (Kirchmeyer et al., 2022) and CAPE (Takamoto et al., 2023), two SOTA adaptation methods. We also compare to two specifically designed ViT architectures: `[CLS]` ViT that performs adaptation by learning a `[CLS]` embedding and `ViT-in-context` designed for in-context training. CODA is a meta-learning framework designed for learning parametric PDEs. It leverages common knowledge from multiple environments where trajectories from a same environment $e$ share the same PDE parameter values. CODA training performs adaptation in the parameter space by learning shared parameters across all environments and a context vector $c^e$ specific to each environment. At inference, CODA adapts to a new environment by tuning $c^e$ with several gradient steps. CAPE was not designed to perform adaptation via extra-trajectories, but instead needs the correct parameter values as input to condition a neural solver. We adapt it to our setting, by learning a context $c^e$ instead of using the real parameter values. During adaptation, we only tune this context $c^e$ via gradient updates. `[CLS]` ViT is a specifically designed baseline based on a vision transformer (Peebles & Xie, 2023), integrating a `[CLS]` token that serves as a learned parameter for each environment. This token lets the model handle different dynamics, and during inference, we adapt the `[CLS]` vector via gradient

updates, following the same approach used in CODA and CAPE. `ViT-in-context` is a transformer with separate temporal and spatial attention (Ho et al., 2019), where we stack context examples and preceding target frames in the temporal axis to provide in-context examples. Note that all these baselines are deterministic.

**Metrics** We evaluate the performance using the Relative $L^2$ norm between the predicted rollout trajectory $\hat{\boldsymbol{u}}_*^{\text{trajectory}}$ and the ground truth $\boldsymbol{u}_*^{\text{trajectory}}$: $L_{\text{test}}^2 = \frac{1}{N_{\text{test}}} \sum_{j \in \text{test}} \frac{||\hat{u}_j^{\text{trajectory}} - u_j^{\text{trajectory}}||_2}{||u_j^{\text{trajectory}}||_2}$.

**Results** As evidenced in Table 5, `Zebra` demonstrates strong overall performance in the one-shot adaptation setting, often surpassing gradient-based adaptation methods. For the more challenging datasets, such as *Burgers*, *Wave-b*, and the 2D cases, `Zebra` consistently achieves lower relative L2 errors, highlighting its capacity to model complex dynamics effectively. Notably, `Zebra` excels in 2D environments, outperforming both CODA and `[CLS]` ViT and avoiding the divergence issues encountered by CAPE. While `Zebra` performs comparably to CODA on simpler datasets like *Advection* and *Combined*, its overall stability and versatility across a range of scenarios, particularly in 2D settings, highlight its competitiveness. Overall, `Zebra` stands out as a reliable and scalable solution for adaptation for solving parametric PDEs, demonstrating that in-context learning offers a robust alternative to existing gradient-based adaptation methods. The ablation study (see Appendix D.1) highlights an important advantage of `Zebra`'s generative capabilities. When trained deterministically to predict the conditional expectation of the next token, the model accumulates significant error during the autoregressive rollout. In contrast, `Zebra`, trained to model trajectory distributions, can sample from this distribution at inference time, resulting in predictions that are more robust to error accumulation.

## 4.3. Out-of-distribution generalization

**Datasets** We evaluate our models on new PDE instances under the following distribution shifts: (i) *Heat*: We modify the forcing term parameterization. *Wide Forcing*: The forcing coefficients are varied from Appendix B.3, with

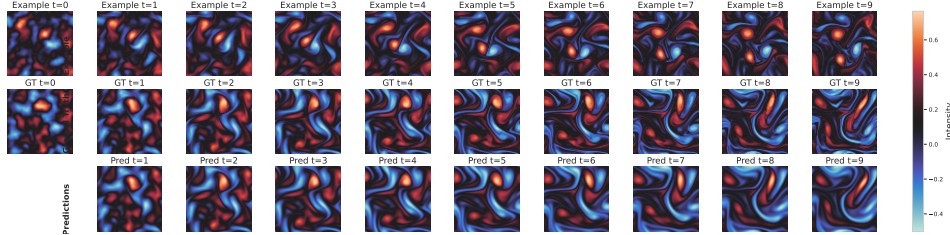

*Figure 2.* **One-shot** prediction on *Vorticity* in the turbulent OoD regime $\nu \in [1e-5, 1e-4]$. Top-row is the example given in context, mid-row is the ground truth trajectory, and the bottom-row is the generation with `Zebra`.

parameters sampled as $A_j \in [-1.0, 1.0]$, $\omega_j \in [-0.8, 0.8]$ compared to the training ranges $[-0.5, 0.5]$ and $[-0.4, 0.4]$. *Gaussian:* $\delta(t, x) = \sum_{j=1}^{5} A_j \exp\left(-\frac{(x-\mu_j)^2}{2\sigma_j^2}\right) \sin(\omega_j t)$, introducing smooth, localized excitations. This results in solutions similar to the homogeneous case away from peaks. *Square:* $\delta(t, x) = \sum_{j=1}^{5} A_j \sin\left(\omega_j t + 2\pi \frac{l_j x}{L} + \phi_j\right)$, which preserves the parameterization but introduces discontinuities and high-frequency content, resulting in a more challenging OoD scenario. (ii) *Vorticity*: In the *close* setting, viscosity is sampled from $[5 \times 10^{-4}, 10^{-3}]$, while the training range is $[10^{-3}, 10^{-2}]$. The *far* setting corresponds to a turbulent regime with viscosity in $[10^{-5}, 10^{-4}]$; see Figure 2 for a qualitative example. (iii) *Wave 2D*: We increase both the wave celerity and damping beyond the training distribution. The wave celerity $c$ is sampled from $[500, 550]$ vs. $[100, 500]$ during training, and the damping term $k$ from $[50, 60]$ vs. $[0, 50]$ in training.

**Setting** We evaluate all the models in a one-shot setting on trajectories with out-of-distribution PDE parameters on new initial conditions, making this a particularly challenging test of generalization.

**Results** We report the scores in Table 2. Overall, all methods experience performance degradation due to the distribution shift, with `Zebra` achieving the best results in three out of four experiments, while CODA and CAPE perform the worst. This poor performance for CAPE and CODA is expected on the 2D datasets, as they already struggled to generalize within the training distribution. However, for the *Heat* equation, errors for CAPE and CODA double, whereas `Zebra` maintains similar accuracy, demonstrating greater robustness to distribution shifts. Comparing `Zebra` and `ViT-in-context` to CAPE and CODA, it is remarkable that adaptation through in-context learning appears to be a more effective alternative than gradient-based adaptation for out-of-distribution generalization.

Out-of-distribution generalization remains a challenging task, particularly under strong shifts. On the *Vorticity* dataset, `Zebra` adapts to large shifts in viscosity and predicts the large-scale component of the dynamics. As shown

in Figure 2, the predictions are not as sharp as the ground truths, as the VQVAE was not explicitly trained to capture the part of the spectrum present in turbulent trajectories.

*Table 2.* Out-of-distribution results. Test results in relative L2 on the trajectory. '–' indicates inference has diverged. `n/a` indicates that the model was not evaluated in this setting.

|  | Wide | Heat Gaussian | Square | Wave 2D | Vorticity 2D close | far |
|---|---|---|---|---|---|---|
| CAPE | 0.47 | n/a | n/a | – | – | – |
| CODA | 1.03 | n/a | n/a | 1.51 | 1.71 | – |
| ViT-in-context | 0.52 | 0.40 | 0.66 | **0.68** | 0.30 | 0.37 |
| Zebra | **0.15** | **0.32** | **0.36** | **0.68** | **0.24** | **0.32** |

### 4.4. Generative ability of the model

The evaluation in Section 4.2 already shows that as a generative model, `Zebra` is less prone to error accumulation that deterministic auto-regressive models. We illustrate here additional benefits from the generative capabilities of `Zebra` through two example tasks: *uncertainty quantification* and *new trajectory generation*. Further analysis of the behavior of `Zebra` is provided in Appendix D.2 and Appendix D.3.

**Uncertainty quantification** Given a context example and an initial condition, `Zebra` can generate multiple trajectories thanks to the sampling operation at the classifier level. Statistics can then be derived from this sample of the trajectory distribution in order to assess for example the uncertainty associated to a prediction. An illustration is provided in Figure 3, the red curve represents the ground truth, the blue curve is the predicted mean and the blue shading indicates the empirical confidence interval ($3 \times$ standard deviation). Mean and standard deviation are calculated pointwise. We provide a more detailed analysis in Appendix D.2. In particular, it shows as expected that (i) uncertainty can be calibrated via the temperature parameter $\tau$ (Figure 17), and (ii) it decreases with additional context (Table 9).

**New trajectory generation** As a second illustration, we assess `Zebra`'s ability to generate relevant new trajectories conditioned on in-context examples alone, i.e. without prompting with an initial state query. This is similar to conditional image or text generation in LLMs. The gen-

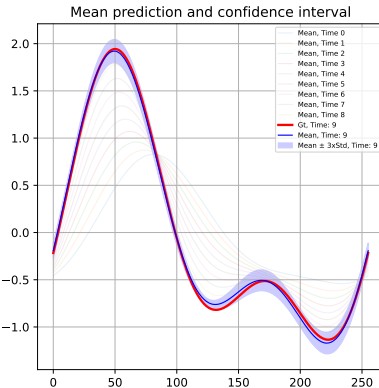

**Figure 3.** Uncertainty quantification with `Zebra` in a one-shot setting on *Heat* equation.

erated trajectories are sampled conditioned on a context trajectory from a new unseen environment. The key finding here is that `Zebra` effectively generates faithful trajectory distributions that closely match the real simulated trajectory distribution. Hence, given some examples from an unknown environment, `Zebra` could be used to generate trajectories that comply with the distribution in this environment. Qualitatively, Figure 4 illustrates how the real (from a held out sample) and the generated distributions match at two different time steps ($t = 0, t = 9$): the PCA projection, indicates a strong alignment. Quantitatively, Table 3 shows that the Wasserstein distance of the generated trajectories is comparable to the Wasserstein distance between validation and test samples. As a calibration measure, we also provide in Table 3 the Wasserstein distance between the real distribution and a Gaussian distribution. Further details are provided in Appendix D.3.

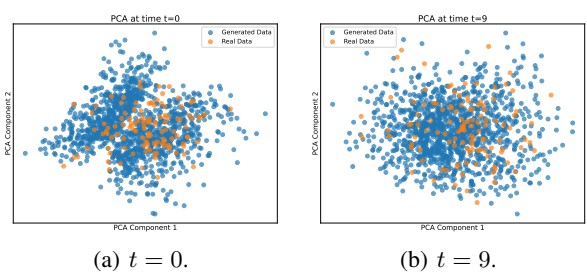

(a) $t = 0$.          (b) $t = 9$.

**Figure 4.** PCA Visualization of generated (blue) vs. real (orange) trajectories on *Combined Equation*

### 4.5. Accelerating inference

Since `Zebra` requires no gradient steps at inference, it is already faster than gradient-based adaptation (see Table 4). However, its autoregressive nature introduces significant overhead at inference: generating trajectories token by token increases solver calls by one to two orders of magnitude

**Table 3.** Comparison of distributions using the Wasserstein distance between generated trajectories and real trajectories.

| Distance Metric | Advection | Combined Equation |
|---|---|---|
| Gaussian noise vs. real data | 18.22 | 16.15 |
| Validation data vs. test data | 5.11 | 1.87 |
| `Zebra`-generated data vs. real data | 5.57 | 2.21 |

compared to direct surrogate modeling, making inference costly. To address this, we propose a fast inference method that accelerates inference by orders of magnitude relative to both the original model and gradient-based adaptation (Table 4). Instead of token-wise autoregressive generation, we predict entire frames at once. This is achieved by replacing the token-wise autoregressive generation process with a frame-wise autoregressive surrogate, implemented as a UNet. The UNet, conditioned on a context embedding output by the transformer, takes frame $u^t$ as input and predicts $\hat{u}^{t+\Delta t}$. A crucial component is the context embedding, which captures underlying dynamics from example trajectories. This is learned by introducing a new token, `[DYN]`, at the transformer's input, analogous to `[CLS]` in BERT, allowing attention to encode context dynamics effectively. The implementation is detailed in Appendix C and in Figure 7. Table 4 shows that this reduces inference time by one to two orders of magnitude, making `Zebra` highly efficient.

**Table 4.** Inference times for one-shot adaptation. Average time in seconds to predict a single trajectory given a context trajectory and an initial condition. Times include adaptation and forecast for CODA and CAPE, while for `Zebra`, they include encoding, autoregressive prediction, and decoding.

| | Advection | Vorticity 2D |
|---|---|---|
| CAPE | 18s | 23s |
| CODA | 31s | 28s |
| Zebra | 3s | 21s |
| Zebra + UNet | **0.10s** | **0.14s** |

As shown in Table 5, this framework matches or outperforms pretrained `Zebra` in most cases. The dynamics embedding captures meaningful context, enabling efficient UNet training. In contrast, methods like CAPE and CODA must learn both model weights and environment embeddings simultaneously, making training less efficient.

**Table 5.** `Zebra` vs `Zebra` + UNet, for the in-distribution one-shot setting. Test results in relative L2 error.

| | Advection | Combined | Wave 2D | Vorticity 2D |
|---|---|---|---|---|
| Zebra | 0.00794 | **0.00965** | 0.207 | 0.119 |
| Zebra+UNet | **0.0072** | 0.0138 | **0.150** | **0.0869** |

## 5. Limitations

**Reconstruction quality** The fidelity of the generated trajectories is constrained by the decoder's ability to reconstruct fine details from the quantized latent space. While the reconstruction quality is sufficient for the applications considered in this study, it may be inadequate for systems characterized by high-frequency physical phenomena. To address this limitation, scaling the codebook size (Yu et al., 2023a; Mentzer et al., 2023) or exploring alternatives to vector quantization (Li et al., 2024) could improve reconstruction accuracy—provided that the model's in-context learning capabilities are preserved.

**Irregular grids** Our current encoder and decoder rely on convolutional blocks, which limits the architecture to data defined on regular grids. Adopting more flexible encoding and decoding schemes—such as those proposed in Serrano et al. (2024)—could enable support for irregularly sampled inputs or more complex geometric domains.

**Data requirements** Our data scaling analysis indicates that the framework requires a large number of training trajectories to generalize effectively. As such, the current method is not well suited for data-scarce regimes and depends on significant diversity in the parameter space. A more thorough investigation is needed to determine the minimal level of parameter variability required to ensure generalization in or close to the training distribution of PDE parameters.

**Scalability and complexity** The transformer in our architecture applies causal attention across the full sequence—comprising the number of context examples $N$, time steps $T$, and spatial tokens per frame $h \times w$. Extending to 3D data with a third spatial dimension $d$ increases the complexity to $\mathcal{O}(NThwd)^2$. This quadratic growth can lead to prohibitively long sequences in higher-dimensional settings, and may necessitate architectural changes such as axial or factorized attention mechanisms.

## 6. Conclusion

This study introduces `Zebra`, a novel generative model that adapts language model pretraining techniques for solving parametric PDEs. We propose a pretraining strategy that enables `Zebra` to develop in-context learning capabilities. Our experiments demonstrate that the pretrained model performs competitively against specialized baselines across various scenarios. Additionally, as a generative model, `Zebra` facilitates uncertainty quantification and can generate new trajectories, providing valuable flexibility in applications.

## Acknowledgements

We acknowledge the financial support provided by DL4CLIM (ANR-19-CHIA-0018-01), DEEP- NUM (ANR-21-CE23-0017-02), PHLUSIM (ANR-23-CE23-0025-02), and PEPR Sharp (ANR- 23-PEIA-0008, ANR, FRANCE 2030). We also thank Emmanuel de Bézenac, Jean-Yves Franceschi, and Étienne Le Naour for insightful and stimulating discussions.

## Impact Statement

This paper presents work whose goal is to advance the field of Machine Learning. There are many potential societal consequences of our work, none which we feel must be specifically highlighted here.

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

# A. Related Work

## A.1. Learning parametric PDEs

**The classical ML paradigm**    The classical ML paradigm for solving parametric PDEs consists in sampling from the PDE parameter distribution trajectories to generalize to new PDE parameter values. It is the classical ERM approach. The natural way for generalizing to new PDE parameters is to explicitly embed them in the neural network (Brandstetter et al., 2022b). (Takamoto et al., 2023) proposed a channel-attention mechanism to guide neural solvers with the physical coefficients given as input; it requires complete knowledge of the physical system and are not designed for other PDE parameter values, e.g., boundary conditions. It is commonly assumed that prior knowledge are not available, but instead rely on past states of trajectories for inferring the dynamics. Neural solvers and operators learn parametric PDEs by stacking the past states as channel information as done in (Li et al., 2021), or by creating additional temporal dimension as done in video prediction contexts (Ho et al., 2022; McCabe et al., 2023). Their performance drops when shifts occur in the data distribution, which is often met with parametric PDEs, as small changes in the PDE parameters can lead to various dynamics. To better generalize to new PDE parameter values, (Subramanian et al., 2023) instead leverages fine-tuning from pretrained models to generalize to new PDE parameters. It however often necessitates a relatively large number of fine tuning samples to effectively adapt to new PDE parameter values, by updating all or a subset of parameters (Herde et al., 2024; Hao et al., 2024).

**Gradient-based adaptation**    To better adapt to new PDE parameters values at inference, several works have explored learning on multiple environments. During training, a limited number of environments are available, each corresponding to a specific PDE instance. (Yin et al., 2022) introduced LEADS, a multi-task framework for learning parametric PDEs, where a shared model from all environments and a model specific to each environment are learned jointly. At inference, for a new PDE instance, the shared model remain frozen and only a model specific to that environment is learned. (Kirchmeyer et al., 2022) proposed to perform adaptive conditioning in the parameter space; the framework adapts the weights of a model to each environment via a hyper-network conditioned by a context vector $c^e$ specific to each environment. At inference, the model adapts to a new environment by only tuning $c^e$. (Park et al., 2023) bridged the gap from the classical gradient-based meta-learning approaches by addressing the limitations of second-order optimization of MAML and its variants (Finn et al., 2017; Zintgraf et al., 2019b). Other works have also extended these frameworks to quantify uncertainty of the predictions : (Jiaqi et al., 2024) proposed a conditional neural process to capture uncertainty in the context of multiple environments with sparse trajectories, while (Nzoyem et al., 2024) leveraged information from multiple environments to enable more robust predictions and uncertainty quantification.

**In-context learning for PDE**    Inspired by the in-context learning (ICL) paradigm in large language models (LLMs), recent works have explored adapting this approach for solving PDEs and modeling dynamical systems. One of the earliest efforts in this direction is Yang et al. (2023), which aims to learn operators capable of adapting to different physical scenarios by leveraging in-context examples. Their approach utilizes an encoder-decoder transformer, where the transformer encodes the context prompt. This prompt, together with a query, is then passed to the decoder, which predicts the corresponding output values of the state vector. However, since functions are represented as scattered point tokens, the model encounters computational complexity challenges and is primarily limited to 1D ODEs or sparse 2D data. VICON (Cao et al., 2024) extends this framework by leveraging vision transformers (ViTs) that operate on image patches. However, the two approaches further differ in their problem settings: while our method focuses on adapting a surrogate model using a small number of context trajectories and then unrolling from an initial condition,as traditional PDE solvers do, VICON is designed for future-state forecasting based on access to past trajectory history. Our ViT In-Context baseline already captures an architecture that is structurally similar to VICON, but tailored to our adaptation formulation, thereby offering an implicit point of comparison.

Chen et al. (2024) takes a different approach, focusing on unsupervised pretraining for operator learning. Their method involves pretraining the encoder-decoder of neural operators on proxy tasks (such as masked prediction or super-resolution) that require only snapshots of the dynamics rather than full simulation data, followed by fine-tuning on target dynamics. In this case, ICL is used only at inference time, where context examples similar to a query input are retrieved from the training set, and their solutions are aggregated and averaged to form the final prediction. This setting differs significantly from ours. Notably, all these approaches rely on deterministic ViT-like architectures, whereas our method employs a generative stochastic model.

Note that in-context learning is still a not well understood phenomenon and that different hypotheses are being explored which attempt to fill this gap (Dong et al., 2024). Two prevalent explanations come from a Bayesian perspective on ICL as

introduced in a popular paper (Xie et al., 2022) and the gradient descent view as introduced e.g. in Dai et al. (2023) that identied a dual form between transformer attention and gradient descent highlighting relations between GPT-based ICL and explicit fine tuning.

### A.2. Generative models

**Auto-regressive Transformers for Images and Videos** Recent works have explored combining language modeling techniques with image and video generation, typically using a VQ-VAE (Oord et al., 2017) paired with a causal transformer (Esser et al., 2021) or a bidirectional transformer (Chang et al., 2022). VQGAN (Esser et al., 2021) has become the leading framework by incorporating perceptual and adversarial losses to improve the visual realism of decoder outputs from quantized latent representations. However, while these methods succeed in generating visually plausible images, they introduce a bias—driven by perceptual and adversarial losses—that leads the network to prioritize perceptual similarity and realism, often causing reconstructions to deviate from the true input. In contrast, `Zebra` focuses on maximizing reconstruction accuracy, and we did not observe benefits from using adversarial or perceptual losses during training.

In video generation, models like Magvit (Yu et al., 2023a) and Magvit2 (Yu et al., 2023b) adopt similar strategies, using 3D CNN encoders to compress sequences of video frames into spatiotemporal latent representations by exploiting the structural similarities between successive frames in a video. However, such temporal compression is unsuitable for modeling partial differential equations (PDEs), where temporal dynamics can vary significantly between frames depending on the temporal resolution. With `Zebra`, we spatially compress observations using the encoder and learn the temporal dynamics with an auto-regressive transformer, avoiding temporal compression.

**Generative PDE surrogate models** Recent advances have leveraged diffusion-based generative models for surrogate modeling of PDEs. Huang et al. (2024) introduce DiffusionPDE, which employs a diffusion model to complete partial observations and solve PDEs under limited supervision. Hu et al. (2024) propose the Wavelet Diffusion Neural Operator (WDNO), which performs diffusion in the wavelet domain and uses multi-resolution training to capture sharp features and generalize across spatial scales, with a focus on long-term simulation and control. Lino et al. (2025) develop a diffusion graph network that learns full solution distributions of complex fluids; it enables direct sampling of equilibrium states on unstructured meshes, facilitating efficient estimation of turbulent statistics. Finally, Hao et al. (2024) present DPOT, an autoregressive transformer pre-trained with noise-perturbed inputs on a large collection of PDE simulations.

## B. Dataset details

*Table 6.* **Dataset Summary**

| Dataset Name | Number of env. | Trajectories per env. | Main parameters |
|---|---|---|---|
| *Advection* | 1200 | 10 | Advection speed |
| *Heat* | 1200 | 10 | Diffusion and forcing |
| *Burgers* | 1200 | 10 | Diffusion and forcing |
| *Wave boundary* | 4 | 3000 | Boundary conditions |
| *Combined equation* | 1200 | 10 | $\alpha, \beta, \gamma$ |
| *Wave 2D* | 1200 | 10 | Wave celerity and damping |
| *Vorticity 2D* | 1200 | 10 | Diffusion |

### B.1. Advection

We consider a 1D advection equation with advection speed parameter $\beta$:

$$\partial_t u + \beta \partial_x u = 0$$

For each environment, we sample $\beta$ with a uniform distribution in $[0, 4]$. We sample 1200 parameters, and 10 trajectories per parameter, constituting a training set of 12000 trajectories. At test time, we draw 12 new parameters and evaluate the performance on 10 trajectories each.

We fix the size of the domain $L = 128$ and draw initial conditions as described in Equation (3) in appendix B.5 and generate

solutions with the method of lines and the pseudo-spectral solver described in (Brandstetter et al., 2022b). We take 140 snapshots along a 100s long simulations, which we downsample to 14 timestamps for training. We used a spatial resolution of 256.

### B.2. Burgers

We consider the Burgers equation as a special case of the combined equation described in Appendix B.5 and initially in (Brandstetter et al., 2022b), with fixed $\gamma = 0$ and $\alpha = 0.5$. However, in this setting, we include a forcing term $\delta(t, x) = \sum_{j=1}^{J} A_j \sin(\omega_j t + 2\pi \ell_j x/L + \phi_j)$ that can vary across different environments. We fix $J = 5$, $L = 16$. We draw initial conditions as described in Equation (3).

For each environment, we sample $\beta$ with a log-uniform distribution in $[1e - 3, 5]$, and sample the forcing term coefficients uniformly: $A_j \in [-0.5, 0.5]$, $\omega_j \in [-0.4, -0.4]$, $\ell_j \in \{1, 2, 3\}$, $\phi_j \in [0, 2\pi]$. We create a dataset of 1200 environments with 10 trajectories for training, and 12 environments with 10 trajectories for testing.

We use the solver from (Brandstetter et al., 2022b), and take 250 snapshots along the 4s of the generations. We employ a spatial resolution of 256 and downsample the temporal resolution to 25 frames.

### B.3. Heat

We consider the heat equation as a special case of the combined equation described in Appendix B.5 and initially in (Brandstetter et al., 2022b), with fixed $\gamma = 0$ and $\alpha = 0$. However, in this setting, we include a forcing term $\delta(t, x) = \sum_{j=1}^{J} A_j \sin(\omega_j t + 2\pi \ell_j x/L + \phi_j)$ that can vary across different environments. We fix $J = 5$, $L = 16$. We draw initial conditions as described in Equation (3).

For each environment, we sample $\beta$ with a log-uniform distribution in $[1e - 3, 5]$, and sample the forcing term coefficients uniformly: $A_j \in [-0.5, 0.5]$, $\omega_j \in [-0.4, -0.4]$, $\ell_j \in \{1, 2, 3\}$, $\phi_j \in [0, 2\pi]$. We create a dataset of 1200 environments with 10 trajectories for training, and 12 environments with 10 trajectories for testing.

We use the solver from (Brandstetter et al., 2022b), and take 250 snapshots along the 4s of the generations. We employ a spatial resolution of 256 and downsample the temporal resolution to 25 frames.

### B.4. Wave boundary

We consider a 1D wave equation as in (Brandstetter et al., 2022b).

$$\partial_{tt} u - c^2 \partial_{xx} u = 0, \quad x \in [-8, 8]$$

where $c$ is the wave velocity ($c = 2$ in our experiments). We consider Dirichlet $\mathcal{B}[u] = u = 0$ and Neumann $\mathcal{B}[u] = \partial_x u = 0$ boundary conditions.

We consider 4 different environments as each boundary can either respect Neumann or Dirichlet conditions, and sample 3000 trajectories for each environment. This results in 12000 trajectories for training. For the test set, we sample 30 new trajectories from these 4 environments resulting in 120 test trajectories.

The initial condition is a Gaussian pulse with a peak at a random location. Numerical ground truth is generated with the solver proposed in (Brandstetter et al., 2022b). We obtain ground truth trajectories with resolution $(n_x, n_t) = (256, 250)$, and downsample the temporal resolution to obtain trajectories of shape $(256, 60)$.

### B.5. Combined equation

We used the setting introduced in (Brandstetter et al., 2022b), but with the exception that we do not include a forcing term. The combined equation is thus described by the following PDE:

$$[\partial_t u + \partial_x(\alpha u^2 - \beta \partial_x u + \gamma \partial_{xx} u)](t, x) = \delta(t, x), \tag{2}$$

$$\delta(t, x) = 0, \quad u_0(x) = \sum_{j=1}^{J} A_j \sin(2\pi \ell_j x/L + \phi_j). \tag{3}$$

For training, we sampled 1200 triplets of parameters uniformly within the ranges $\alpha \in [0, 1]$, $\beta \in [0, 0.4]$, and $\gamma \in [0, 1]$. For each parameter instance, we sample 10 trajectories, resulting in 12000 trajectories for training and 120 trajectories for testing. We used the solver proposed in (Brandstetter et al., 2022a) to generate the solutions. The trajectories were generated with a spatial resolution of 256 for 10 seconds, along which 140 snapshots are taken. We downsample the temporal resolution to obtain trajectories with shape (256, 14).

### B.6. Vorticity

We propose a 2D turbulence equation. We focus on analyzing the dynamics of the vorticity variable. The vorticity, denoted by $\omega$, is a vector field that characterizes the local rotation of fluid elements, defined as $\omega = \nabla \times \mathbf{u}$. The vorticity equation is expressed as:

$$\frac{\partial \omega}{\partial t} + (\mathbf{u} \cdot \nabla)\omega - \nu \nabla^2 \omega = 0 \tag{4}$$

Here, $\mathbf{u}$ represents the fluid velocity field, $\nu$ is the kinematic viscosity with $\nu = 1/Re$. For the vorticity equation, the parametric problem consists in learning dynamical systems with changes in the viscosity term.

For training, we sampled 1200 PDE parameter values in the range $\nu = [1e - 3, 1e - 2]$. For test, we evaluate our model on 120 new parameters not seen during training in the same paramter range. For each parameter instance, we sample 10 trajectory, resulting in 12000 trajectories for training and 1200 for test.

**Data generation** For the data generation, we use a 5 point stencil for the classical central difference scheme of the Laplacian operator. For the Jacobian, we use a second order accurate scheme proposed by Arakawa that preserves the energy, enstrophy and skew symmetry (Arakawa, 1966). Finally for solving the Poisson equation, we use a Fast Fourier Transform based solver. We discretize a periodic domain into $512 \times 512$ points for the DNS and uses a RK4 solver with $\Delta t = 1e - 3$ on a temporal horizon $[0, 2]$. We then perform a temporal and spatial down-sample operation, thus obtaining trajectories composed of 10 states on a $64 \times 64$ grid.

We consider the following initial conditions:

$$E(k) = \frac{4}{3}\sqrt{\pi} \left(\frac{k}{k_0}\right)^4 \frac{1}{k_0} \exp\left(-\left(\frac{k}{k_0}\right)^2\right) \tag{5}$$

Vorticity is linked to energy by the following equation :

$$\omega(k) = \sqrt{\frac{E(k)}{\pi k}} \tag{6}$$

### B.7. Wave 2D

We propose a 2D damped wave equation, defined by

$$\frac{\partial^2 \omega}{\partial t^2} - c^2 \Delta \omega + k \frac{\partial \omega}{\partial t} = 0 \tag{7}$$

where $c$ is the wave speed and $k$ is the damping coefficient. We are only interested in learning $\omega$. To tackle the parametric problem, we sample 1200 parameters in the range $c = [0, 50]$ and $k = [100, 500]$. For validation, we evaluate our model on 120 new parameters not seen during training in the same paramter range. For each parameter instance, we sample 10 trajectory, resulting in 12000 trajectories for training and 1200 for validation.

**Data generation** For the data generation, we consider a compact spatial domain $\Omega$ represented as a $64 \times 64$ grid and discretize the Laplacian operator similarly. $\Delta$ is implemented using a $5 \times 5$ discrete Laplace operator in simulation. For boundary conditions, null neumann boundary conditions are imposed. We set $\Delta t = 6.25e - 6$ and generate trajectories on the temporal horizon $[0, 5e - 3]$. The simulation was integrated using a fourth order runge-kutta schema from an initial condition corresponding to a sum of gaussians:

$$\omega_0(x, y) = C \sum_{i=1}^{p} \exp\left(-\frac{(x - x_i)^2 + (y - y_i)^2}{2\sigma_i^2}\right) \tag{8}$$

where we choose $p = 5$ gaussians with $\sigma_i \sim \mathcal{U}_{[0.025,0.1]}$, $x_i \sim \mathcal{U}_{[0,1]}$, $y_i \sim \mathcal{U}_{[0.,1]}$. We fixed $C$ to 1 here. Thus, all initial conditions correspond to a sum of gaussians of varying amplitudes.

## C. Architecture details

### C.1. Baseline implementations

For all baselines, we followed the recommendations given by the authors. We report here the architectures used for each baseline:

- CODA: For CODA, we implemented a U-Net (Ronneberger et al., 2015) and a FNO (Li et al., 2020) as the neural network decoder. For all the different experiments, we reported in the results the best score among the two backbones used. We trained the different models in the same manner as Zebra, i.e. via teacher forcing (Radford et al., 2018). The model is adapted to each environment using a context vector specific to each environment. For the size of the context vector, we followed the authors recommendation and chose a context size equals to the number of degrees of freedom used to define each environment for each dataset. At inference, we adapt to a new environment using 250 gradient steps.

- CAPE: For CAPE (Takamoto et al., 2023), we adapted the method to an adaptation setting. Instead of giving true physical coefficients as input, we learn to auto-decode a context vector $c^e$ as in CODA, which implicitly embeds the specific characteristics of each environment. During inference, we only adapt $c^e$ with 250 gradient steps. For the architectures, we use UNET and FNO as the backbones, and reported the best results among the two architectures for all settings.

- `[CLS]` ViT: For the ViT, we use a simple vision transformer architecture (Dosovitskiy et al., 2021), but adapt it to a meta-learning setting where the CLS token encodes the specific variations of each environment. At inference, the CLS token is adapted to a new environment with 100 gradient steps.

- `ViT-in-context`: We implement `ViT-in-context` using a standard transformer architecture with separate temporal and spatial attention mechanisms, following Ho et al. (2019). During both training and inference, context examples are stacked along the temporal dimension. The model is trained to predict the next frame in the target sequence, conditioned on both the context examples and the preceding frames of the target sequence.

### C.2. Zebra additional details

**Zebra** We describe the pretraining strategy in Section 3, and provide details on the architecture and its hyperparameters in Appendix C. The datasets used are described in Appendix B. We plan to release the code, the weights of the models, and the datasets used in this study upon acceptance.

For clarity, we outline the pretraining steps of `Zebra` in Figure 5 illustrated with the *vorticity 2D* dataset.

We also provide illustrations of our inference pipeline in Figure 6. We finally include a schematic view of the different generation possibilities with `Zebra` in Figure 8, using the sequence design adopted during pretraining.

**Zebra + UNet** `Zebra` is competitive both in-distribution and out-of-distribution, while also enabling uncertainty quantification due to its generative nature. Additionally, `Zebra` can generate novel trajectories and initial conditions, providing a way to sample complex initial states. As such Zebra is already faster than gradient-based adaptation methods, but since the model generates trajectory solutions token by token, the number of calls to the transformer increases by one or two orders of magnitude compared to direct surrogate modeling, making the process costly.

In this section, we propose a hybrid approach that leverages `Zebra`'s pretrained knowledge in combination with a conventional neural surrogate model. The objective is to develop a framework that encodes context trajectories into an embedding vector, which then conditions a neural surrogate, similar to classical adaptation methods such as CODA and CAPE.

To achieve this, we finetune `Zebra` as an encoder to adapt a conditional surrogate model, such as a UNet (Ronneberger et al., 2015). We introduce a `[DYN]` token (short for dynamics), which is appended to the right of the context sequence

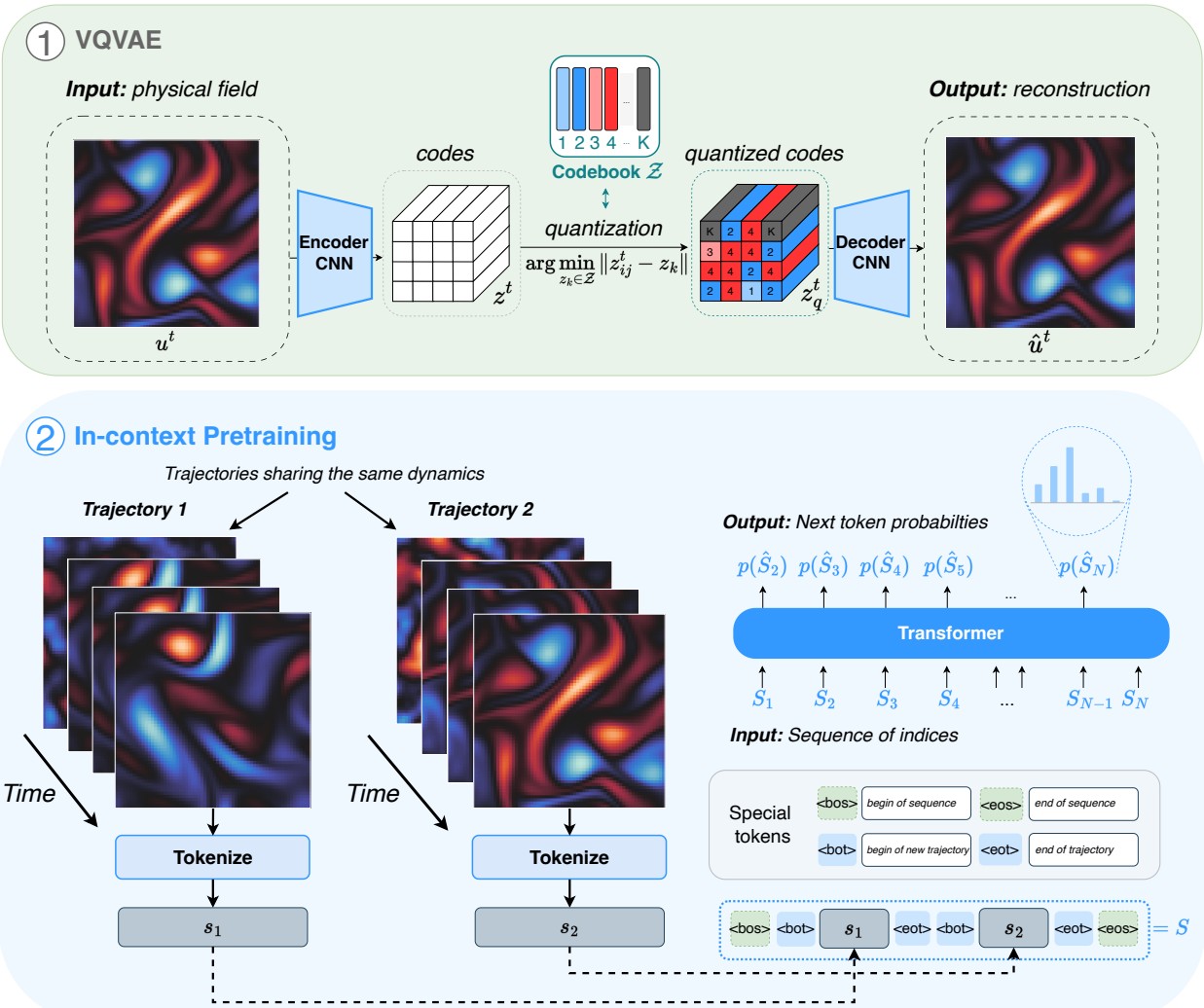

*Figure 5.* Zebra's pretraining includes two stages. 1) A finite vocabulary of physical phenomena is learned by training a VQ-VAE on spatial representations. 2) During the pretraining, multiple trajectories sharing the same dynamics are tokenized and concatenated into a common sequence $S$. The transformer is trained to predict the next-token by minimizing the cross-entropy loss.

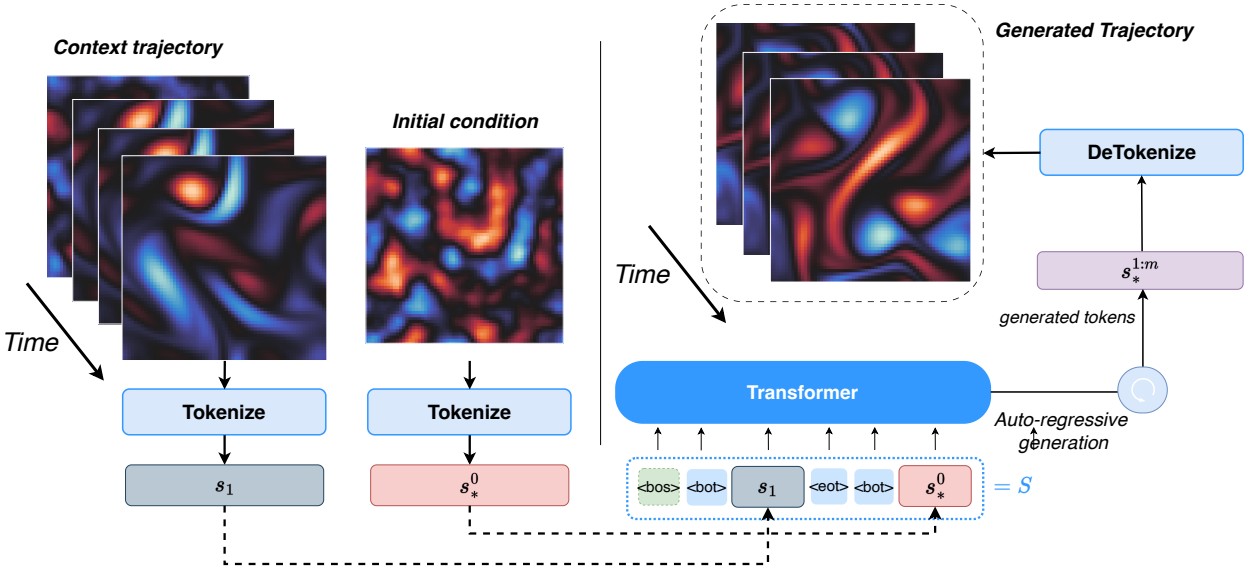

*Figure 6.* Zebra's inference pipeline from context trajectory. The context trajectory and initial conditions are tokenized into index sequences that are concatenated according to the sequence design adopted during pretraining. The transformer then generates the next tokens to complete the sequence. We detokenize these indices to get back to the physical space.

during both training and inference. This allows Zebra to extract a dynamics embedding from the transformer's output, defined as:

$$\xi_S = \text{Transformer}(S, \texttt{[DYN]})$$

where $S$ represents a sequence of tokens encoding the context trajectories. The embedding $\xi_S$ captures key properties of the dynamics and is mapped to the conditioning space of a UNet, which adapts the model to each specific dynamics. Following Gupta & Brandstetter (2022), the UNet conditioning modifies the network's biases. Once the dynamics embedding is extracted, the UNet can directly predict the next state from the current state :

$$\hat{\boldsymbol{u}}^{t+\Delta t} = \text{UNet}(\boldsymbol{u}^t, \xi_S).$$

With this architecture, we effectively extract the key dynamics from example trajectories and use this information to adapt a neural surrogate, significantly accelerating inference. The complete pipeline at inference is illustrated in Figure 7.

To efficiently finetune Zebra, we apply LoRA (Hu et al., 2021), keeping the transformer's weights frozen while learning low-rank updates. This setup enables the UNet to leverage Zebra's pre-learned representations while achieving a substantial speedup. Compared to standalone Zebra, integrating a UNet improves inference speed by a factor of $\times 30$ in 1D and $\times 150$ in 2D. The method is also considerably faster than CODA and CAPE while maintaining competitive performance across tasks. The inference times are summarized in Table 4.

Finally we illustrate the inference pipeline for accelerating the inference of Zebra with the UNet in Figure 7.

### C.3. Auto-regressive transformer

Zebra's transformer is based on Llama's architecture, which we describe informally in Figure 9. We use the implementation provided by HuggingFace (Wolf, 2019) and the hyperparameters from Table 7 in our experiments. For training the transformer, we used a single NVIDIA TITAN RTX for the 1D experiments and used a single A100 for training the model on the 2D datasets. Training the transformer on 2D datasets took 20h on a single A100 and it took 15h on a single RTX for the 1d dataset.

### C.4. VQVAE

The quantizer used at the token level is a VQVAE model (Oord et al., 2017). As illustrated in Figure 10 this is an encoder-decoder architecture with an intermediate quantizer component.

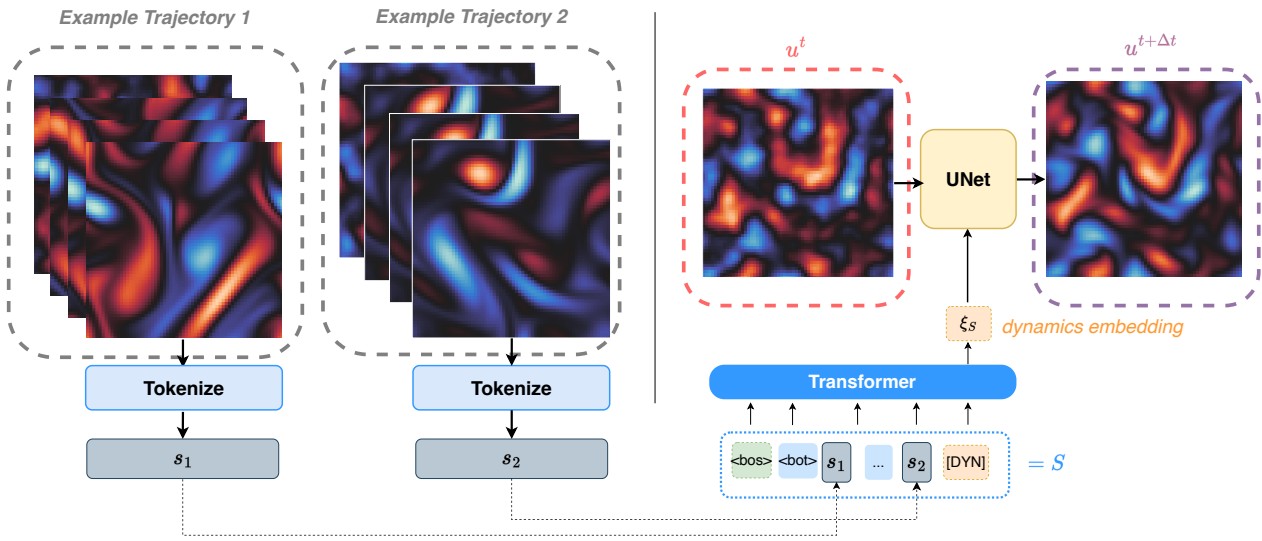

*Figure 7.* `Zebra` + UNet inference pipeline. `Zebra` serves as an encoder, utilizing the special token `[DYN]` to generate a dynamics embedding $\xi$ from the context example trajectories. Once this embedding is obtained, the UNet can autoregressively forecast the sequence significantly faster than a next-token transformer.

*Table 7.* Hyperparameters for Zebra's Transformer

| Hyperparameters | Advection | Heat | Burgers | Wave b | Combined | Vorticity 2D | Wave 2D |
|---|---|---|---|---|---|---|---|
| max_context_size | 2048 | 2048 | 2048 | 2048 | 2048 | 8192 | 8192 |
| batch_size | 4 | 4 | 4 | 4 | 4 | 2 | 2 |
| num_gradient_accumulations | 1 | 1 | 1 | 1 | 1 | 4 | 4 |
| hidden_size | 256 | 256 | 256 | 256 | 256 | 384 | 512 |
| mlp_ratio | 4.0 | 4.0 | 4.0 | 4.0 | 4.0 | 4.0 | 4.0 |
| depth | 8 | 8 | 8 | 8 | 8 | 8 | 8 |
| num_heads | 8 | 8 | 8 | 8 | 8 | 8 | 8 |
| vocabulary_size | 264 | 264 | 264 | 264 | 264 | 2056 | 2056 |
| start learning_rate | 1e-4 | 1e-4 | 1e-4 | 1e-4 | 1e-4 | 1e-4 | 1e-4 |
| weight_decay | 1e-4 | 1e-4 | 1e-4 | 1e-4 | 1e-4 | 1e-4 | 1e-4 |
| scheduler | Cosine | Cosine | Cosine | Cosine | Cosine | Cosine | Cosine |
| num_epochs | 100 | 100 | 100 | 100 | 100 | 30 | 30 |

## a) Conditional generation from similar trajectories

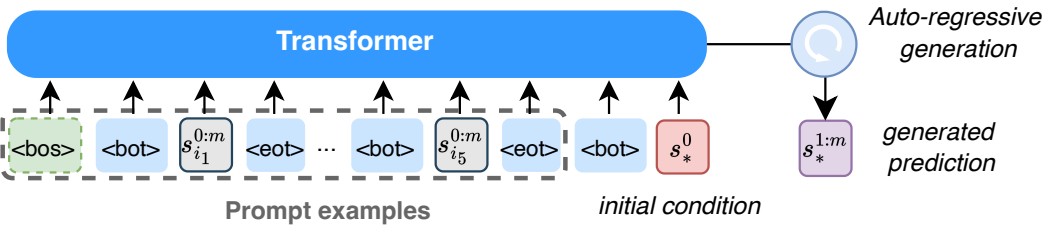

## b) Unconditional generation from similar trajectories

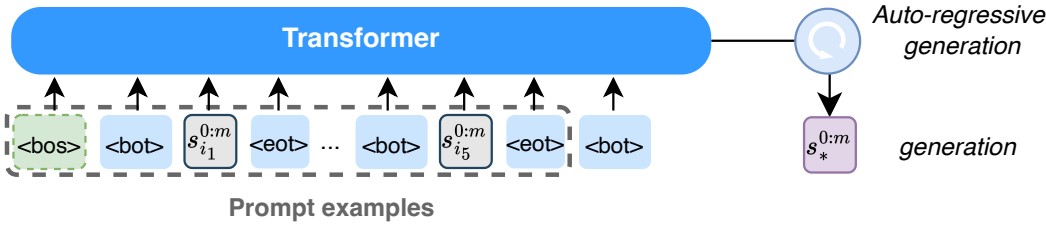

## c) Unconditional generation

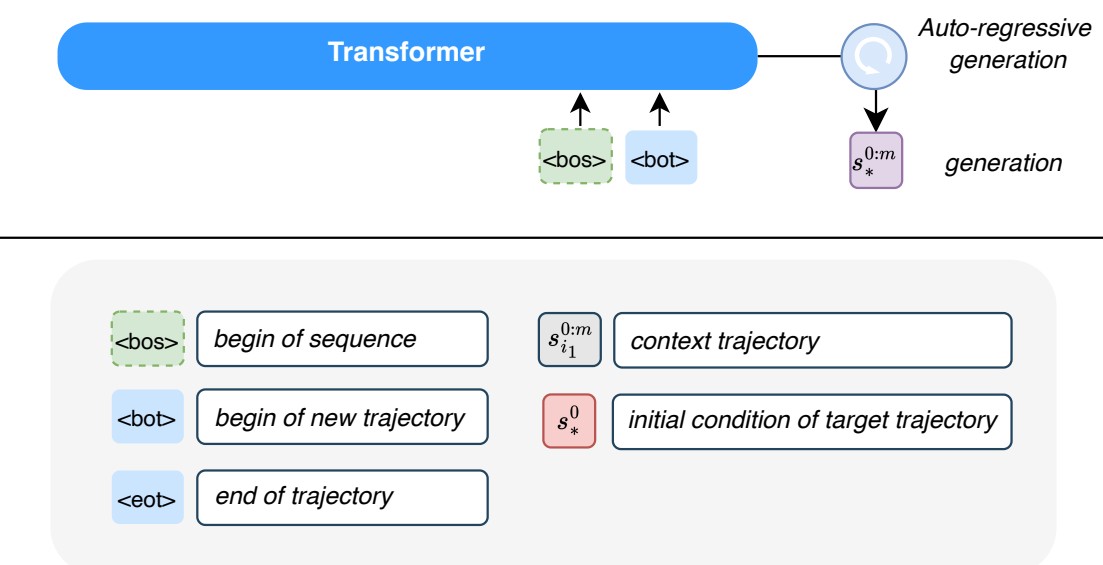

*Figure 8.* Generation possibilities with `Zebra`.

*Figure 9.* `Zebra`'s transformer architecture is based on Llama (Touvron et al., 2023).

The encoder spatially compresses the input function $\boldsymbol{u}^t$ by reducing its spatial resolution $H \times W$ to a lower resolution $h \times w$ while increasing the channel dimension to $d$. This is achieved through a convolutional model $\mathcal{E}_w$, which maps the input to a continuous latent variable $\mathbf{z}^t = \mathcal{E}_w(\boldsymbol{u}^t)$, where $\mathbf{z}^t \in \mathbb{R}^{h \times w \times d}$. The latent variables are then quantized to discrete codes $\mathbf{z}_q^t$ using a codebook $\mathcal{Z}$ of size $K = |\mathcal{Z}|$ and through the quantization step $q$. For each spatial code $\mathbf{z}_{[ij]}^t$, the nearest codebook entry $z_k$ is selected:

$$\mathbf{z}_{q,[ij]}^t = q(\mathbf{z}_{[ij]}^t) := \arg\min_{z_k \in \mathcal{Z}} \|\mathbf{z}_{[ij]}^t - z_k\|.$$

The decoder $\mathcal{D}_\psi$ reconstructs the signal $\hat{\boldsymbol{u}}^t$ from the quantized latent codes $\hat{\mathbf{z}}_q^t$. Both models are jointly trained to minimize the reconstruction error between the function $\boldsymbol{u}^t$ and its reconstruction $\hat{\boldsymbol{u}}^t = \mathcal{D}_\psi \circ q \circ \mathcal{E}_w(\boldsymbol{u}^t)$. The codebook $\mathcal{Z}$ is updated using an exponential moving average (EMA) strategy, which stabilizes training and ensures high codebook occupancy.

The training objective is:

$$\mathcal{L}_{\text{VQ}} = \frac{\|\boldsymbol{u}^t - \hat{\boldsymbol{u}}^t\|_2}{\|\boldsymbol{u}^t\|_2} + \alpha \|\text{sg}[\mathbf{z}_q^t] - \mathcal{E}_w(\boldsymbol{u}^t)\|_2^2,$$

where the first term is the Relative L2 loss commonly used in PDE modeling, and the second term is the commitment loss, ensuring encoder outputs are close to the codebook entries. The parameter $\alpha$, set to 0.25, balances the two components. Here, sg denotes the stop-gradient operation that detaches a tensor from the computational graph.

We provide a schematic view of the VQVAE framework in Figure 10 and detail the architectures used for the encoder and decoder on the 1D and 2D datasets respectively in Figure 11 and Figure 12. As detailed, we use residual blocks to process latent representations, and downsampling and upsampling block for decreasing / increasing the spatial resolutions. We provide the full details of the hyperparameters used during the experiments in Table 8. For training the VQVAE, we used a single NVIDIA TITAN RTX for the 1D experiments and used a single V100 for training the model on the 2D datasets. Training the encoder-decoder on 2D datasets took 20h on a single V100 and it took 4h on a single RTX for 1D dataset.

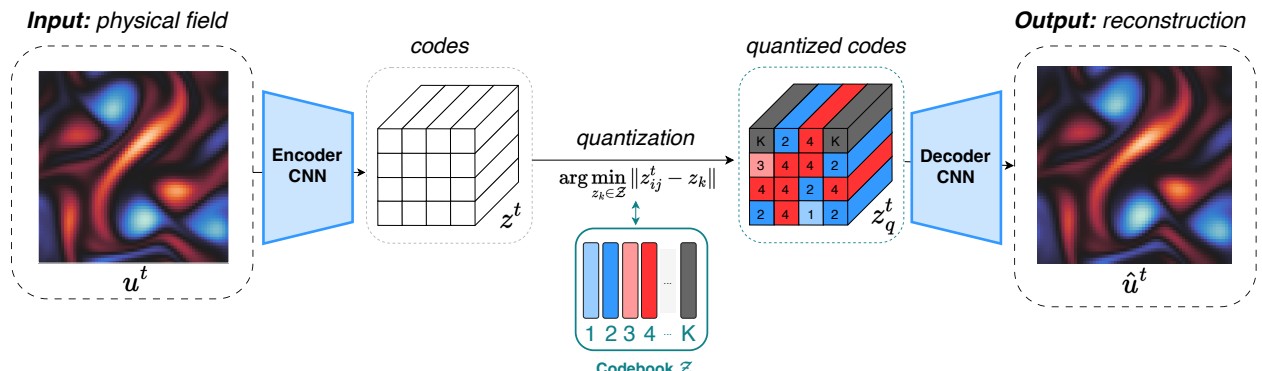

*Figure 10.* Zebra's VQVAE is used to obtain compressed and discretized latent representation. By retrieving the codebok index for each discrete representation, we can obtain discrete tokens encoding physical observations that can be mapped back to the physical space with high fidelity.

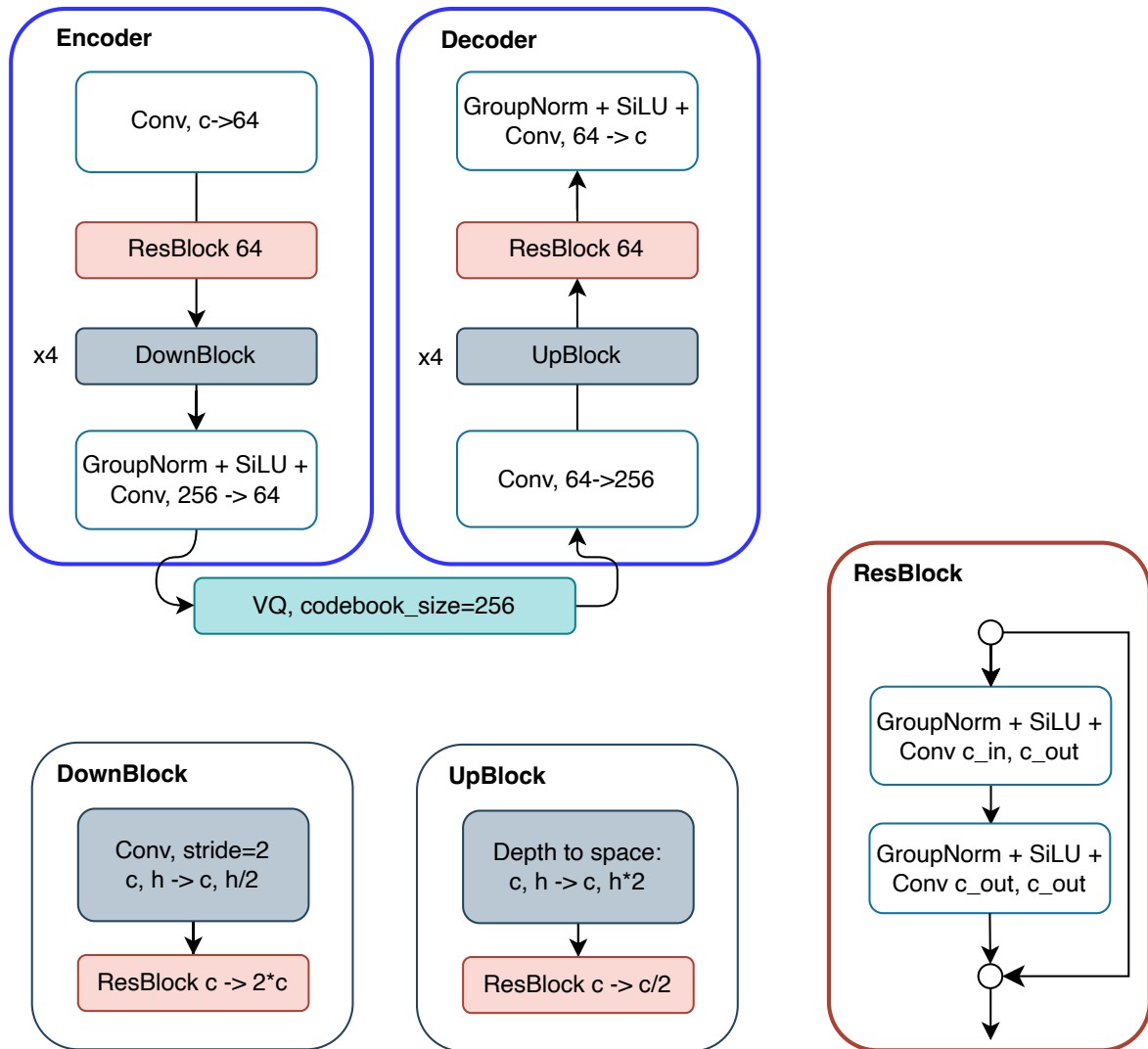

*Figure 11.* Architecture of `Zebra`'s VQVAE for 1D datasets. Each convolution acts only on the spatial dimension and uses a kernel of size 3. The Residual Blocks are used to process information and increase or decrease the channel dimensions, while the Up and Down blocks respectively up-sample and down-sample the resolution of the inputs. In 1D, we used a spatial compression factor of 16 on all datasets. Every downsampling results in a doubling of the number of channels, and likewise, every upsampling is followed by a reduction of the number of channels by 2. We choose a maximum number of channels of 256.

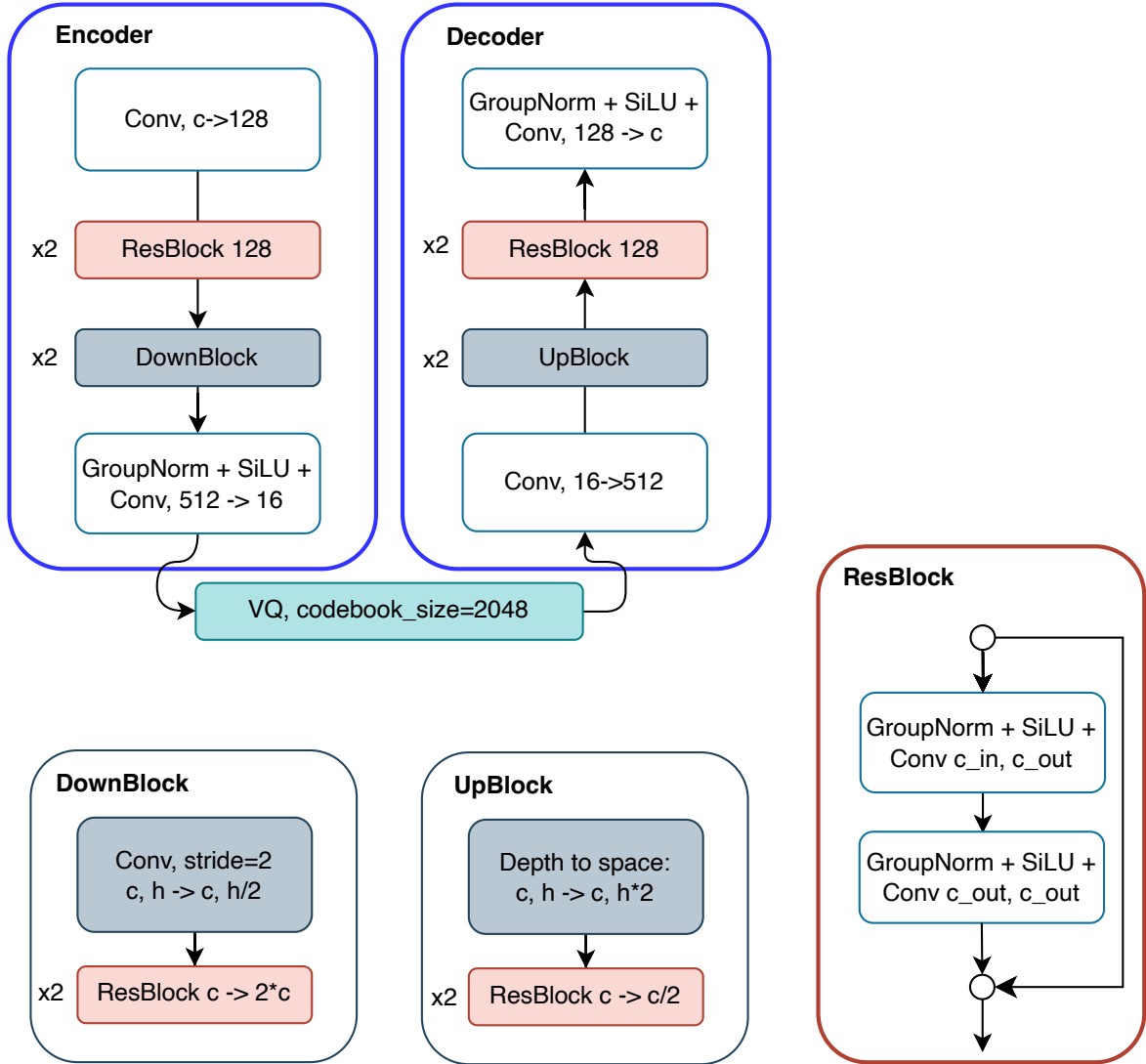

*Figure 12.* Architecture of Zebra's VQVAE for 2D datasets. Each convolution acts only on the spatial dimensions and uses a kernel of size 3. The Residual Blocks are used to process information and increase or decrease the channel dimensions, while the Up and Down blocks respectively up-sample and down-sample the resolution of the inputs. In 2D, we used a spatial compression factor of 4 for *Vorticity*, and 8 for *Wave2D*. Every downsampling results in a doubling of the number of channels, and likewise, every upsampling is followed by a reduction of the number of channels by 2. We choose a maximum number of channels of 1024.

*Table 8.* Hyperparameters for Zebra's VQVAE

| Hyperparameters | *Advection* | *Heat* | *Burgers* | *Wave b* | *Combined* | *Vorticity* | *Wave 2D* |
|---|---|---|---|---|---|---|---|
| start_hidden_size | 64 | 64 | 64 | 64 | 64 | 128 | 128 |
| max_hidden_size | 256 | 256 | 256 | 256 | 256 | 1024 | 1024 |
| num_down_blocks | 4 | 4 | 4 | 4 | 4 | 2 | 3 |
| codebook_size | 256 | 256 | 256 | 256 | 256 | 2048 | 2048 |
| code_dim | 64 | 64 | 64 | 64 | 64 | 16 | 16 |
| num_codebooks | 2 | 2 | 2 | 2 | 2 | 1 | 2 |
| shared_codebook | True | True | True | True | True | True | True |
| tokens_per_frame | 32 | 32 | 32 | 32 | 32 | 256 | 128 |
| start learning_rate | 3e-4 | 3e-4 | 3e-4 | 3e-4 | 3e-4 | 3e-4 | 3e-4 |
| weight_decay | 1e-4 | 1e-4 | 1e-4 | 1e-4 | 1e-4 | 1e-4 | 1e-4 |
| scheduler | Cosine | Cosine | Cosine | Cosine | Cosine | Cosine | Cosine |
| num_epochs | 1000 | 1000 | 1000 | 1000 | 1000 | 300 | 300 |

## D. Additional Quantitative results

### D.1. Alternative pretrainings

We experimented with different pretraining strategies before settling on the pretraining approach proposed in Zebra. The most intuitive way to adapt the next-token prediction objective for dynamics modeling is to operate in a continuous latent space, omitting the quantization step used in Zebra and therefore using a deterministic transformer instead of a generative model. As shown in previous studies (Li et al., 2024; Agarwal et al., 2025), we obtained better reconstruction results using an autoencoder instead of a VQVAE.

However, we encountered two critical challenges with this approach. First, the training loss plateaued quickly, as illustrated in Figure 13. Compared to training a generative transformer (with the negative log-likelihood) as in Zebra, the deterministic variant trained with MSE loss exhibited instability and failed to improve over training steps.

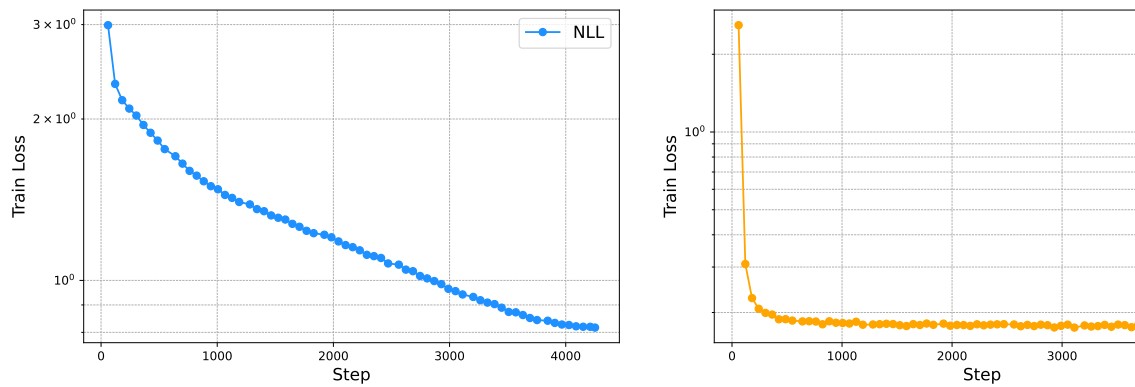

(a) Training loss (negative log-likelihood) with Zebra

(b) Training loss (MSE) with deterministic transformer

*Figure 13.* Comparison of the training process between the Zebra transformer and a deterministic transformer on *Advection*. In both cases, the model is trained to predict the next token. Zebra utilizes a discrete vocabulary and learns a probability distribution over the next token, whereas the deterministic transformer is optimized to predict the mean using MSE loss.

Second, while the model was able to predict the next token at inference, it could not generate an entire trajectory. It performed particularly poorly in the one-shot adaptation setting. As demonstrated in Figure 14, errors accumulated quickly during inference, leading to rapid divergence from the ground truth. This ultimately resulted in poor reconstructions when feeding the predicted tokens into the decoder, as seen in Figure 15.

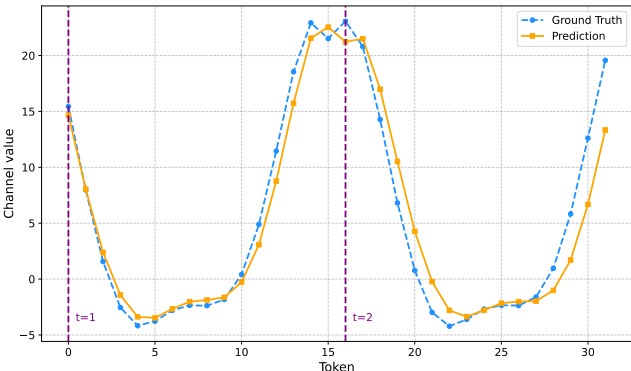

*Figure 14.* When trained with MSE for next-token prediction, inference suffers from instabilities, causing errors to grow exponentially. Here, we show the evolution over the sequence tokens of a particular channel to illustrate the phenomenon. The dashed lines show the temporal transitions between two subsequent frames, as here each frame is represented by 16 tokens.

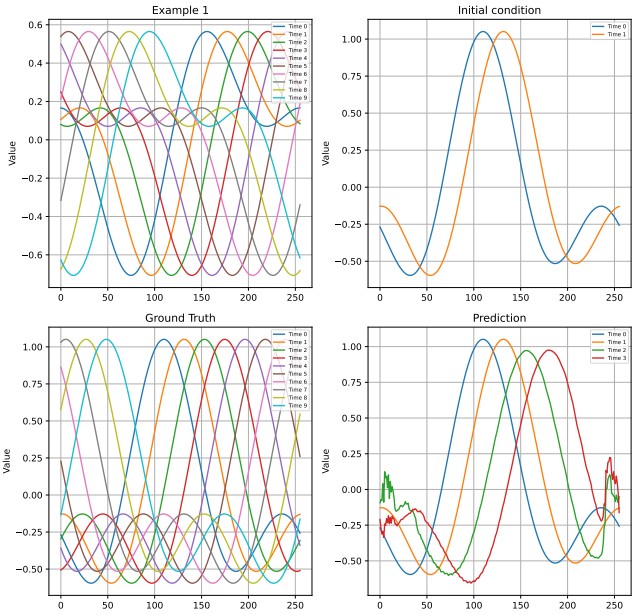

*Figure 15.* The deterministic transformer, trained with the next-token objective, performs poorly in the one-shot adaptation task.

These observations highlight the importance of the generative aspect when adopting a next-token objective. For this reason, we opted for a quantized representation combined with a transformer modeling a discrete distribution, a standard approach in image and video generation, but which has never been explored for modeling physical phenomena. While alternative strategies exist (Tian et al., 2024), they involve non-trivial extensions and are left for future works.

That said, next-token prediction pretraining may not be the only viable framework for developing in-context capabilities. To explore this, we experimented with a direct next-frame prediction approach using a deterministic setup trained with relative L2 loss. This method, which we called `VIT-in-context` serves as a baseline for evaluating other in-context pretrainings. It is based on a video transformer operating on patches with bidirectional attention. While its training behavior was more stable, inference results remained unsatisfactory.

### D.2. Uncertainty quantification

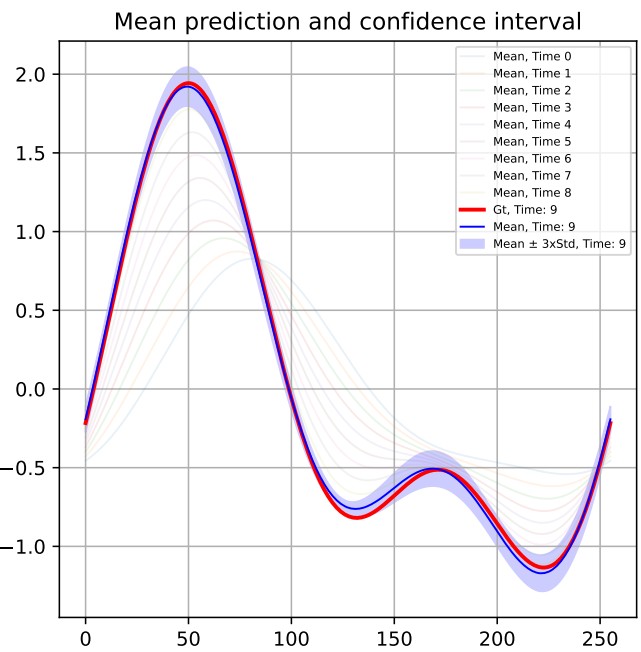

*Figure 16.* Uncertainty quantification with `Zebra` in a one-shot setting on *Heat* equation

**Setting** Since `Zebra` is a generative model, it allows us to sample multiple plausible trajectories for the same conditioning input, enabling the computation of key statistics across different generations. By calculating the pointwise mean and standard deviation, we can effectively visualize the model's uncertainty in its predictions. In Figure 16, the red curve represents the ground truth, the blue curve is the predicted mean and the blue shading indicates the empirical confidence interval ($3 \times$ standard deviation).

**Metrics** Motivated by this observation, we investigate how varying the model's temperature parameter $\tau$ affects its predictions; specifically in the one-shot adaptation setting described in Section 4.2. By adjusting $\tau$, we aim to assess its impact on both the accuracy and variability of the predictions. We generate 10 different trajectories and use them to compute several metrics. We employ four metrics to evaluate the model's uncertainty:

1. **Relative $L^2$ loss**: This assesses the accuracy of the generated trajectories by measuring the bias of the predictions relative to the ground truth.

2. **Relative standard deviation**: We estimate the variability of the predictions using the formula: Relative Std $= \frac{||\hat{\sigma}_*||_2}{||\hat{m}_*||_2}$ where $\hat{m}_*$ and $\hat{\sigma}_*$ represent the empirical mean and standard deviation of the predictions, computed pointwise across 10 generations.

3. **Confidence level**: We create pointwise empirical confidence intervals $CI(x) = [\hat{m}_*(x) - 3\hat{\sigma}_*(x), \hat{m}_*(x) + 3\hat{\sigma}_*(x)]$ and compute the confidence level as: Confidence level $= \frac{1}{n_x} \sum_x \mathbf{1}_{u_*(x) \in CI(x)}$. This score indicates how often the ground truth falls within the empirical confidence interval generated from sampling multiple trajectories.

4. **Continuous Ranked Probability Score** (CRPS, Gneiting & Raftery (2007) ) is a proper scoring rule that measures the accuracy of a probabilistic forecast by quantifying the difference between the predicted cumulative distribution function (CDF) and the empirical CDF of the observed value, with lower values indicating better calibration and sharpness.

5. **Root Mean Squared Calibration Error** (RMSCE, Hendrycks et al. (2018)) quantifies the discrepancy between predicted confidence and actual accuracy at a given confidence level.

**Results** When modeling uncertainty, the model achieves a tradeoff between the quality of the mean prediction approximation and the guarantee for this prediction to be in the corresponding confidence interval. Figure 17 illustrates the trade-off between mean prediction accuracy and uncertainty calibration. At lower temperatures, we achieve the most accurate predictions, but with lower variance, i.e. with no guarantee that the target value is within the confidence interval around the predicted mean. Across most datasets, the confidence level then remains low (less than 80% for $\tau < 0.25$), indicating that the true solutions are not reliably captured within the empirical confidence intervals. Conversely, increasing the temperature results in less accurate mean predictions and higher relative standard deviations, but the confidence intervals become more reliable, with levels exceeding 95% for $\tau > 0.5$. Therefore, the temperature can be calibrated depending on whether the focus is on accurate point estimates or reliable uncertainty bounds.

To better calibrate this temperature, we can therefore use a proper scroring metric such as the CRPS, and we can pick the temperature parameter that has the lowest CRPS value for a given value (see Figure 17). We can see that Zebra models really well the distribution for Combined and Advection and not so much for Burgers and Heat somehow.

Finally, we examine how the model's uncertainty evolves as additional information is provided as input. Specifically, we compare Zebra's average error and relative uncertainty when conditioned on one example trajectory, with one or two frames as initial conditions. Table 9 reports the relative L2 loss and relative standard deviation for both scenarios. The results clearly show that including the first two frames as initial conditions reduces both the error and the relative standard deviation consistently. This indicates that, while some of the uncertainty remains aleatoric, the epistemic uncertainty is reduced as more input information becomes available.

*Table 9.* Uncertainty quantification in the one-shot setting. Conditioning from a trajectory example and 1 frame or 2 frames as initial conditions. Metrics include relative $L^2$ loss (average accuracy) and relative standard deviation (average spread around the average prediction). The temperature is fixed at 0.1.

|  |  | *Advection* | *Heat* | *Burgers* | *Wave b* | *Combined* |
|---|---|---|---|---|---|---|
| Rel. $L^2$ | 1 frame | 0.006 | 0.156 | 0.115 | 0.154 | 0.008 |
| Rel. $L^2$ | 2 frames | 0.004 | 0.047 | 0.052 | 0.075 | 0.005 |
| Rel. Std. | 1 frame | 0.003 | 0.062 | 0.048 | 0.074 | 0.005 |
| Rel. Std. | 2 frames | 0.002 | 0.019 | 0.018 | 0.040 | 0.003 |

**Comparison with ViT-in-Context** We quantitatively evaluate the uncertainty using CRPS and RMSCE in the one-shot prediction task, as reported in Table 10, comparing `Zebra` with ViT-in-Context. Since ViT-in-Context is a deterministic model, we introduce stochasticity by: (1) adding Gaussian noise with a small standard deviation ($\sigma = 0.1$) to the input, thereby perturbing the input; and (2) enabling stochastic depth at inference time by keeping the DropPath mechanism (Huang et al., 2016). Across all settings, `Zebra` consistently outperforms these simpler baselines, often by an order of magnitude in both CRPS and RMSCE.

**Uncertainty over time** Finally, we illustrate in Table 11 how CRPS and RMSCE evolve over successive autoregressive steps in the predicted trajectory for `Zebra` on the *Burgers* equation. As expected, CRPS increases over time due to error accumulation during rollout. In contrast, RMSCE remains stable, indicating that the model maintains consistent calibration throughout the prediction.

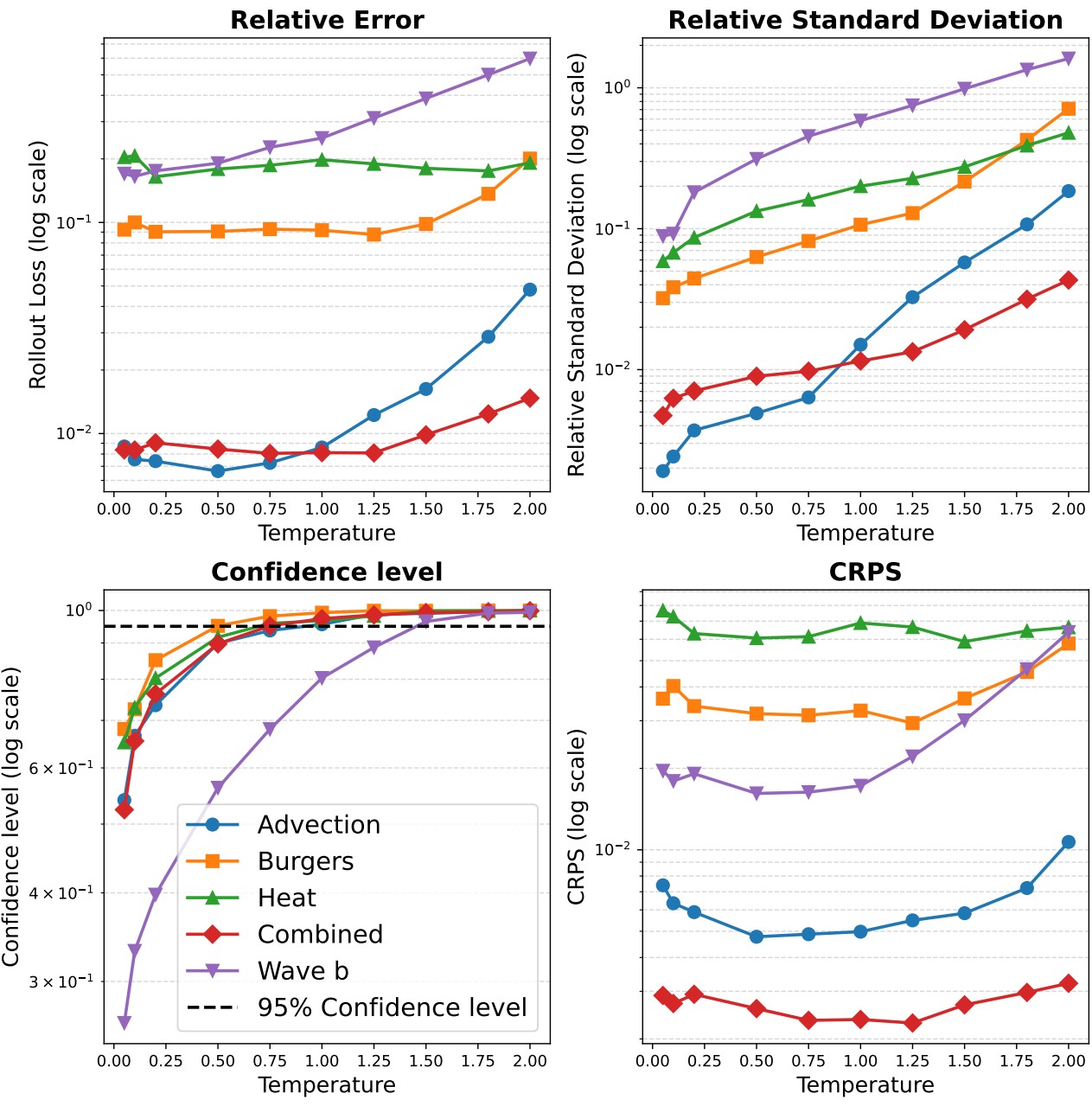

*Figure 17.* Uncertainty quantification with `Zebra`. The main parameter of this study is the temperature (x-axis). We then look from left to right at (1) The rollout loss, i.e. the relative $L^2$ loss between the predictions and the ground truth; (2) The relative standard deviation to quantify the spread around the mean; (3) The confidence level, that measures the frequency of observations that lie within the empirical confidence interval. (4) The CRPS that measures the quality of the uncertainty, can be used to pick the temperature with the most calibrated uncertainty.

*Table 10.* CRPS and RMSCE results across PDE benchmarks. Lower is better.

| Metric | Model | Advection | Heat | Burgers | Wave b | Combined |
|--------|-------|-----------|------|---------|--------|----------|
| | ViT + noise | 0.0705 | 0.176 | 0.227 | 0.093 | 0.098 |
| CRPS | ViT Dropout | 0.0363 | 0.213 | 0.196 | 0.024 | 0.024 |
| | Zebra | **0.0026** | **0.043** | **0.020** | **0.0129** | **0.0018** |
| | ViT + noise | 0.132 | 0.241 | 0.265 | 0.249 | 0.045 |
| RMSCE | ViT Dropout | 0.386 | 0.547 | 0.529 | 0.340 | **0.064** |
| | Zebra | **0.074** | **0.055** | **0.048** | **0.124** | 0.074 |

*Table 11.* CRPS and RMSCE over time for Zebra on the *Burgers* dataset.

| Metric | $t = 1$ | $t = 2$ | $t = 3$ | $t = 4$ | $t = 5$ | $t = 6$ | $t = 7$ | $t = 8$ | $t = 9$ |
|--------|------|------|------|------|------|------|------|------|------|
| CRPS | 0.0057 | 0.0106 | 0.0148 | 0.0184 | 0.0216 | 0.0244 | 0.0271 | 0.0296 | 0.0320 |
| RMSCE | 0.0511 | 0.0513 | 0.0505 | 0.0487 | 0.0483 | 0.0489 | 0.0467 | 0.0457 | 0.0457 |

### D.3. Analysis of the generation

`Zebra` is capable of generating completely novel trajectories for new environments, including the initial conditions. An example of a generated trajectory for *Vorticity 2D* is shown in Figure 18, where the top row shows the context trajectory used to guide the generation, and the bottom row displays the model's generated trajectory, including the initial condition. In this section, we evaluate on *Combined* and *Advection* the quality of the generated trajectories with `Zebra`.

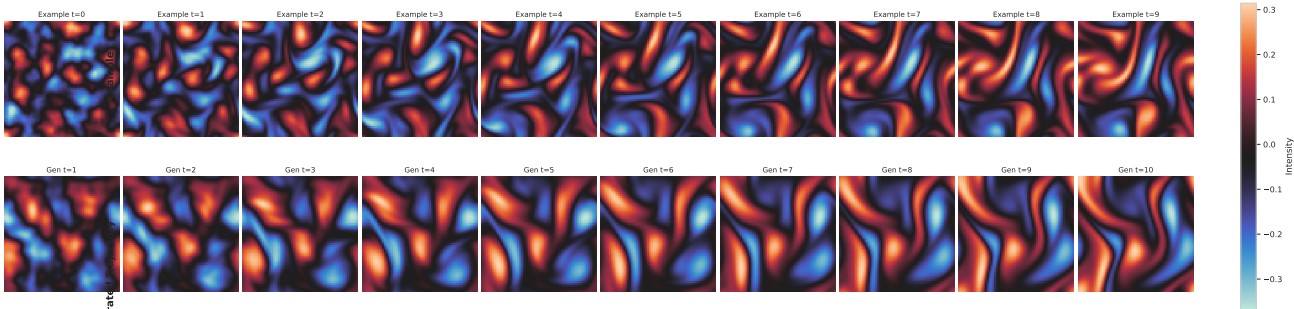

*Figure 18.* Unconditional generation on *Vorticity 2D*. The top-row is the example used to guide the generation, and the bottom-row is the generated example. The model also generates the initial condition.

**Setting** We evaluate whether our pretrained model can generate new samples conditioned on a trajectory observed in a previously unseen test environment. Unlike previous settings, the transformer is not explicitly conditioned on tokens derived from a real initial condition. Instead, we expect it to generate trajectories, including their initial conditions, that follow the same dynamics as the observed context.

Our evaluation focuses on four key aspects. First, we assess whether the generated trajectories faithfully follow the dynamics of the context. Second, we analyze the diversity of the generated trajectories to determine if they are significantly different from one another. Third, we compare the generated samples with those produced by numerical solvers to evaluate whether their distributions align. Finally, we examine the types of initial conditions generated by `Zebra`.

**Metrics** To assess *fidelity*, we generate ground truth trajectories using the physical solver originally used to construct the dataset. These simulations start from the initial conditions generated by `Zebra`, using ground truth environment parameters that `Zebra` itself does not have access to. We then compute the $L^2$ distance between the generated trajectories and those obtained from the physical solver.

For *diversity*, we measure the average pairwise $L^2$ distance between different trajectories generated by Zebra. The results for both fidelity and diversity are reported in Table 12 for the Advection and Combined Equations.

To compare the distribution of generated trajectories with that of numerical solvers, we compute the Wasserstein distance using the Sinkhorn algorithm (Cuturi, 2013). As baselines, we compare against two references. First, we compute the Wasserstein distance with purely random samples drawn from Gaussian noise, providing an upper bound on the problem. Second, we measure the Wasserstein distance between two independent sets of numerical solver trajectories (the validation and test sets), allowing us to quantify the variability inherent in the dataset itself.

Finally, as a qualitative analysis, we perform a principal component analysis on the trajectories generated by Zebra and visualize the first two principal components in Figure 19 for *Combined*.

**Sampling**  We use a default temperature of $\tau = 1.0$. For each context trajectory, we sample 10 new trajectories in parallel.

**Results**  Table 12 shows that Zebra generates new initial conditions and trajectories that respect the same physical laws as the given context. The model seems to have learned the statistical relationships between initial conditions and later timestamps. The high average $L^2$ distance between samples indicates that the generated trajectories are diverse. This can also be observed in Figure 19, where the generated samples effectively cover the distribution of real samples.

*Table 12.* Fidelity and diversity metrics. The $L^2$ distance measures fidelity to the context dynamics, while the average $L^2$ quantifies sample diversity.

| Model | $L^2$ | Average $L^2$ between samples |
|---|---|---|
| Advection | 0.0185 | 1.57 |
| Combined Equation | 0.0136 | 1.59 |

To further assess distribution alignment, we compute the Wasserstein distance between the generated trajectories and those obtained with numerical solvers. The results in Table 13 indicate that Zebra achieves lower Wasserstein distances than Gaussian noise but remains slightly above the cross-distribution baseline (which compares the validation and test distributions), suggesting reasonable alignment with the true data distribution.

*Table 13.* Comparison of distributions using the Wasserstein distance between Zebra-generated trajectories and numerical solver samples.

| Distance Metric | Advection | Combined Equation |
|---|---|---|
| Gaussian noise vs. real data | 18.22 | 16.15 |
| Validation data vs. test data | 5.11 | 1.87 |
| Zebra-generated data vs. real data | 5.57 | 2.21 |

### D.4. Dataset scaling analysis

We investigate how the one-shot error on the test set evolves as we vary the size of the training dataset. To this end, we train the auto-regressive transformer on datasets containing 10, 100, 1000, and 12,000 trajectories and evaluate Zebra's generations on the test set, starting with two frames as inputs. The training time is proportional to the dataset size: for example, the number of training steps for 1,000 trajectories is 10 times the number of steps for 100 trajectories. The results are presented in Figure 20.

First, we observe that Zebra requires a substantial amount of data to generalize effectively to different parameter values, even within the training distribution. This aligns with findings in the literature that transformers, especially auto-regressive transformers, excel at scaling —performing well on very large datasets and for larger model architectures. However, for smaller datasets, this approach may not be the most efficient. We believe that Zebra's potential resides when applied to large amounts of data, making it an ideal candidate for scenarios involving large-scale training.

Second, for the *Combined equation*, we notice that performance plateaus between 100 and 1,000 trajectories. This may be due to insufficient training or a lack of diverse examples, as the *Combined equation* is more challenging compared to the *Advection equation*, whose performance follows a more log-linear trend. This suggests that additional data or targeted training strategies might be needed to achieve better generalization for more complex equations.

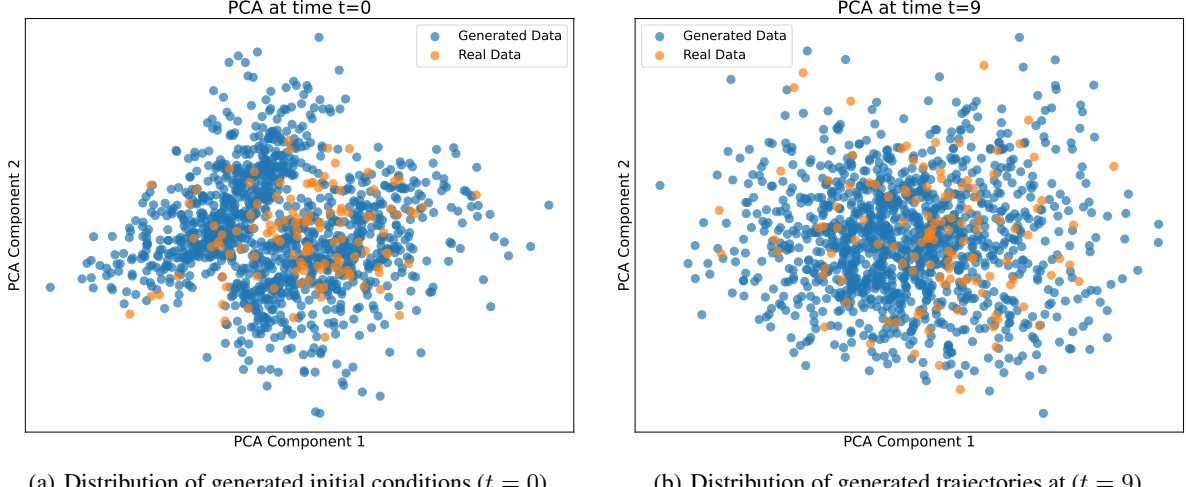

(a) Distribution of generated initial conditions ($t = 0$).    (b) Distribution of generated trajectories at ($t = 9$).

*Figure 19.* Qualitative analysis of generated trajectories. `Zebra` generates new initial conditions and trajectories for unseen test environments. PCA projections visualize both generated and true trajectories in a lower-dimensional space at $t = 0$ and $t = 9$.

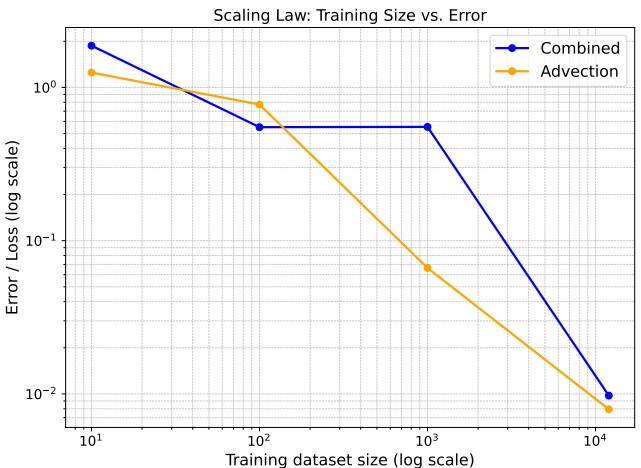

*Figure 20.* Dataset scaling analysis. One-shot prediction error on the test set vs. the training dataset size.

### D.5. Inference time comparison

Table 14 compares the inference time for one-shot adaptation across different methods when predicting a single trajectory given a context trajectory and an initial condition. For `Zebra`, the inference process, which includes encoding, auto-regressive prediction, and decoding, is much faster in 1D and slightly faster in 2D. With `Zebra`, the bottleneck at inference is the autoregressive generation of tokens, which speed is about 128 tokens per second on a V100 for 2D and an RTX for 1D. The decoding is fast and can be done in parallel for the trajectory in one forward pass. In contrast, for CODA and CAPE, the majority of the inference time is spent on adaptation and gradient-based steps. Here the times were reported with 100 gradient steps, note that we used 250 for the rest of the experiments. We believe Zebra's inference time could be further optimized by (1) improving the optimization code and leveraging specialized hardware such as H100 (for flash attention) and LPUs (which show significant speed-ups against GPUs), and (2) increasing the number of tokens sampled per step (as in e.g. next-scale prediction (Tian et al., 2024)).

However, we have shown that it is possible to greatly accelerate inference by employing a deterministic neural surrogate on top of `Zebra`, which acts as a dynamic encoder. This framework is order of magnitudes faster than gradient-based adaptation methods.

*Table 14.* Inference times for one-shot adaptation. Average time in seconds to predict a single trajectory given a context trajectory and an initial condition. Times include adaptation and forecast for CODA and CAPE, while it includes encoding, auto-regressive prediction and decoding for Zebra.

|              | *Advection* | *Vorticity 2D* |
|--------------|-------------|----------------|
| CAPE         | 18s         | 23s            |
| CODA         | 31s         | 28s            |
| Zebra        | 3s   | 21s     |
| Zebra + UNet | **0.10s**   | **0.14s**      |

### D.6. Influence of the codebook size

The codebook size $K$ is a crucial hyperparameter. It directly affects the quality of the reconstructions, since a larger codebook can improve the reconstructions quality. However, it also impacts the dynamics modeling stage: the smaller the codebook, the easier it is for the transformer to learn the statistical correlations between similar trajectories. To have a sense of this trade-off, we report the relative reconstruction errors and the one-shot prediction errors in Table 15. The reconstruction error decreases when the codebook size increases. However, the one-shot prediction error decreases from 32 to 64 codes but then gradually increases from 64 to 512. We can see that it follows a U-curve in Figure 21. This phenomenon was observed in a different context in (Cole et al., 2024).

*Table 15.* Influence of the codebook size. Reconstruction error and one-shot prediction error on *Burgers* for different codebook sizes. Errors in relative L2.

| Codebook Size | Reconstruction | One-shot Prediction |
|---------------|----------------|---------------------|
| 32            | 0.0087         | 0.116               |
| 64            | 0.0043         | 0.097               |
| 128           | 0.0024         | 0.124               |
| 256           | 0.0019         | 0.163               |
| 512           | 0.0015         | 1.093               |

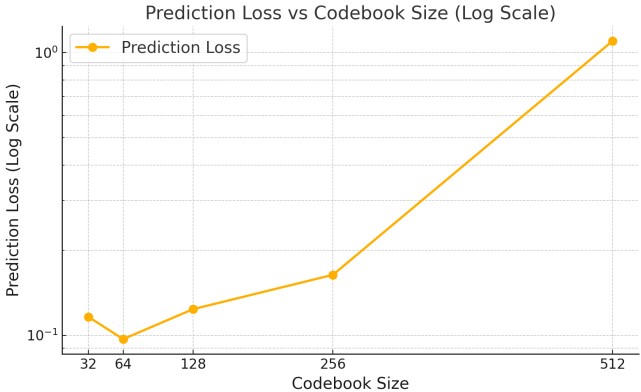

*Figure 21.* One-shot accuracy vs codebook size. One-shot prediction error on the test set for various codebook sizes. Error in relative L2.

### D.7. Reconstruction errors

We report the accuracy of the reconstructions from our encoder-decoder for different datasets in Table 16. Here, no dynamics is involved, we simply evaluate the quality of the encoding and of the decoding. On 1D and 2D datasets, the decoding errors are respectively of 0.1 % and 1% on the test set.

*Table 16.* Reconstruction errors. Test relative L2 loss between reconstructions from Zebra's VQVAE and the ground truths.

|  | *Advection* | *Heat* | *Burgers* | *Wave b* | *Combined* | *Wave 2D* | *Vorticity 2D* |
|---|---|---|---|---|---|---|---|
| VQVAE of `Zebra` | 0.0003 | 0.0019 | 0.0016 | 0.0011 | 0.0022 | 0.010 | 0.017 |

To assess how faithful signal encoding and deconding influences prediction accuracy, we evaluate `Zebra` in an out-of-distribution setting on the *Vorticity 2D* dataset across several viscosity ranges. As viscosity decreases and departs from the training distribution, both the context and target trajectories exhibit increasingly high-frequency components which were not seen by the VQVAE. We report reconstruction scores alongside one-shot prediction errors in Table 17.

*Table 17.* Impact of encoding and decoding quality on prediction accuracy. Reconstruction and one-shot prediction error with `Zebra`. Metric is relative L2.

| Viscosity Range | Type | Reconstruction | One-shot Prediction |
|---|---|---|---|
| $[10^{-3}, 10^{-2}]$ | In-distribution | 0.02 | 0.12 |
| $[5 \times 10^{-4}, 10^{-3}]$ | Close OoD | 0.13 | 0.24 |
| $[10^{-5}, 10^{-4}]$ | Far OoD | 0.22 | 0.32 |

As the viscosity decreases and deviates from the training distribution, the reconstruction error increases, which is expected since the VQVAE was not exposed to these regimes. However, the one-shot prediction error remains stable. This suggests that the encoder continues to capture the essential characteristics of the context dynamics in this one-shot OoD setting, enabling the transformer to leverage these sequence indices for accurate forecasting of the low-frequency components.

### D.8. Influence of the number of context examples

While all models were evaluated on a one-shot adaptation task, `Zebra` was trained with up to 5 context examples as input. This allows us to analyze the impact of context size on prediction accuracy. As shown in Table 18, performance saturates after approximately 3 in-context examples.

*Table 18.* Effect of number of context example. Few-shot prediction error (relative L2) on 1D datasets as a function of the number of context examples.

| # Examples | Advection | Heat | Burgers | Combined |
|:---:|:---:|:---:|:---:|:---:|
| 1 | 0.0074 | 0.1563 | 0.1150 | 0.0095 |
| 2 | 0.0077 | 0.1310 | 0.1020 | 0.0074 |
| 3 | 0.0072 | 0.1272 | 0.1000 | 0.0078 |
| 4 | 0.0071 | 0.1272 | 0.0990 | 0.0075 |
| 5 | 0.0071 | 0.1310 | 0.1000 | 0.0073 |

# E. Qualitative results

We provide visualizations of the trajectories generated with Zebra under different settings in the following figures. **One-shot prediction**: Figure 22, Figure 24, Figure 26, Figure 28, Figure 30, Figure 35, Figure 38. **Uncertainty quantification**: Figure 23, Figure 25, Figure 27, Figure 29, Figure 27.

## E.1. Advection

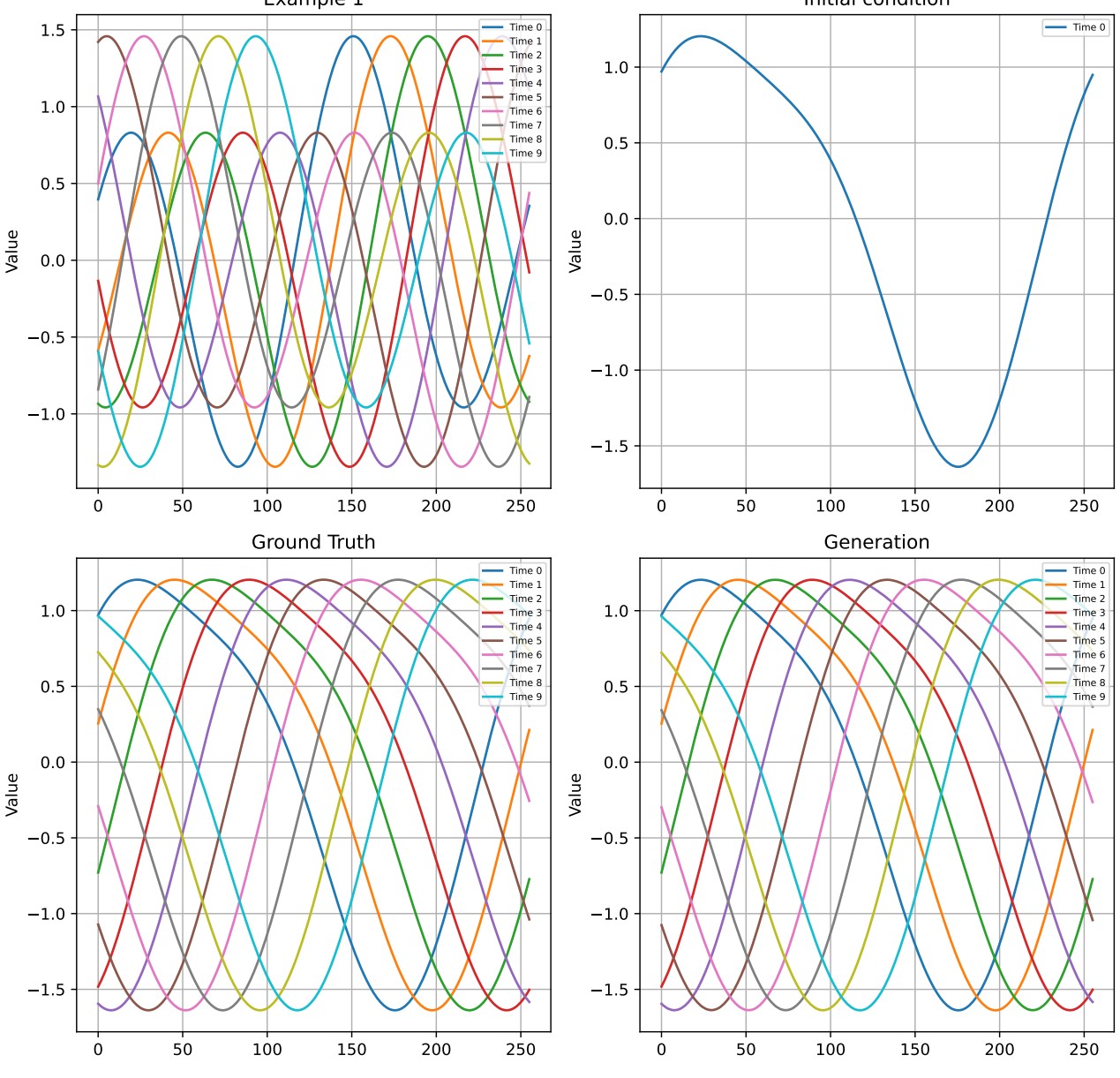

*Figure 22.* **One-shot** adaptation on Advection

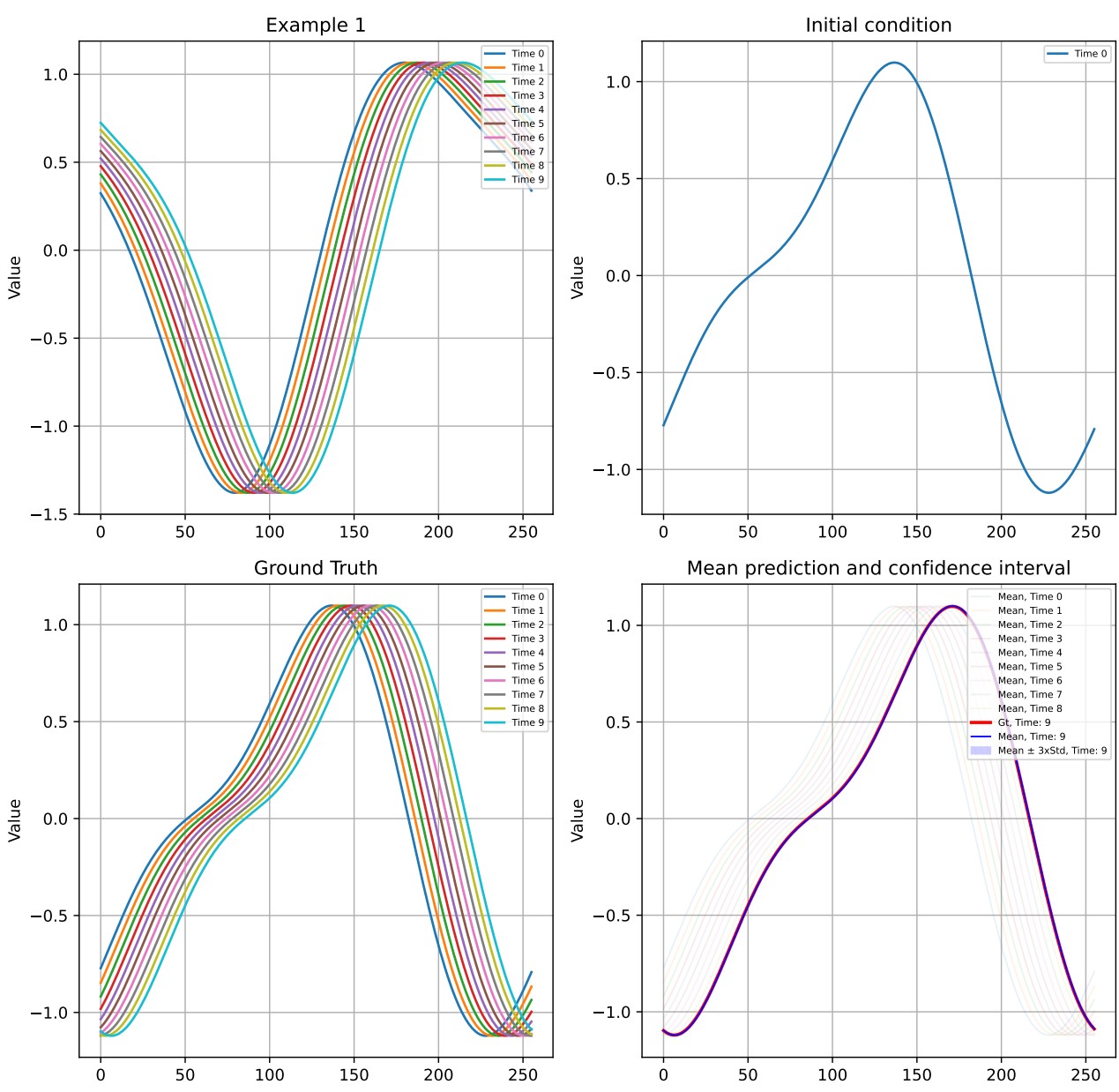

*Figure 23.* **Uncertainty quantification** on Advection

## E.2. Burgers

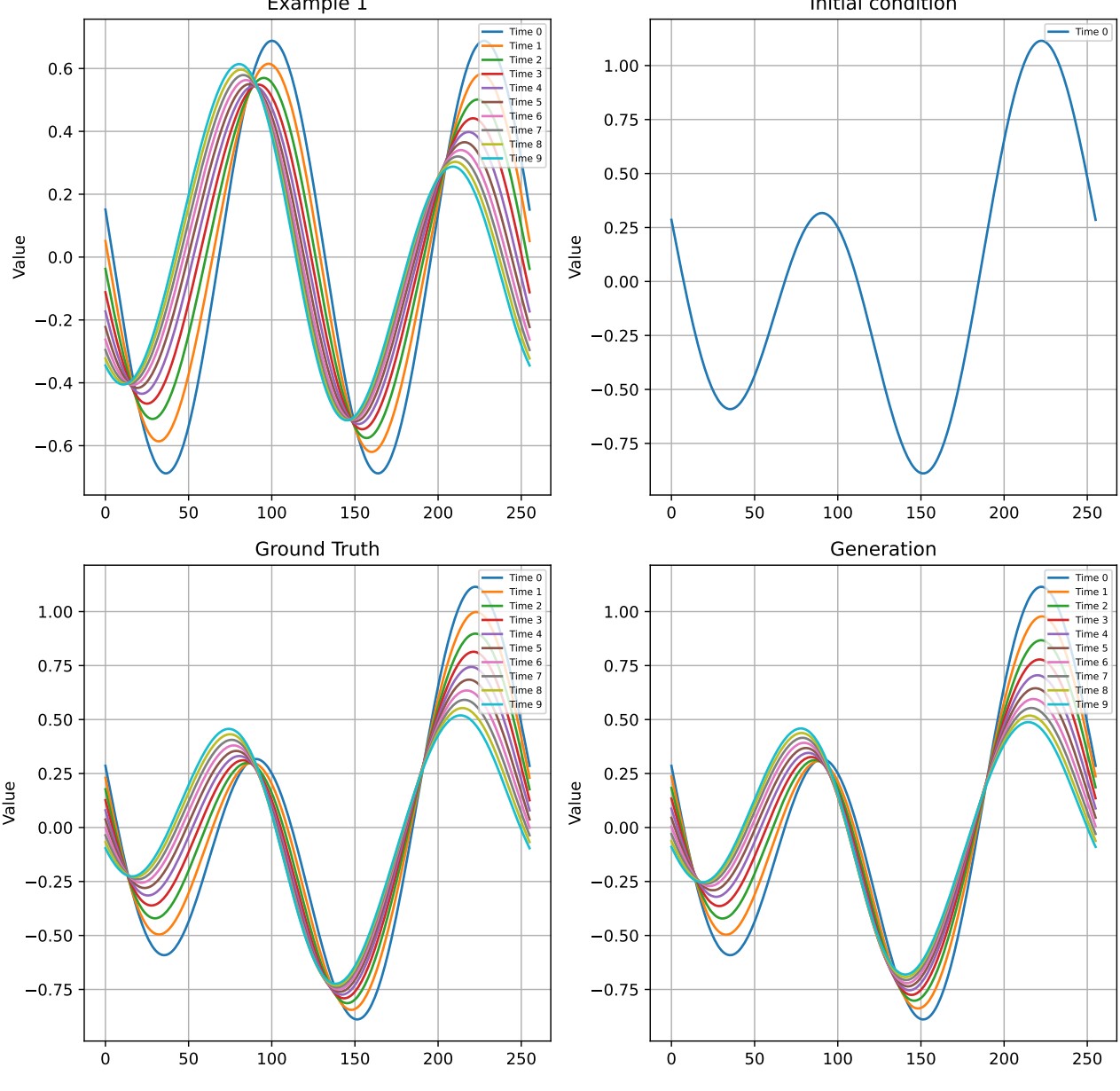

*Figure 24.* **One-shot** adaptation on Burgers

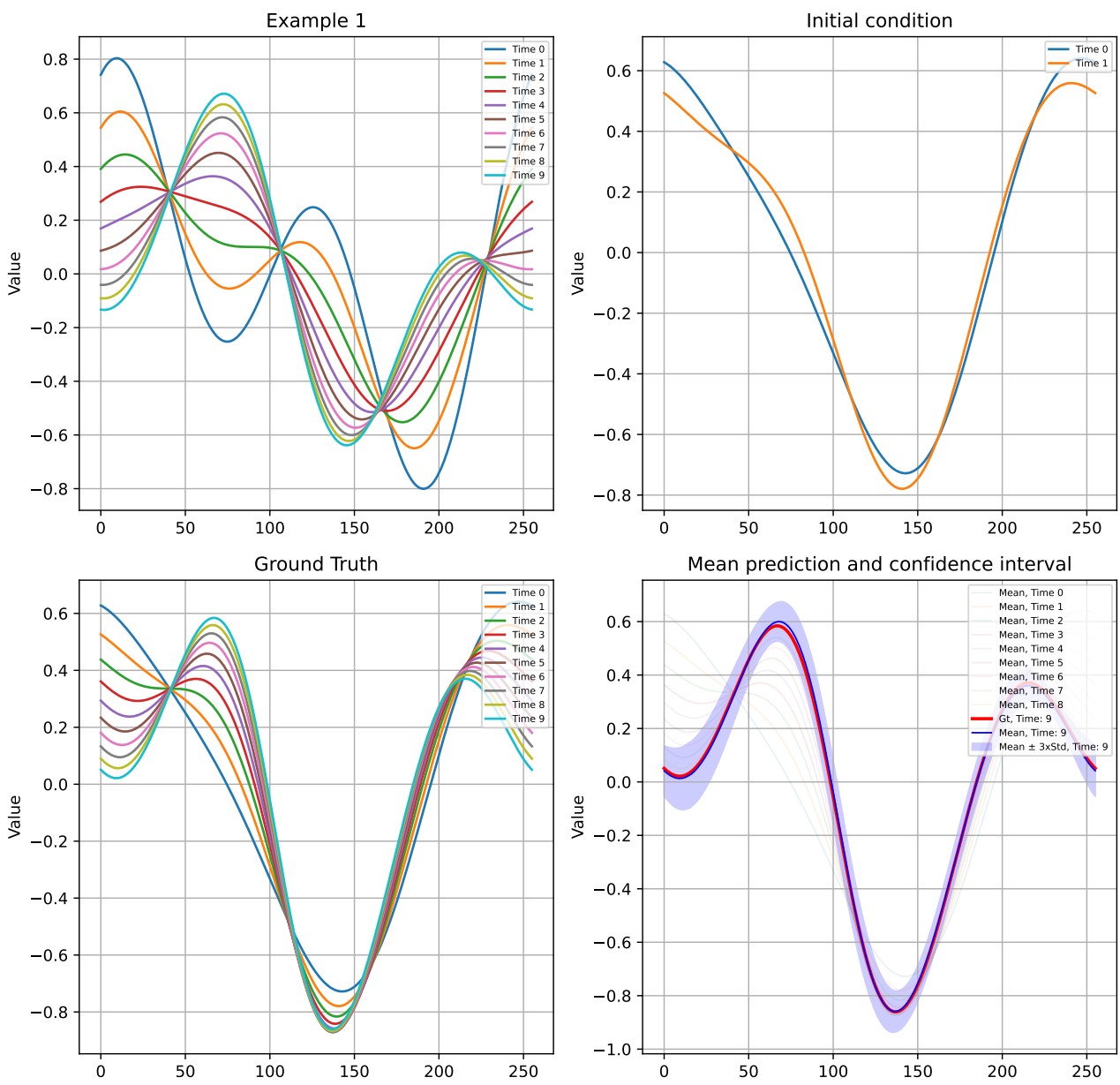

*Figure 25.* **Uncertainty quantification** on Burgers

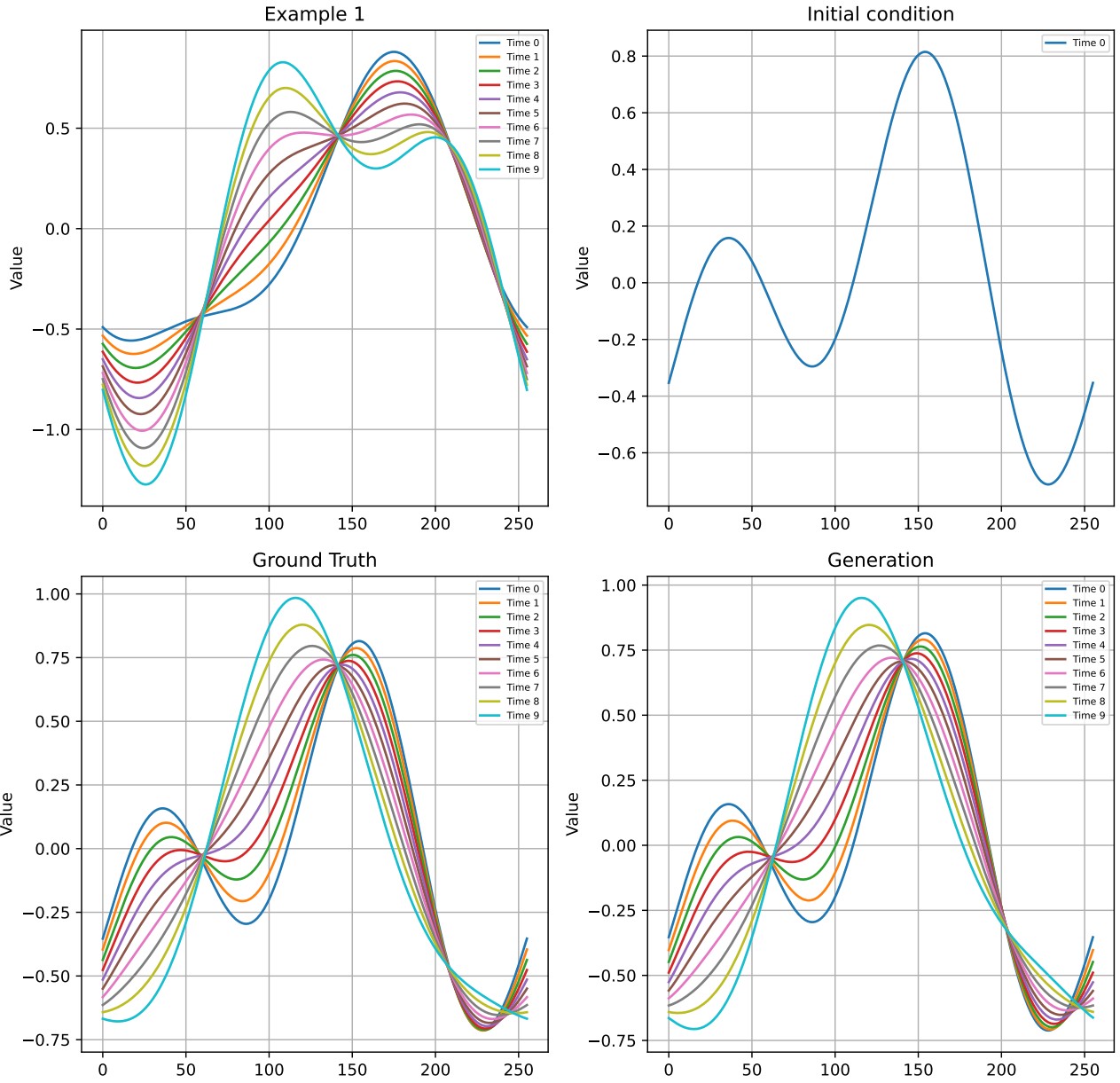

*Figure 26.* **One-shot** adaptation on Heat

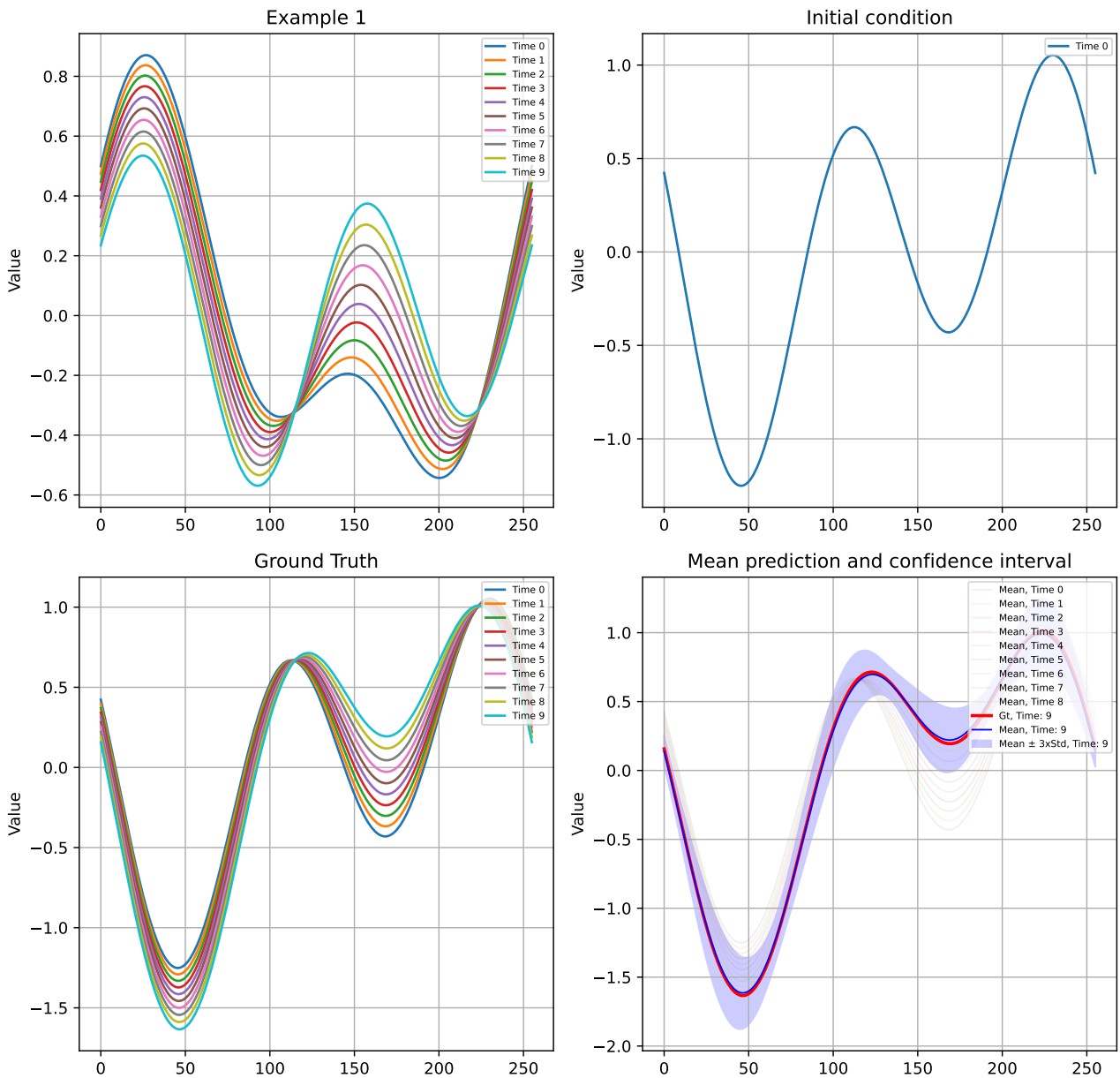

*Figure 27.* **Uncertainty quantification** on Heat

### E.3. Heat

### E.4. Wave boundary

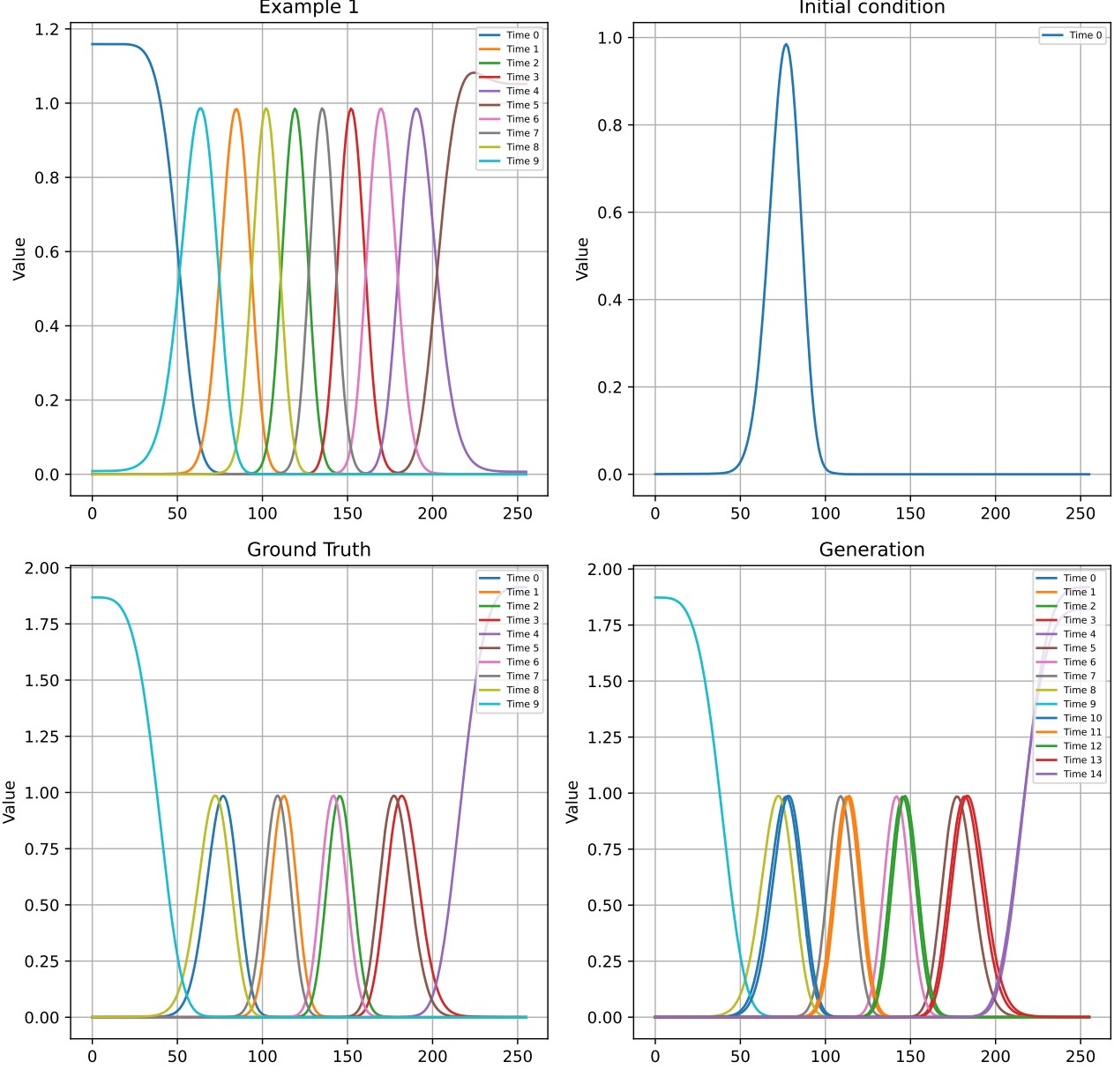

*Figure 28.* **One-shot** adaptation on Wave b

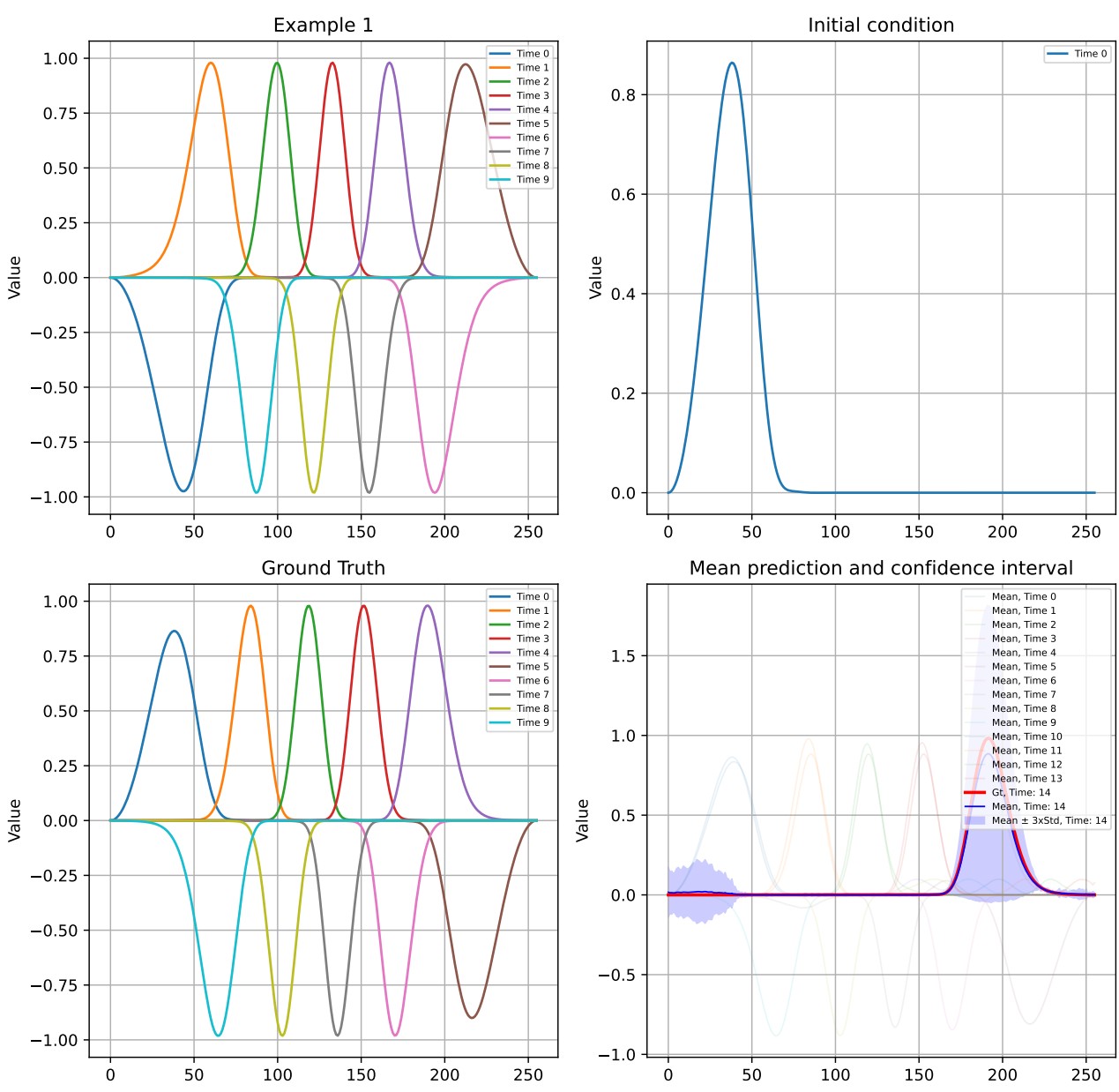

*Figure 29.* **Uncertainty quantification** on Wave b

## E.5. Combined equation

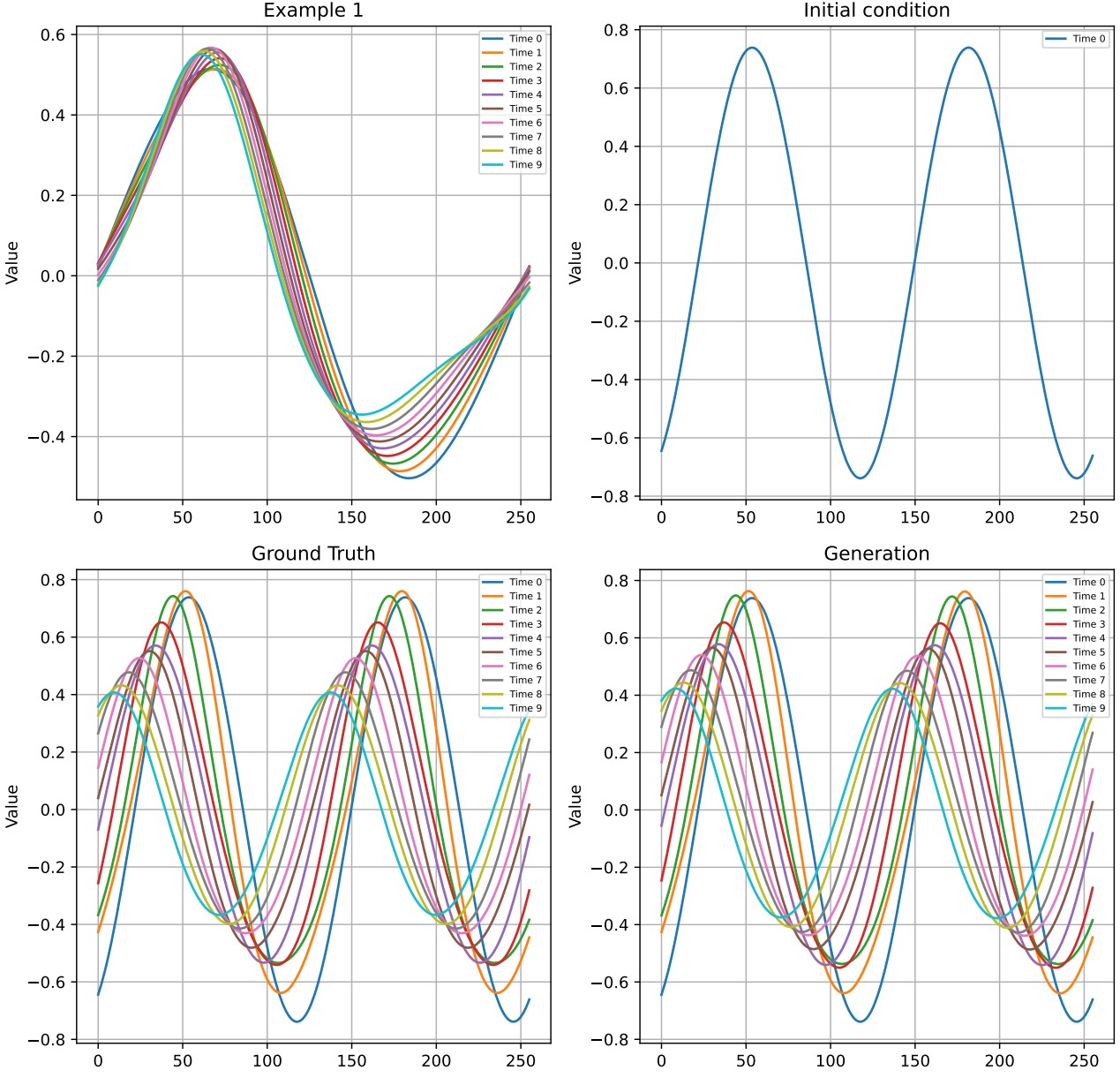

*Figure 30.* **One-shot** adaptation on Combined

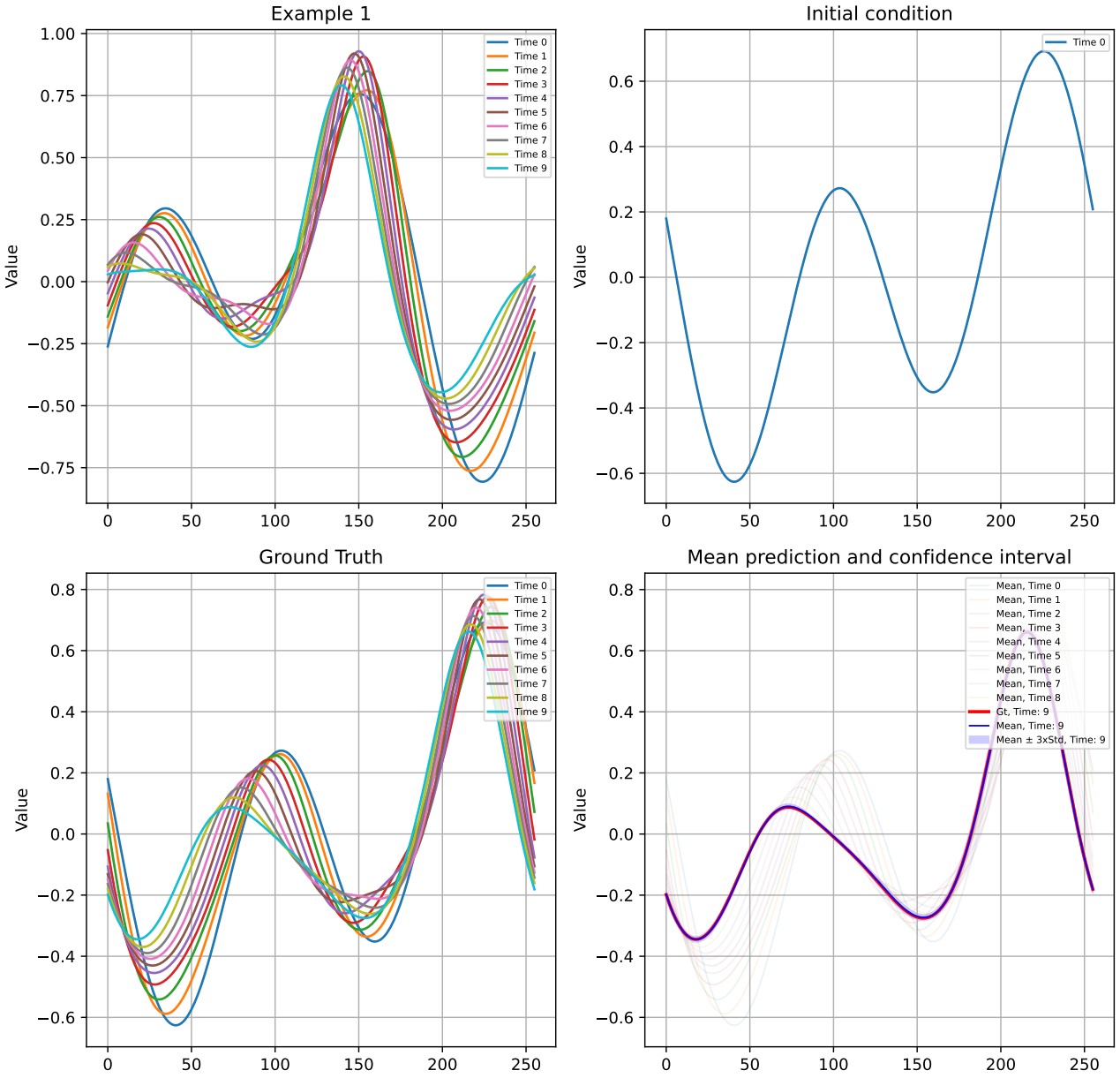

*Figure 31.* **Uncertainty quantification** on Combined equation

## E.6. Vorticity 2D

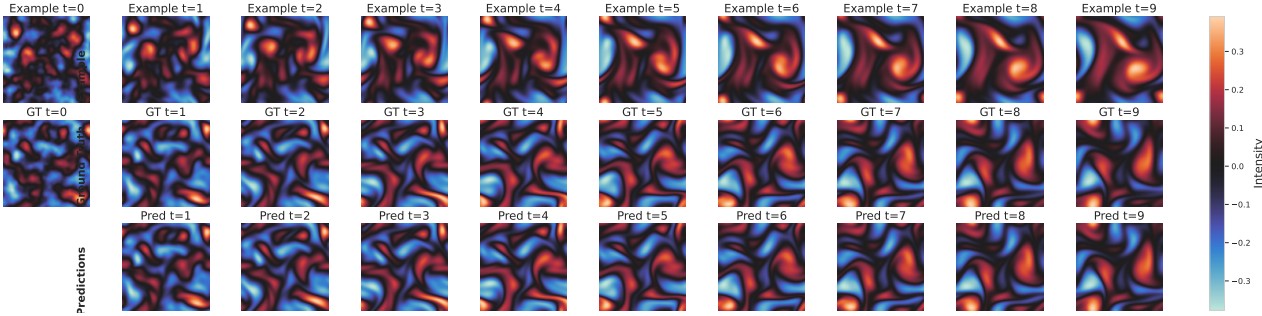

*Figure 32.* **One-shot** adaptation on *Vorticity 2D*. Example 1.

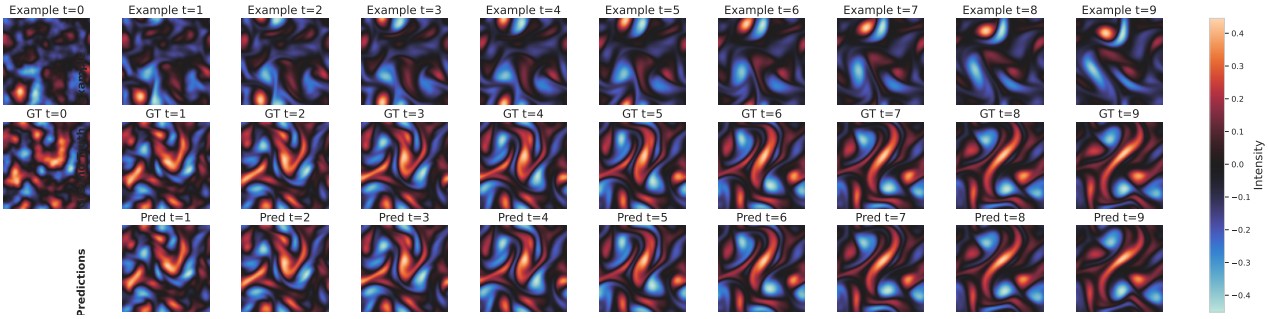

*Figure 33.* **One-shot** adaptation on *Vorticity 2D*. Example 2.

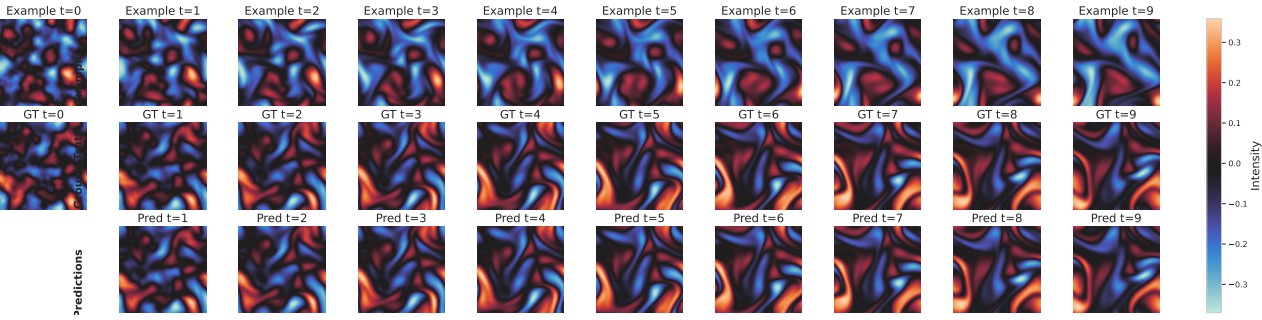

*Figure 34.* **One-shot** adaptation on *Vorticity 2D*. Example 3.

### E.6.1. OUT-OF-DISTRIBUTION

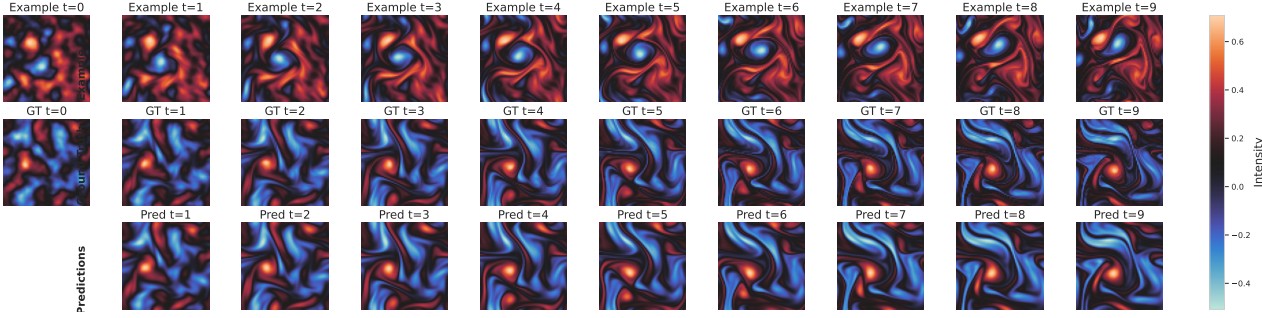

*Figure 35.* **One-shot OoD** adaptation on *Vorticity 2D*. Example 1.

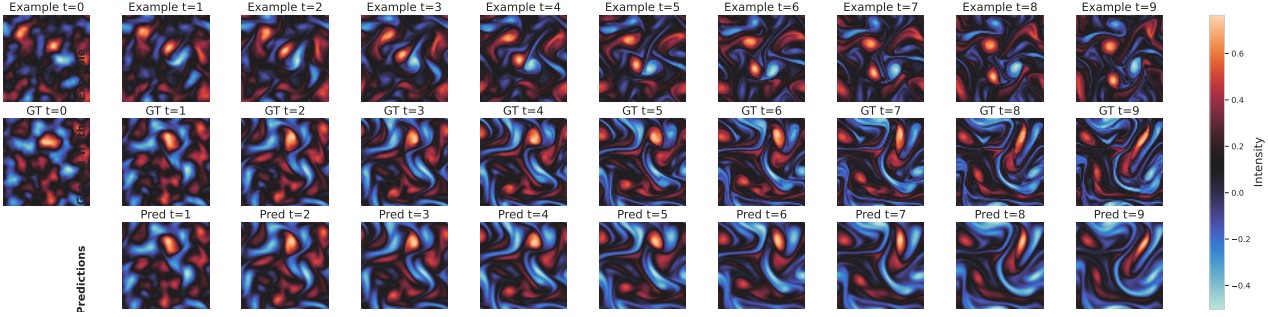

*Figure 36.* **One-shot OoD** adaptation on *Vorticity 2D*. Example 2.

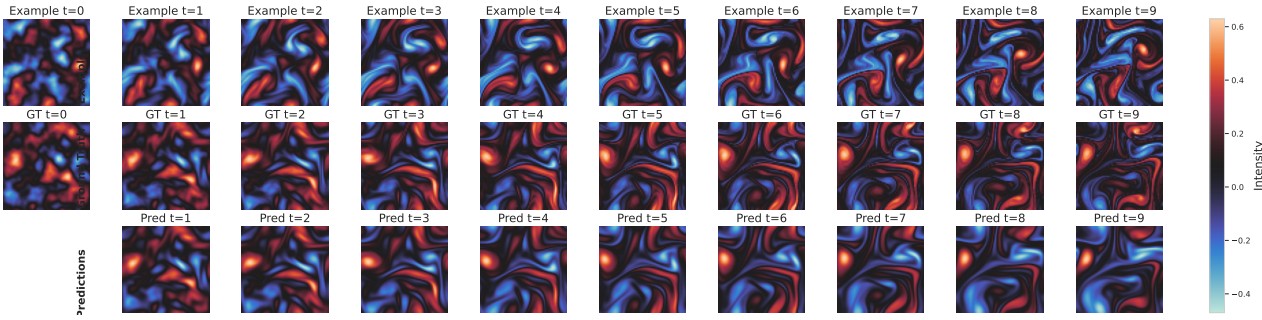

*Figure 37.* **One-shot OoD** adaptation on *Vorticity 2D*. Example 3.

## E.7. Wave 2D

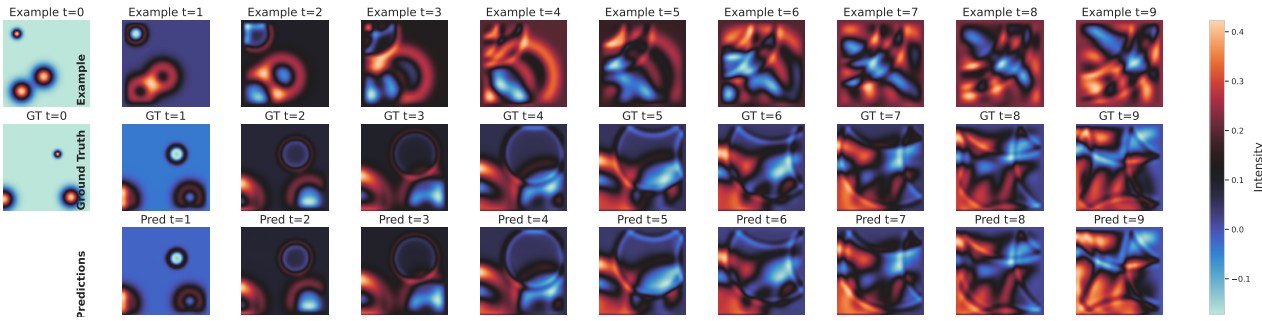

*Figure 38.* **One-shot** adaptation on *Wave 2D*. Example 1.

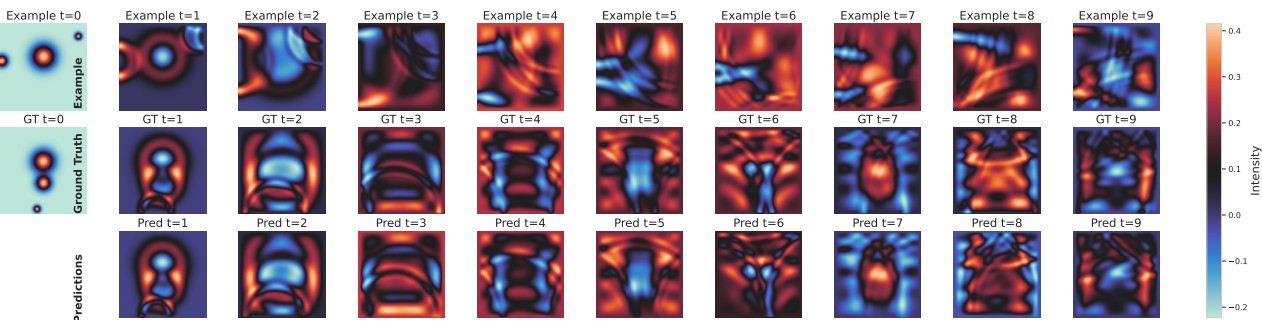

*Figure 39.* **One-shot** adaptation on *Wave 2D*. Example 2.

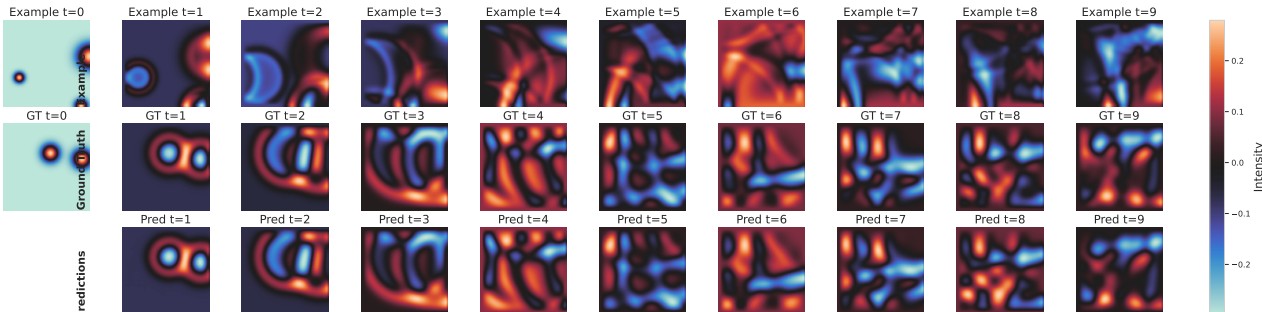

*Figure 40.* **One-shot** adaptation on *Wave 2D*. Example 3.

