# OpenReview forum: "Zebra: In-Context Generative Pretraining for Solving Parametric PDEs"
_ICML.cc/2025/Conference — ICML 2025 poster_

### Official Review · Reviewer_tD1F · 2025-03-05

**Overall Recommendation:** 2

**Summary:**

This paper proposes a model for modeling physical dynamics by leveraging the power of large language models (LLMs). The model handles data given in the physical domain through quantized representations, utilizing in-context learning. However, the advantages of this approach over other LLM-based foundation models are unclear, and the specific contribution to addressing dynamics governed by partial differential equations (PDEs) remains ambiguous.

**Claims And Evidence:**

Recent advancements in large language models LLMs have led to the proposal of several PDE-based foundation models using LLMs. However, it is unclear what specific advantages the proposed approach offers compared to existing models (S. Subramanian, 2023; M. Herde, 2024). The method seems to primarily incorporate widely used techniques like VQVAE, which have been extensively explored in deep learning literature, yet there is no discussion on how these techniques offer advantages for handling sequential data governed by PDE dynamics. It would be beneficial to include an ablation study on the VQVAE and special tokens to assess their impact.

The ability to handle downstream tasks is just as important as solving forward PDEs in foundation models. For this, the generalization ability is crucial. Where does the generalization ability of the proposed method come from? Does the proposed quantized representation offer inherent advantages in enhancing the model's generalization?


**References**

S. Subramanian et al., Towards Foundation Models for Scientific Machine Learning, 2023.

M. Herde et al., Poseidon: Efficient Foundation Models for PDEs, 2024.

**Essential References Not Discussed:**

There are no such essential related works that are missing from the paper.

**Experimental Designs Or Analyses:**

The paper conducts in-context training based on a ViT architecture, yet there is an existing an in-context model based on ViT (Y. Can et al, VICON: Vision In-Context Operator Networks for Multi-Physics Fluid Dynamics Prediction, 2024.) I recommend comparing performance, memory, and time efficiency between the proposed model and this existing model in a 2D setting.

**Methods And Evaluation Criteria:**

The training data used in this paper appears simplistic and does not reflect practical scenarios. The PDEs addressed are basic, and even the example of the Burgers' equation includes viscosity, which prevents shock formation, making it a challenging scenario. While these benchmarks are commonly used in the literature, it is important to acknowledge that the problems tackled in this work are not difficult. Additionally, practical data is rarely clean and often contains noise or corruption, as well as irregularly structured spatial data. I recommend testing the robustness of the proposed method in such more realistic and challenging scenarios.

**Other Comments Or Suggestions:**

There are instances in the main text where citations have two sets of parentheses, which should be corrected.

**Other Strengths And Weaknesses:**

Please refer to the above responses.

**Questions For Authors:**

Please refer to the above responses.

**Relation To Broader Scientific Literature:**

It is unclear how this paper contributes to advancements in both deep learning technology and PDE solvers.

**Theoretical Claims:**

This paper has no theoretical content.

---

> ### Author Rebuttal · Authors · 2025-03-31
>
> ### 1. On the originality
> There seems to be a misunderstanding, and we apologize if our explanations were not sufficiently clear. We appreciate the opportunity to clarify the originality of our contribution.
> Objective: Our goal is to develop a family of models capable of solving parametric PDEs. We argue that classical empirical risk minimization strategies are not well-suited for this task and instead advocate for a learning-to-adapt approach. To this end, we introduce a novel method and demonstrate, across a broad range of scenarios, that it performs comparably to—and often surpasses—state-of-the-art models.
> Models: To the best of our knowledge, this is the first instance of an LLM-based approach for PDE modeling. By LLM, we refer to generative probabilistic models, akin to the decoders used in language modeling. Due to their probabilistic nature, these models enable the generation of trajectory distributions. In contrast, the models you referenced (S. Subramanian, 2023; M. Herde, 2024), as well as those listed in Appendix A.1 under "The classical ML paradigm", are deterministic vision transformers (ViTs) rather than generative models. These models rely on fine-tuning when exposed to new data, making them ill-suited for scenarios with limited data, and they cannot be used to generate trajectory distributions. Additionally, in our experiments, ViTs performed poorly for in-context learning.
> Key Benefits and Novelty: Our formulation draws inspiration from the in-context learning capabilities of generative LLMs and explores their potential application to solving parametric PDEs. In this context, in-context learning means that, when provided with examples of a new task, the model can generate an appropriate response to a query. In our case, the examples consist of trajectories from the same dynamical system, and the queries correspond to new initial conditions. This approach enables the model to generalize to new dynamics using only a few examples, without requiring retraining. Achieving this level of generalization with minimal data and computational cost, compared to existing baselines, is precisely our objective.
> We acknowledge that solving downstream tasks may be important in certain contexts, but this falls outside the scope of our current work.
>
> ### 2. On the training data
> We respectfully disagree. Our datasets encompass seven distinct equations with a wide range of parameter values, leading to diverse and complex dynamics. These variations include differences in PDE coefficients, initial and boundary conditions, and forcing terms. As a result, the learning task is inherently challenging, as evidenced by the comparatively weaker performance of baseline models.
> Our model, consistently outperforms these baselines in both in-distribution and OoD settings. Additionally, our experimental setup considers a broader variety of parametric scenarios than the referenced papers. Even in the latest large-scale PDE benchmark, The Well [1], the distribution of PDE parameters is more limited compared to our experiments. In this sense, our evaluation setting presents a more rigorous challenge.
>
> [1] The well: a large-scale collection of diverse physics simulations for machine learning. Ohana et al. 2024.
>
> ### 3. On VICON
> We thank the reviewer for bringing this reference to our attention. We were not aware of this paper prior to submission, and we will include it in the related work section.
> However, a direct comparison is not feasible, as the referenced model addresses a different problem. While our approach focuses on adapting to a new parametric setting based solely on an initial condition—akin to traditional PDE solvers—the referenced work leverages past trajectory history for future-state forecasting. Additionally, no code is available for this unpublished paper, making direct empirical comparison difficult.
> That said, our ViT In-Context baseline already captures a setup that is closely aligned with the referenced work but adapted to our problem setting. This serves as an implicit comparison between the two strategies.
> Finally, we note that this paper was publicly released only in late November 2024. Under ICML guidelines, it therefore qualifies as concurrent work (see below).
> Excerpt from the reviewer guidelines at https://icml.cc/Conferences/2025/ReviewerInstructions
> “Concurrent Works
> Authors cannot expect to discuss other papers that have only been made publicly available within four months of the submission deadline. (This cut-off is adopted from the AISTATS and ICLR reviewing instructions.) Such recent papers should be considered as concurrent and simultaneous. Good judgement is necessary to decide whether a paper that has not yet been peer-reviewed should be discussed. The guideline is to follow the best practices of the specific subfield; the Area Chair can help in each case with these.”
>
> ### 4. Relation To Broader Scientific Literature
> We hope that the clarifications provided help to reassess this statement.

---

> > ### Comment · Reviewer_tD1F · 2025-04-08
> >
> > Thank you for your detailed response and for clarifying the proposed methodology. I apologize for asking you to compare with recent work on VICON. A brief mention of it in the revised version would be appreciated.
> >
> > Regarding the originality, I respectfully disagree with the statement that "this is the first instance of an LLM-based approach for PDE modeling," as there are several concurrent studies exploring the use of in-context learning with LLMs for solving PDEs. Additionally, although the approach using LLMs in this work is different, I am aware that the VICON series, named ICON, has been utilizing LLMs since 2023. While the proposed methodology empirically demonstrate some improvement in performance over existing approaches, and the use of diverse training data is certainly valuable, I believe it may be premature to claim this approach as a groundbreaking contribution to the scientific literature on PDE-based modeling.
> > In particular, while your methodology leverages the capabilities of LLMs to generate new trajectories that were not previously achievable, it still faces significant challenges in terms of interpretability and lacks guarantees or analysis regarding accuracy and stability. If there are applications where rapid processing of many problems is more important than stability or precise accuracy, it could be beneficial to highlight these aspects in the paper, as such scenarios might be more suitable for this approach.
> >
> > Regarding the classical solutions of data, there are already robust methods capable of solving these problems effectively without the need for complex AI approaches. The remaining challenges includes ill-posed PDEs, complex geometry, and high-dimensional problems. While LLM-based approaches are relatively new and still evolving, I remain uncertain about the extent of their contribution to this field if they are unable to address these challenges.
> >
> >
> > I appreciate the authors' thoughtful response and will raise my score accordingly. However, I still have reservations regarding the novelty and contribution of the methodology, and therefore, I am unable to recommend acceptance at this time.

---

> > > ### Author Response · Authors · 2025-04-09
> > >
> > > Thank you very much for your response to our rebuttal.
> > >
> > > We agree that ICON [Yang et al., 2023] explores in-context learning for surrogate modeling. We initially considered it as a potential baseline. However, we quickly found it inapplicable to our setting: in ICON, attention is applied pointwise over the input data, which becomes computationally prohibitive in our case. Our preliminary experiments on simple 1D datasets failed to converge due to this complexity, and therefore we did not retain it as a baseline.
> > >
> > > Regarding the generative aspect, ICON relies on a deterministic predictor for the physical input and does not employ a generative model. To the best of our knowledge, there are currently no prior approaches using LLM-based models for PDE modeling—where by LLM, we refer to generative probabilistic models, akin to the decoders used in language modeling, as explained in our rebuttal. That said, we would be happy to acknowledge and discuss any relevant references we may have overlooked.
> > >
> > > Finally, concerning the role and relevance of neural surrogates, we believe that the development of fast surrogate models remains an important and active area of research, with a broad range of applications. This ongoing interest is reflected in the recent surge of work in the ML community. Our research contributes to this collective effort.
> > >
> > > Yang, L., Liu, S., Meng, T., & Osher, S. J. (2023). In-context operator learning with data prompts for differential equation problems. Proceedings of the National Academy of Sciences of the United States of America, 120(39).

---

### Official Review · Reviewer_ikHe · 2025-03-12

**Overall Recommendation:** 4

**Summary:**

This paper introduces Zebra, a novel generative autoregressive transformer designed to solve parametric partial differential equations (PDEs) without requiring gradient adaptation at inference. The key innovation is leveraging in-context learning to dynamically adapt to new PDE parameters. Zebra employs a two-stage framework: first, a vector-quantized variational auto-encoder (VQ-VAE) compresses physical states into discrete tokens; second, a generative autoregressive transformer is pre-trained on these tokens with a next-token prediction objective. During inference, Zebra conditions on example trajectories with similar dynamics but different initial conditions to predict new trajectories. As a generative model, Zebra enables uncertainty quantification and trajectory sampling. The authors demonstrate Zebra's effectiveness across various PDE scenarios, showing it outperforms gradient-based adaptation methods while being computationally more efficient. They also propose an accelerated inference procedure using a UNet conditioned on a dynamics embedding, further improving speed by orders of magnitude.

**Claims And Evidence:**

The claims in the paper are generally well-supported by convincing evidence. The authors provide comprehensive experimental results across seven different PDE datasets, comparing Zebra against state-of-the-art adaptation methods (CODA, CAPE, and other baselines). The empirical evidence demonstrates Zebra's superior performance, particularly in challenging scenarios like 2D environments and out-of-distribution settings. The authors substantiate their claims about Zebra's generative capabilities through quantitative metrics (Wasserstein distance between generated and real distributions) and qualitative visualizations. The paper also provides clear evidence for the efficiency gains of their proposed accelerated inference method, with specific timing measurements showing improvements of one to two orders of magnitude.

**Essential References Not Discussed:**

The paper does not cite recent advances in generative state-space models for physical systems, such as Hao et al. (2024) "DPOT: Auto-regressive Denoising Operator Transformer for Large-scale PDE Pretraining." This work approaches similar problems using denoising techniques and offers an alternative perspective on generative pretraining for PDEs.

**Experimental Designs Or Analyses:**

I reviewed the experimental design and analyses in the paper, finding them largely sound and well-executed. The authors' one-shot adaptation experiments across seven datasets provide a comprehensive evaluation of Zebra's capabilities. Their comparison against multiple baselines (CAPE, CODA, [CLS]ViT, and ViT-in-context) establishes a fair benchmark for assessing performance.

**Methods And Evaluation Criteria:**

The proposed methods and evaluation criteria in this paper are well-suited for the parametric PDE problem. The authors carefully designed their approach to address the specific challenges of adapting to various PDE parameters without gradient updates at inference time. The evaluation uses seven diverse PDE datasets (five 1D and two 2D) that encompass various physical phenomena and parameter variations, providing a comprehensive testbed. The authors appropriately evaluate adaptation capabilities in both in-distribution and out-of-distribution settings, which is crucial for real-world applications where parameters may shift.

**Other Comments Or Suggestions:**

When describing the sequence design, the notation could be clarified to better distinguish between different trajectories and their corresponding token representations. The transition between mathematical notation and practical implementation is sometimes difficult to follow.

The paper would benefit from a more detailed discussion of computational complexity as a function of physical domain size, particularly for scaling to higher-dimensional PDEs beyond the 2D cases presented.

**Other Strengths And Weaknesses:**

Pros:
The paper demonstrates several notable strengths. Its most significant contribution is the innovative application of in-context learning principles from language models to physical systems, creating a bridge between these domains. This cross-pollination of ideas shows considerable originality and opens new avenues for solving complex physical problems without gradient-based adaptation.

Zebra's generative nature provides meaningful uncertainty quantification, which is crucial for high-stakes scientific and engineering applications. This capability differentiates it from deterministic approaches and enhances its utility for decision-making under uncertainty.

Cons:

Could be improved with more concise explanations of the transformer architecture. The technical details, while comprehensive, sometimes obscure the core conceptual innovations. Additionally, the paper would benefit from more explicit discussion of limitations, particularly regarding the types of PDEs where Zebra might underperform.

The paper lacks discussion of computational resources required for training, which is important for understanding the practical accessibility of this approach to researchers with varying computational capabilities.

**Questions For Authors:**

1. Given that the discretization of physical fields through VQ-VAE is fundamental to your approach, how does Zebra's performance depend on the fidelity of this discretization? Specifically, how would performance degrade for highly turbulent or chaotic systems where fine-scale features are critical?

2. While you demonstrate Zebra's computational efficiency compared to gradient-based methods, could you elaborate on the memory requirements and training costs?

3. The paper shows impressive performance in one-shot adaptation, but how does Zebra's performance scale with additional context examples? Is there a point of diminishing returns, and does the benefit of additional context vary across different PDE types?

**Relation To Broader Scientific Literature:**

The paper also connects to recent foundation models for physics, such as those proposed by Subramanian et al. (2023) and Herde et al. (2024), but distinguishes itself by focusing on efficient adaptation rather than comprehensive pretraining. Additionally, it relates to specific work on using discrete representations for physical systems, building upon the VQ-VAE framework introduced by van den Oord et al. (2017).
Most directly, Zebra extends the in-context operator learning approach proposed by Yang et al. (2023), but overcomes its limitations by incorporating a generative framework and developing a scalable architecture that can handle complex 2D dynamics.

**Theoretical Claims:**

The paper does not contain formal mathematical proofs for theoretical claims. It primarily focuses on empirical validation of the proposed Zebra framework through extensive experimentation

---

> ### Author Rebuttal · Authors · 2025-03-31
>
> ### 1. DPOT
> Thank you for mentioning this interesting work. We will include this reference, along with those suggested by other reviewers, in the final version and discuss how it differs from our approach and problem setting.
>
> ### 2. Explanations and Limitations
> Thanks for the comment. Clear explanations are indeed crucial. We will provide a concise overview of our model and its differences from prior work before the detailed description.
>
> As for limitations, our scaling analysis (Appendix D.4, Fig. 20) suggests the framework suits equations with diverse physical parameters and many training trajectories. In data-scarce regimes, this approach may be less effective—a general issue for data-driven models in complex physics. Additionally, discrete tokenization may be a bottleneck for simulations with significant high-frequency content. Finally, we have not tested extensions to more demanding scenarios (e.g., 3D) due to compute constraints. We will highlight these limitations more clearly in the appropriate section.
>
> ### 3. Computational Resources
> Thank you for pointing this out. In 1D, VQVAE training takes ~4 hours on an RTX 24 GB GPU, and the transformer takes ~15 hours. In 2D, these increase to ~20 h each on a single A100 80 GB. Overall, our method requires more training than vanilla ViT in-context but is comparable to meta-learning baselines.
>
> ### 4. Notations
> We agree. In particular, distinguishing between flattened and non-flattened indices can be confusing. We will revise and simplify the notations accordingly.
>
> ### 5. Complexity
> This is a great point. Our method applies attention over the full sequence—i.e., context examples $N$, timesteps $T$, and tokens per frame $h \times w$. Adding a third spatial dimension $d$ yields a complexity of $\mathcal{O}((NThwd)^2)$. This can result in long sequences, and more demanding cases may require architectural changes, e.g., axial transformers. We will discuss this further.
>
> ### Q1: VQVAE
> We agree. Fidelity depends on codebook size: larger sizes reduce discretization error. Appendix Fig. 21 and Table 13 show how reconstruction and prediction errors evolve with codebook size. While reconstruction improves with larger codebooks, one-shot prediction shows a trade-off—e.g., for Burgers' equation, the best size is 64. Larger codebooks increase transformer classifier complexity, as the number of parameters scales with the number of classes.
>
> As for the behavior for turbulent systems, we performed OoD experiments for the vorticity equation. Here are the results:
>
> | **Viscosity Range**              | **Type**             | **Reconstruction** | **Prediction** |
> |----------------------------------|----------------------|--------------------|----------------|
> | $[10^{-2},\ 10^{-3}]$            | In-distribution      | 0.02               | 0.12           |
> | $[5 \times 10^{-4},\ 10^{-3}]$ | Close OOD            | 0.13               | 0.24           |
> | $[10^{-5},\ 10^{-4}]$          | Far OOD              | 0.22               | 0.32           |
>
> As the viscosity decreases and moves out of the training distribution, the reconstruction error increases, which is expected since the VQ-VAE was not exposed to these regimes. However, the one-shot prediction error increases less sharply. This suggests that the encoder still captures the main structures of the flow that are important for prediction. The transformer can rely on these latent representations to make accurate forecasts, even if the reconstruction is less precise. Overall, the model shows pretty good robustness to distribution shifts.
>
> ### Q2: Memory Requirements
>
> The memory requirements remain manageable for 1D experiments. However, in 2D, they become more demanding, as training sequences can reach lengths of up to 8192 tokens. To handle this, we use a single A100 GPU with 80 GB of memory, which is sufficient to train the model effectively. For reference, this sequence length is comparable to that used during the pretraining phase of LLaMA 3.
>
> ### Q3: Number of Examples
> Great remark. We ran additional experiments varying the number of context examples on 1D datasets. We can see that the performance reaches a plateau after about 3 in-context examples:
>
> | # Examples | Advection | Heat   | Burgers | Combined |
> |------------|-----------|--------|---------|----------|
> | 1          | 0.0074    | 0.1563 | 0.1150  | 0.0095   |
> | 2          | 0.0077    | 0.1310 | 0.1020  | 0.0074   |
> | 3          | 0.0072    | 0.1272 | 0.1000  | 0.0078   |
> | 4          | 0.0071    | 0.1272 | 0.0990  | 0.0075   |
> | 5          | 0.0071    | 0.1310 | 0.1000  | 0.0073   |

---

### Official Review · Reviewer_qxAk · 2025-03-13

**Overall Recommendation:** 4

**Summary:**

This paper proposes an in-context generative auto-regressive transformer for solving parametric PDEs and outperforms gradient adaptation methods.

**Claims And Evidence:**

Given that the authors claim this is the first successful application of discretized representations in physical systems, they should further analyze the impact of discretized representations on modeling physical systems. For example, an ablation study could be conducted to compare it with continuous representation.

**Essential References Not Discussed:**

L84. There are other state-of-the-art generative models for PDE simulation. [1, 2, 3]

[1] Huang J, Yang G, Wang Z, et al. DiffusionPDE: Generative PDE-Solving under Partial Observation. The Thirty-eighth Annual Conference on Neural Information Processing Systems.

[2] Peiyan Hu, Rui Wang, et al. Wavelet Diffusion Neural Operator. The Thirteenth International Conference on Learning Representations.

[3] Mario Lino Valencia, Tobias Pfaff, Nils Thuerey. Learning Distributions of Complex Fluid Simulations with Diffusion Graph Networks. The Thirteenth International Conference on Learning Representations.

**Experimental Designs Or Analyses:**

For the ‘New trajectory generation’ experiment in Section 4.4, a more accurate evaluation should involve inputting the initial state and other parameters of the generated sequence into the solver to obtain the true trajectory, and then comparing it with the generated trajectory.

**Methods And Evaluation Criteria:**

For the experiment in Section 4.3 ‘Out-of-distribution generalization,’ could the prediction accuracy be explored when the external force term differs in form from the external force terms in the training dataset? In other words, when the external force term is no longer a sum of sine functions?

**Other Comments Or Suggestions:**

It would be clearer to plot differences between the ground truths and predictions in Figure 2.

**Other Strengths And Weaknesses:**

Strengths:

The paper is overall novel as it explores a new direction. Moreover, its experiments are comprehensive and thorough, and the writing is clear.

Weaknesses:

More analysis of the encoder-decoder architecture would be better, as the current analysis is quite limited, yet it is an important part of the algorithm.

**Questions For Authors:**

Sec 4.1. Why use a lot more unseen environments for the 2D datasets (120) than for the 1D datasets for testing (12)? There should be an explanation.

**Relation To Broader Scientific Literature:**

This paper explores a new direction in PDE simulation, namely in-context learning, which differs from previous works, and they also use discretized representations.

**Theoretical Claims:**

The paper does not have theoretical claims.

---

> ### Author Rebuttal · Authors · 2025-03-31
>
> ### 1. Continuous vs Discrete
>
> Thanks for the comment. We agree that encoder quality is crucial for model performance. However, the answer is not trivial and we will try to answer in different points.
>
> * Our goal was to build an autoregressive probabilistic predictor for parametric PDE dynamics that adapts to new dynamics (new values of the PDE parameters). We took inspiration from in-context learning in language modeling, which relies on discrete tokens from a finite vocabulary.
> * *Discrete representations*: To evaluate encoder quality, we tested different codebook sizes. Larger codebooks preserve more information (closer to continuous encoding), as shown in Fig. 21 and Table 13. There is a tradeoff between reconstruction quality (improved with larger codebooks) and forecasting performance. We used a codebook size of 256 as a compromise across 1D datasets.
> * *Continuous representations*: A generative model on continuous representations would require predicting distributions over continuous states at each time step. We initially explored this, but it remains an open problem. Recent attempts for image generation [1] do not yet translate to the more complex problem of modeling PDE dynamics. Thus, a direct comparison with continuous autoregressive probabilistic models is not feasible.
> * It is however possible to compare with deterministic approaches operating on continuous representations, and this is what we did with the vision transformers (ViT) in tables 1 and 2. Additionally, Appendix D.1 (Fig. 15) shows that using a deterministic transformer to predict the next token over continuous latent representations performs poorly in the one-shot adaptation setting.
>
> [1] Autoregressive Image Generation without Vector Quantization, Li et al. 2024.
>
> ### 2. New forcing terms for Heat
>
> Great point. We added experiments with out-of-distribution (OoD) forcing terms: **Gaussian** and **Square** patterns.
>
> * **Gaussian:**
> We define forcing as a sum of spatial Gaussian pulses modulated in time:
> $$
> f(x, t) = \sum_{i=1}^n A_i \exp\left(-\frac{(x - \mu_i)^2}{2\sigma_i^2} \right) \sin(\omega_i t)
> $$
> This introduces smooth but localized excitation. Solutions resemble the homogeneous case in regions far from peaks, making this a "close" OoD test.
>
> * **Square:**
> We define forcing as a sum of square waves:
> $$
> f(x, t) = \sum_{i=1}^n A_i \text{sign} \left( \sin\left( \omega_i t + \frac{2\pi l_i x}{L} + \phi_i \right) \right)
> $$
> This preserves parameterization but introduces discontinuities and high-frequency content, making it a more challenging OoD case.
>
> We evaluate both without retraining, using context examples from these new trajectories. Results below compare ViT-in-context and Zebra:
>
> | Model           | Gaussian Forcing | Square Forcing |
> |----------------|------------------|----------------|
> | ViT-in-context | 0.4023           | 0.6584         |
> | Zebra          | **0.3263**       | **0.3648**     |
>
> While both models show performance degradation, Zebra remains significantly more robust than the baseline.
>
> ### 3. New trajectory generation
>
> Our apologies if this was unclear. This actually corresponds to the fidelity metric described in Appendix D.3 and we will indicate it more clearly in the main text. Table 10 in the appendix reports the average L2 distance between the generated trajectory and the ground truth trajectory, which is simulated using the numerical solver with the true physical parameters (inaccessible to Zebra) and the initial condition generated by Zebra. These results show that our model can produce consistent trajectories for a given set of physical parameters.
>
> ### 4. Experiments from Appendix D.
>
> Thanks for the suggestion. We agree and will reference these findings directly in the main text.
>
> ### 5. Generative models
>
> Thanks, we will include these references in a section on SOTA generative PDE simulators.
>
> ### 6. Encoder-decoder analysis
>
> We agree that encoding quality is key. As discussed in point 1, we analyzed the impact of codebook size on reconstruction and forecasting.  We found out that although the quality of the encoding benefits from larger codebooks, there is a compromise between the codebook size and the quality of the forecast. VQ-VAE offers a strong balance between performance and complexity. Let us know if any further analysis would be valuable for the camera-ready version.
>
> ### 7. Plot differences
>
> Agreed, we will add additional plots to show the differences between ground truths and predictions.
>
> ### Q1. # of environments.
>
> Actually, this was arbitrary. We can align all test settings to 120 environments for the camera-ready version; this will not significantly change the results.

---

> > ### Comment · Reviewer_qxAk · 2025-04-05
> >
> > Thanks for your responses. Most of my concerns are addressed and I believe that the paper will be improved after the revision. So I raise my score to 4.

---

### Official Review · Reviewer_6LCA · 2025-03-15

**Overall Recommendation:** 4

**Summary:**

The paper addresses the problem of predicting PDE solutions without explicit knowledge of the underlying dynamics using a generative auto-regressive transformer. The model is trained using contexts composed of different trajectories. During inference, the model is given a context containing trajectories governed by the same dynamics and the initial condition to evolve.

**Claims And Evidence:**

The main claims in the paper are well-supported by clear and convincing evidence. In particular, the pretraining strategy is novel in its application to PDEs and demonstrates strong results in one-shot adaptation compared to alternative adaptation methods.

However, the claim regarding the benefits of using a VQ-VAE to provide the model with generative capabilities could be better substantiated (see the primary weakness below).

**Essential References Not Discussed:**

All essential references have been discussed.

**Experimental Designs Or Analyses:**

The main figures and tables of the paper are sound and correctly support the main claims of the paper.

- Table 1 and Figure 2 demonstrate the zero-shot capabilities of the model
- Table 2 presents compelling out-of-distribution results

The experiments assessing the generative aspect of the model, particularly in uncertainty quantification (Figure 3) and distribution quality (Figure 4, Table 3), are a bit weak.

- Figure 3: It is unclear what this figure illustrates beyond showing that the generative model produces a reasonable mean prediction, with the ground truth contained within three standard deviations. A more convincing validation would involve a scenario where the true uncertainty is known and verifying whether the model accurately reproduces that uncertainty. However, the dataset used likely does not provide ground-truth uncertainty levels.

- Figure 4 and Table 3: Their purpose is unclear. If they aim to demonstrate that the model produces states that fall within the training distribution, this is a relatively weak result. Any model achieving reasonable validation accuracy should satisfy this criterion, provided the validation set is not significantly different from the training set.

**Methods And Evaluation Criteria:**

The methods and evaluation criteria are appropriate for supporting the paper’s main claims.

**Other Comments Or Suggestions:**

- The paper could emphasize more clearly that it tackles PDE prediction from an initial state only. At inference, even though the context contains trajectories with the correct dynamics, the model is given only an initial state from the trajectory to evolve. Since the model is presented with only one state to evolve, this task is technically more challenging than predicting a trajectory with multiple past states (which explains I think why you did not compare to (McCabe et al. 2023)). However, this also makes it a more restricted task, and this should be explicitly stated. For example, if a single frame does not fully capture the system’s state (e.g., missing pixels, incorrect high frequencies), having more past frames would reduce this uncertainty, but that falls outside your proposed task.
- l074: "provides richer information than deterministic...". This holds only if the conditional distributions are correctly calibrated, which the paper does not adequately demonstrate. Otherwise, generating a conditional distribution is trivial, but proper calibration is key.
- l079: "scales better than gradient-based adaptation approaches"—vague. What aspect of scaling are you referring to?
- l163: Unclear what you mean by "into index representations."
- l181: The figure is confusing—why does the generated trajectory seem unrelated to the query? Is there an issue?
- l182: The caption is poorly written and should be clarified. Example: "Sequences have similar dynamics but different initial conditions, generating trajectories that follow the same underlying physical behavior from a new initial condition (query)", "according to the sequence design used."
- l189: What does "sequence" refer to here? A sequence of image patches? A sequence of states? The explanation of the pretraining strategy should be clearer.
- l202: Double "((" and "))".
- l222: Why is there no <eot> token before the last <bot> token?
- l257: The main text should specify the resolution of the states used.
- l298: "Deterministic models tend to predict conditional 'blurry' expectations..."—this is not a valid argument. Blurriness in deterministic models can result from poor high-frequency reconstruction by the autoencoder, an issue that a VQ-VAE may also face. If you mean that a VQ-VAE reconstructs high frequencies faster, this needs to be demonstrated.
- l338: Figure 2 appears to have an issue—captions or y-labels seem misaligned in the first column.

**Other Strengths And Weaknesses:**

Strengths:

- The in-context pretraining strategy is quite relevant to the ML & PDE community, offering the potential for training larger models with enhanced in-context capabilities.

- The reduction in inference costs proposed by the model is valuable to the field.

Weaknesses:

- The justification for incorporating a variational autoencoder in the architecture is unclear, as uncertainty quantification is not a central focus. Moreover, the uncertainty appears to apply only to individual states rather than the overall prediction. If each frame is treated independently, the model merely generates independent random perturbations per frame, rather than meaningful uncertainty estimates over entire trajectories.

**Questions For Authors:**

- What proportion of the contexts in the training dataset contain trajectories from different dynamics?
- Your model should enable strong one-shot performance without the need for retraining (unlike CODA, Poseidon, MPP). However, this obscures the fact that your model is trained on a much broader variety of contexts, including multiple trajectories, sometimes from different dynamics. Could you comment on how much additional training is required for your method to develop in-context capabilities, compared to (McCabe et al. 2023) for example?

**Relation To Broader Scientific Literature:**

Compared to gradient-based adaptation techniques such as (e.g. CODA paper, Kirchmeyer et al. 2022), which require retraining for every new environment, the paper introduces an approach capable of tackling zero-shot problems through in-context learning. The proposed model builds upon the transformer introduced in (McCabe et al. 2023) by enhancing its in-context capabilities, enabling it to predict from a single initial condition, which this model cannot achieve.

**Theoretical Claims:**

The paper does not introduce theoretical claims in the sense of proving a new theorem.

---

> ### Author Rebuttal · Authors · 2025-03-31
>
> ### 1. On the uncertainty
>
> Thank you for your comments. We agree that the main contribution lies in adaptation through in-context learning, with uncertainty emerging as a byproduct. We aim to clarify this distinction and welcome suggestions for improving it.
>
> The generative capacity stems from the transformer, not the VQ-VAE. Our decoder-only transformer, as in language modeling, predicts the next token from past tokens and samples from this distribution. Since each sampled token depends on previous ones, uncertainty propagates through time.
>
> The VQ-VAE encodes continuous frames into discrete tokens and compresses the input, making transformer training feasible. It strikes a balance between fidelity and compression. In contrast, deterministic vision transformers (e.g., ViT, MPP, Poseidon) lack generative ability and do not benefit from in-context examples as our model does (see Table 1).
>
> Although uncertainty modeling is not our focus, we found it relevant to highlight Zebra’s ability to sample diverse trajectories (e.g., Fig. 3). Below, we evaluate uncertainty quality using CRPS and RMSCE. CRPS measures accuracy of probabilistic predictions; RMSCE evaluates calibration when ground-truth uncertainties are unknown. Lower is better. Zebra consistently outperforms baselines, often by an order of magnitude:
>
> * CRPS and RMSCE Results
>
> | Metric | Model        | Advection | Heat  | Burgers | Wave b | Combined |
> |--------|--------------|-----------|-------|---------|--------|----------|
> | CRPS   | ViT + noise  | 0.0705    | 0.176 | 0.227   | 0.093  | 0.098    |
> |        | ViT Dropout  | 0.0363    | 0.213 | 0.196   | 0.024  | 0.024    |
> |        | Zebra        | **0.0026**| **0.043** | **0.020** | **0.0129** | **0.0018** |
> | RMSCE  | ViT + noise  | 0.132     | 0.241 | 0.265   | 0.249  | **0.045** |
> |        | ViT Dropout  | 0.386     | 0.547 | 0.529   | 0.340  | 0.064    |
> |        | Zebra        | **0.074** | **0.055** | **0.048** | **0.124** | 0.074    |
>
> Below, CRPS and RMSCE over time for Zebra on Burgers. As expected, CRPS grows with error accumulation. RMSCE remains stable, indicating consistent calibration.
>
> | Dataset   | Metric | t=1    | t=2    | t=3    | t=4    | t=5    | t=6    | t=7    | t=8    | t=9    |
> |-----------|--------|--------|--------|--------|--------|--------|--------|--------|--------|--------|
> | **Burgers**   | CRPS | 0.0057 | 0.0106 | 0.0148 | 0.0184 | 0.0216 | 0.0244 | 0.0271 | 0.0296 | 0.0320 |
> |            | RMSCE | 0.0511 | 0.0513 | 0.0505 | 0.0487 | 0.0483 | 0.0489 | 0.0467 | 0.0457 | 0.0457 |
>
> ### 2. New trajectory generation
>
> This experiment is exploratory. The model generates a full trajectory, including its initial state, when prompted with examples (no initial state given). Multiple diverse outputs can be sampled from the same context. See Appendix D.3, Table 10.
>
> ### 3. Clarifying the scope
>
> We will clarify this in the introduction. Our focus on forecasting from an initial state highlights the role of context.
>
> ### 4. "provides richer information..."
>
> Agreed. We will remove the sentence.
>
> ### 5. On scaling gradient-based adaptation
>
> Our concern is the high computational cost of meta-learning with double-loop optimization. Alternatives like CODA avoid this but drastically increase parameters count. Our approach avoids both issues.
>
> ### 6. Index representations
>
> We agree this needs clearer explanation. Lines 146–152 mention that each frame is encoded into tokens via VQ-VAE. These tokens come from a finite vocabulary and are replaced by indices $s$ for processing in the transformer—similar to language modeling.
>
> ### 7. On the main figure
>
> We only show final timesteps, which may seem unrelated. This will be clarified in the caption. See Appendix E.6 for better illustrations.
>
> ### 8. Caption revision
>
> The model takes as input example trajectories $u_1$, $u_2$ and a new initial condition $u^0_*$. These are tokenized into $s_1$, $s_2$, $s^0_*$, concatenated, and fed into a transformer that predicts next tokens. The output is then detokenized into a physical trajectory.
>
> ### 9. "Sequences"
>
> We will clarify this. Each $s_i$ is a sequence of VQ-VAE indices.
>
> ### 10. <eot> token
>
> We will try to include it, possibly with smaller font.
>
> ### 11. Resolution
>
> We used 64×64 in 2D and 256 in 1D.
>
> ### 12. "Blurry" expectations
>
> We will remove this term if it’s unclear. We meant deterministic models tend to predict conditional expectations, which average over possibilities.
>
> ### 13. Fig. 2 issue
>
> We will fix it.
>
> ### 14. Different dynamics
>
> Each context uses a single dynamic. During training, we sample 1–5 examples per context. The dataset has ~1200 dynamics, each with 10 trajectories. For better GPU usage, we stack sequences from different dynamics, but this is marginal.
>
> ### 15. ICL training
>
> We train across a variety of dynamics. This differs from MPP and Poseidon, which train across physics types. In our experiments (e.g., Table 1), all methods use the same data, and Zebra outperforms ViTs.

---

> > ### Comment · Reviewer_6LCA · 2025-04-04
> >
> > I thank the authors for their detailed rebuttal, in particular for the additional compelling experiments and the interesting comments on uncertainty quantification.

---

### Decision · Program_Chairs · 2025-05-01

**Decision:**

Accept (poster)

**Comment:**

This paper introduces a novel type of transformer for in-context inference of parametric PDEs using  specific pre-training strategies and providing uncertainty estimates.

While there were some concerns initially about the evaluations and baselines used, most of these it seems were resolved during the rebuttal and discussion period, although it appeared to me not all of them. For instance, ref. 6LCA raised issues about the material presented in Figs. 3 & 4 which I agree with, but the authors didn’t really comment much on this in their rebuttal. Especially Fig. 4, which only shows PCA projections and apparently broad distributions (hard to judge, given lack of axes scales and a clear reference), I don't find convincing.

Also, in my mind some of the baselines used (CODA, Kirchmeyer et al., 2022; CAPE, Takamoto et al. 2023) are somewhat older, given the fast development in this field, and not quite SOTA anymore. I would like to encourage the authors to carefully check the most recent batches of NeurIPS and ICLR papers on meta- & few-shot learning of dynamical systems, and potentially include some of this work at least in the discussion. Finally, one referee stressed the paper may also need a better contextualization within the broader lit. on PDE solvers and better discussion of limitations (e.g., lack of interpretability or error bounds).

In general, however, while one referee remained skeptical and also expressed some concerns about novelty, three referees clearly advocated acceptance.
I will go with the referees’ majority vote and suggest acceptance, but would appreciate if the points above were acknowledged or addressed in the revision.